



# A learning-based method for efficient large-scale sensitivity analysis and tuning of single column atmosphere model (SCAM)

Jiaxu Guo[1,6], Yidan Xu[5,7], Haohuan Fu[2,6], Wei Xue[3,6], Lanning Wang[4,6], Lin Gan[3,6], Xianwei Wu[1,6], Liang Hu[1], Gaochao Xu[1], and Xilong Che[1]

[1]College of Computer Science and Technology, Jilin University, Changchun, China
[2]Department of Earth System Science, Ministry of Education Key Laboratory for Earth System Modeling, Tsinghua University, Beijing, China
[3]Department of Computer Science and Technology, Tsinghua University, Beijing, China
[4]College of Global Change and Earth System Science, Beijing Normal University, Beijing, China
[5]China Reinsurance (Group) Corporation, Beijing, China
[6]National Supercomputing Center in Wuxi, Wuxi, China
[7]School of Environment and Nature Resources, Renmin University of China, Beijing, China

**Correspondence:** Haohuan Fu (haohuan@tsinghua.edu.cn), Liang Hu (hul@jlu.edu.cn) and Xilong Che (chexilong@jlu.edu.cn)

**Abstract.** The Single Column Atmospheric Model (SCAM) is an essential tool for analyzing and improving the physics schemes of CAM. Although it already largely reduces the compute cost from a complete CAM, the exponentially-growing parameter space makes a combined analysis or tuning of multiple parameters difficult. In this paper, we propose a hybrid framework that combines parallel execution and a learning-based surrogate model, to support large-scale sensitivity analysis (SA) and tuning of combinations of multiple parameters. We start with a workflow (with modifications to the original SCAM) to support the execution and assembly of a large number of sampling, sensitivity analysis, and tuning tasks. By reusing the 3,840 instances with the variation of 11 parameters, we train a neural network (NN) based surrogate model that achieves both accuracy and efficiency (with the computational cost reduced by several orders of magnitude). The improved balance between cost and accuracy enables us to integrate NN-based grid search into the traditional optimization methods to achieve better optimization results with fewer compute cycles. Using such a hybrid framework, we explore the joint sensitivity of multi-parameter combinations to multiple cases using a set of three parameters, identify the most sensitive three-parameter combination out of eleven, and perform a tuning process that reduces the error of precipitation by 5% to 15% in different cases.

## 1 Introduction

Earth System Models (ESMs) are important tools to help people recognize and understand the global climate change. Community Earth System Model (CESM) is one of the most popular and widely used ESMs, which includes atmosphere, ocean, land, and other components (Hurrell et al., 2013). Of these components, the Community Atmosphere Model (CAM) (UCAR., 2020), is the one with the most complexity.



**Figure 1.** The overall workflow of the proposed method. Part I performs the sampling and the collection of results of parallel instances. Part II uses traditional SA methods to derive sensitivities of individual parameters, and at the same time, reuse the samples to derive a Neural Network (NN) based surrogate model. Combining NN-based surrogate model, we can then also perform joint sensitivity analysis of a set of parameters. Guided by the SA results from Part II, Part III performs parameter tuning, also with the NN-based surrogate model. SCAM launcher, the data collector and the jobs therein represent the batch execution of the SCAM algorithm, which is described in more detail in Figure 2.



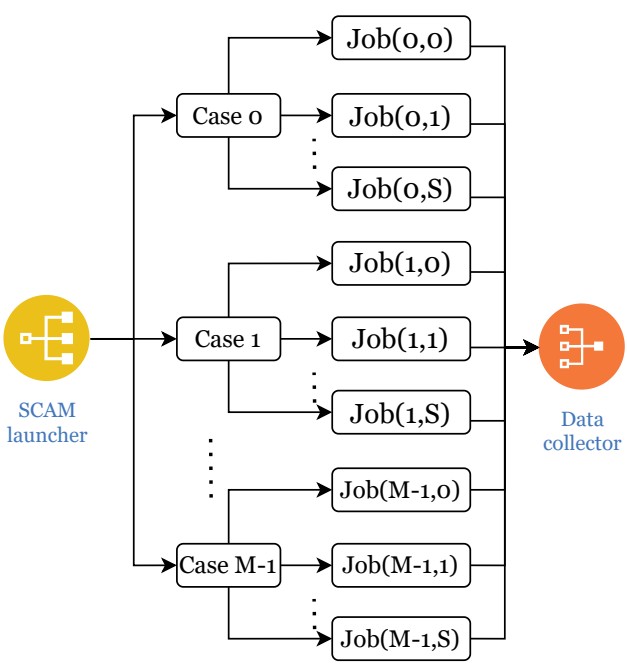

**Figure 2.** Detailed parallelism schematic when running a large number of instances of multiple cases simultaneously. In this process the required SCAM tasks are launched simultaneously by the launcher and the results of their runs are collected by the data collector.

Most of the physics parts in CAM are described as parameterization schemes, with tunable parameters that are often de-
rived from limited measurements or theoretical assumptions. Participated in continuous numerical integration, these tunable parameters become a major source of the model uncertainty, and need to be tuned carefully (Yang et al., 2013).

However, as a general circulation model (GCM), CAM takes a long time and a large amount of resource to run (Zhang et al., 2018). Thus, Single Column Atmospheric Model (SCAM) (Bogenschutz et al., 2013; Gettelman et al., 2019) has been developed as a good alternative model for the purpose of tuning physics parameters (Bogenschutz et al., 2020). In contrast to
the 48,602 columns in a $ne30$ configuration of CAM, SCAM only needs to compute one single column with one process. As a result, SCAM becomes a natural tool for studying how the parameters would affect the uncertainty in the modeling results, and how to reduce such uncertainties accordingly.

Sensitivity analysis (SA) is a method for investigating how uncertainty in the model output is assigned to the different sources of uncertainty in the model input factors, and the participants (Saltelli et al., 2010). A rich set of numerical and
statistical methods have been developed over the years to study the uncertainty in models in many different domains, ranging from natural sciences, to engineering, and risk management in finance and social sciences (Saltelli et al., 2008).

Climate models are among some of the most complex models, and continuous efforts have been put into the SA of tunable parameters in climate models, especially for the their physics parameterization schemes (Yang et al., 2013; Guo et al., 2015;





Pathak et al., 2022). SA of climate models generally involves two steps: generating representative samples with different values

of parameters using a specific sampling method; and explore and identify the sensitivity metrics between the model output and the parameters to study. Typical approaches include: the Morris One-At-a-Time (MOAT) method that uses the Morris sampling scheme (Morris, 1991), which generates samples uniformly and has a good compute efficiency, but lacks the capability to study the interaction between different parameters (Pathak et al., 2020); and the Sobol method that uses the probabilistic framework and adopts the decomposition of variance of the output to describe the sensitivities (Saltelli et al., 2010). In contrast to the

Morris method, the variance-based Sobol method generally requires a lot more samples to achieve a good coverage of the space (even by using the low-discrepancy quasi-random sequence by Sobol (Sobol', 1967) and other researchers), but has the advantage of being capable to study the interaction effects between different parameters. Other similar ideas to achieve a good representation of the sample space with a quasi-random sequence include the quasi-Monte Carlo (QMC) (Caflisch, 1998) and the Latin hypercube (LHC) (McKay et al., 2000) sampling methods. A key problem to consider for the SA stage is to achieve

a balance between the accuracy and the economy of compute (Saltelli et al., 2008). People sometimes use surrogates (such as the generalized linear model (GLM) (Nelder and Wedderburn, 1972)) instead of the actual model to further reduce the compute cost.

After we identify the important tuning targets in the SA stage, we can then tune the parameters to improve the performance of the model. With a general goal to achieve modeling results as close to the observations as possible, we can apply different

optimization methods, such as the Genetic algorithm (GE) (Mitchell, 1996), differential evolution (DE) (Storn and Price, 1997), particle swarm optimization (PSO) (Kennedy and Eberhart, 2002), etc., to identify the most suitable set of parameters.

For both the SA and the tuning stages, the compute cost of running the model become a major constraining factor that stop us from exploring more samples and identifying more optimal solutions. Even for the single column model, existing efforts would focus on evaluating the sensitivity of each single parameter, or a given set of multiple parameters, and usually

investigate only a couple of cases. For example, the study of the sensitivity of simulated shallow cumulus and stratocumulus clouds to the tunable parameters of the subnormal uniform cloud layer (CLUBB) (Guo et al., 2015) investigated the sensitivity of 16 specific parameters, using the QMC sampling method and GLM as a surrogate, with experiments on three different cases (BOMEX, RICO, and DYCOMS-II RF01). Another study (Pathak et al., 2022) used the single-column case (ARM97) to explore 8 parameters related to the cloud processes, with Sobol as the sampling method, and spectral projection (SP) and basis

pursuit denoising (BPDN) as the surrogate model.

In an ideal case, a more thorough study of the parameters that can provide more concrete guidance for the parameter selection in CAM, would require a joint SA and tuning of different single-column cases, as well as combinational study of the most sensitive parameters. However, such a joint and combined exploration that involve multiple parameters and multiple cases would increase the space to explore in an exponential manner, and make the SA and tuning almost an impossible job.

To facilitate researchers to better operate SCAM, and to support a more efficient and convenient parameter tuning for the physical schemes in SCAM, we propose a learning-based method for efficient large-scale SA and tuning, which can support parallel execution of hundreds to thousands of parallel instances, and highly-efficient exploration of combinations of multiple parameters.





**Table 1.** List of single column atmosphere model cases tested.

| Case | Full name | Lat | Lon | Date | Type |
|---|---|---|---|---|---|
| ARM95 | ARM Southern Great Plains | 36 | -97 | July 1995 | Land convection |
| ARM97 | ARM Southern Great Plains | 36 | -97 | June 1997 | Land convection |
| GATEIII | GATE Phase III | 9 | -24 | August 1974 | Tropical convection |
| TOGAII | Tropical W. Pacific Convection | -12 | 131 | December 1992 | Tropical convection |
| TWP06 | Tropical Ocean Global Atmosphere | -2 | 154 | January 2006 | Tropical convection |

**Table 2.** Observed variables included in the IOP file of each case.

| Variable | Description | ARM95 | ARM97 | GATEIII | TOGAII | TWP06 |
|---|---|---|---|---|---|---|
| Prec | Precipitation rate | √ | √ | √ | √ | √ |
| totcld | Total cloud | √ | √ | - | - | √ |
| shflx | Surface sensible heat flux | √ | √ | - | √ | √ |
| lhflx | Surface latent heat flux | √ | √ | - | √ | √ |

We start with a scientific workflow (with modifications to the original SCAM) to support the execution and assembly of a large number of sampling, sensitivity analysis, and tuning tasks. By reusing the 3,840 instances with variation of 11 parameters, we train a NN surrogate model that achieves both accuracy (with an error within 10%) and efficiency (with the computational cost reduced by several orders of magnitude). The improved balance between cost and accuracy enables us to integrate NN-based grid search into the traditional optimization methods to achieve better optimization results with fewer compute cycles.

The main contributions of this paper are:

- We enable a scientific workflow (with modifications to the original SCAM) to support the configuration of parameters through the $namelist$, and execution and assembly of a large number of sampling, sensitivity analysis, and tuning tasks.

- By reusing the 3,840 sampling instances in the Sobol sampling method, we train a Neural Network (NN) based surrogate model that achieves both a good accuracy (with an error within 10%) and a computational cost reduced by several orders of magnitude, which enables us to do sensitivity analysis of combinations of multiple parameters in an efficient way.

- The improved balance between cost and accuracy further enables us to integrate NN-based grid search into the traditional optimization methods. With a better capability to jump out of local optimums, we can achieve better optimization results with fewer compute cycles.

Using our proposed learning-based method, we perform an extensive set of SA and tuning experiments for five cases of SCAM (both independently and jointly), targeting the precipitation performance.





**Table 3.** List of parameters in the framework that can be tuned and applied to the experiment.

| Abbr. | Name | Description | Low Range | Default | High Range | Category |
|-------|------|-------------|-----------|---------|------------|----------|
| pz1 | c0_lnd | Deep convection precipitation efficiency over land | 0.00295 | 0.0059 | 0.00885 | ZM Deep convection |
| pz2 | c0_ocn | Deep convection precipitation efficiency over ocean | 0.0225 | 0.045 | 0.0675 | ZM Deep convection |
| pz3 | ke | Evaporation efficiency of precipitation | 5e-7 | 1e-6 | 1.5e-6 | ZM Deep convection |
| pz4 | tau | Time scale for consumption rate deep CAPE | 1800 | 3600 | 5400 | ZM Deep convection |
| pz5 | capelmt | Threshold value for CAPE | 35 | 70 | 105 | ZM Deep convection |
| pz6 | alfa | Maximum cloud downdraft mass flux fraction | 0.05 | 0.1 | 0.15 | ZM Deep convection |
| pu1 | rpen | Penetrative updraft entrainment efficiency | 2.5 | 5.0 | 7.5 | UW Shallow convection |
| pu2 | kevp | Evaporative efficiency | 1e-6 | 2e-6 | 3e-6 | UW Shallow convection |
| pu3 | rkm | Updraft lateral mixing efficiency | 7 | 14 | 21 | UW Shallow convection |
| pc1 | rhminh | Threshold relative humidity for stratiform high clouds | 0.7 | 0.8 | 0.9 | Cloud fraction |
| pc2 | rhminl | Threshold relative humidity for stratiform low clouds | 0.7975 | 0.8975 | 0.9975 | Cloud fraction |

Besides SA analysis that provides sensitivity evaluation of each single parameter, we are also able to study the sensitivity of a combination of three, four or even five arbitrary parameters.

    At the tuning stage, our improved optimization scheme (targeting the same parameters) lead to 30% more accurate simulation results, with a more than 30% saving in compute cost compared to using only the optimization algorithm.

    Moreover, we apply our method to tune five different cases with a unified or a case-specific set of parameters. Results
demonstrate that case-specific tuned parameters would further reduce the precipitation error by 15% when compared to a set of unified tuned parameters, and suggest a potential improvement from location-wise parameter tuning in the future.

## 2   Enabling a workflow of SA and parameter tuning

### 2.1   Model description

This paper focuses on the single column model of the atmospheric model CAM5, i.e. SCAM5, extracted from CESM version
1.2.2, one of the two versions that are efficiently supported on the Sunway TaihuLight Supercomputer (Fu et al., 2016).

    Our research of this paper is mainly based on five typical cases in SCAM5, as shown in Table 1. As shown in Table 2, the number of observations included in the IOP (Intensive Observation Periods, Gettelman et al. (2019)) file varies from case to case. In order to explore a joint parametric sensitivity analysis and tuning across all the five cases, we pick the intersection of the data owned by these cases, the total precipitation (PRECT), which is also one of the most important outputs of the model,
as the main research subject.





Among the five cases, two cases are located in the Southern Great Plains, which are parts of the Atmospheric Radiation Measurement (ARM) (Zhang et al., 2016), and mainly study land convection. The other three cases are located in the tropics and mainly study tropical convection (Thompson et al., 1979; Webster and Lukas, 1992; May et al., 2008).

The parameters listed in Table 3 are the main study targets (Qian et al., 2015) in this paper and the ones tested in the workflow.
The parameters are selected from the ZM deep convection scheme (Zhang, 1995), the UW shallow convection scheme (Park, 2014), and cloud fraction (Gettelman et al., 2008).

In the original version of SCAM, only some of the parameters to be studied were tunable, while the rest were hard-coded in the code. To improve the flexibility of the model so that the 11 parameters we want to study are tunable, we have modified the source code of the model, to support the tuning and study of a wider range of parameters. The corresponding Fortran source
code, as well as the XML documentation were also modified accordingly.

In addition, the programs running on Sunway TaihuLight needed to be recompiled due to the adoption of a different architecture. We recompiled using a compiler compatible with Sunway after making the above improvements, to enable execution of a larger number of concurrent instances on the Sunway supercomputer. With these upgrades, all parameters in Table 3 are supported for tuning.

## 115 2.2 The workflow of SA and parameter tuning

The submission of assignments and the collection of results is an important issue when carrying out a large number of model experiments at the same time. Prior to conducting the experiments, the user is often presented with a broad set of boundaries of the parameters to be tuned, and the specific configuration of each experiment has to be decided in detail according to these ranges of values. After a large number of experiments have been completed, how to collect the simulation results scattered
across the tests in a centralised manner pending further delineation. This would be very time consuming if done manually. It is therefore necessary to provide the researcher with an automated experiment-diagnosis process. In general, which parameters to tune and how to tune them are questions that deserve our attention.

Based on the above needs, we have designed the SCAM parameter tuning and analysis workflow. It supports a fully-automated parameter tuning and diagnostic analysis process, a large number of concurrent model tests, and the search for
the best combination of parameter values for SCAM performance within a given parameter space. Also, with the help of the training surrogate model, more parameter fetches can be tested in less time. The simulation results of the real model will be used as validation.

The overview of the whole scientific workflow is shown in Figure 1. In order to make full use of computing resources and complete the sampling process as soon as resources allow, the proposed method supports parallel sampling processes, as shown
in Figure 2. If computing resources are available, ideally, it only takes the time to run one simulation to complete the sampling.

Specifically for the application scenario in this paper, the execution process of the workflow is as follows.

1. User configuration: Users specify the required cases to test, the output variables to study, the parameters to analyze and their possible range of values, the number of simulations, etc. in the configuration files.





2. Sampling and data collection (shown as Part I in Figure 1): In this part, our tool generates the sequence of samples to investigate in the sampling step. Our tool currently supports the Sobol sequence, which can later be used by the Sobol, Delta (Plischke et al., 2013), HDMR (Li et al., 2010), and RBD (Plischke et al., 2013) SA methods, and the Morris sampling sequence, which can later be used by the MOAT method (Morris, 1991). Users are suggested to adjust the size of the sequence according to the currently available computational resources. As the results of this step will be used as the training set for generating the surrogate model, users are encouraged to run a large batch when parallel resources are available, so as to improve the performance of the resulting surrogate model. The process of launching the parallel cases and collecting the results is handled by the SCAM launcher and collector.

3. Surrogate model training and sensitivity analysis (shown as Part II in Figure 1): Based on the sampling results from the Morris or the Sobol sequence, we integrate existing methods, such as MOAT, Sobol, Delta, HDMR, and RBD to achieve their individual evaluations of each single parameter's sensitivity, as well as a ensemble result of these methods. We also use the sampling results of the Sobol sequence, which includes more information than the Morris sequence, to train a neural network (NN) based surrogate model. With the efficiency to project a result in seconds rather than minutes, we can apply it for evaluation of sensitivities of a combination of multiple parameters.

4. Parameter tuning and validation (shown as Part III in Figure 1): With the NN-based surrogate model to cover expanded search space with less time, we also propose an optimization method that combines grid searches by the surrogate model, which achieves better results with less compute time. The results of parameter tuning are then validated through running of real SCAM models.

5. Case comparison study: At the very end, we perform a comparison on the optimization results between the joint optimization across five cases and the independent optimization of the five cases. Results demonstrate the different and the correlation of different cases, and the potential of performing grid-specific tuning in the future.

## 3 Methodology

### 3.1 Sampling of SCAM

To improve the model performance by tuning parameters, we must first identify the parameters that have significant impacts on the simulation results, i.e., the parameters that potentially contribute the most uncertainty of the model. Sensitivity analysis (SA), which analyzes and measures each factor's influence on the output of the model, is an important solution to the above question of finding the more sensitive parameters. A parameter is said to be sensitive if it can lead to a greater change in the output compared to other parameters under the same magnitude of change.

The first step of SA is sampling, i.e. generating a sequence of changing inputs and parameters to observe the corresponding change in the output. The different mathematical approach that we take to perform sampling would certainly affect the features that can be captured from the system. For example, the Morris sampling (Morris, 1991), which takes the approach of



changing one variable at a time, shows an advantage in compute efficiency, but lacks the capability to describe the interaction between different parameters. In contrast, Sobol' quasi-random sequences involves an increased number of samples to cover a similar space, when compared with Morris, but has the advantage of considering second-order sensitivities between different parameters (Campolongo et al., 2007).

We integrate both Morris and Sobol for the sampling step in our tuning workflow, as both of them are still used in many
climate model related SA studies (Pathak et al., 2022). The Morris sampling drives the MOAT SA module afterwards, while the Sobol sampling drives four different SA modules (Sobol, Delta, HDMR, and RBD-FAST) shown in Table 4.

**Table 4.** SA methods integrated in the workflow and used as a cross-reference to the proposed method.

| Name of method | Abbr. | Reference |
| --- | --- | --- |
| Morris sensitivity analysis | Morris | Morris (1991) |
| Delta moment-independent measure | Delta | Plischke et al. (2013) |
| Sobol' sensitivity analysis | Sobol | Saltelli et al. (2010) |
| High-dimensional model representation | HDMR | Li et al. (2010) |
| Random balance designs fourier amplitude sensitivity test | RBD-FAST | Goffart et al. (2015) |

## 3.2 Training a NN-based surrogate model

The by-product of the sampling stage is that we also have a well-designed set of samples that can be used to train a surrogate model.

Compared with the other surrogate models used in previous studies, such as the generalized linear model (GLM) for SCAM5 (Guo et al., 2015), and the spectral projection (SP) and basis pursuit denoising (BPD) for SCAM6 (Pathak et al., 2022), we propose to adopt a neural network (NN) based model, which takes the advantage of neural network to capture nonlinear behaviors of an unknown function, and to describe the behavior of the corresponding physics part in a more accurate way.

To keep an acceptable level of accuracy of surrogate models, we choose to train respective models for each different SCAM
case (the underlying assumption is that the model should learn the different patterns in different SCAM case locations). For our specific cases to study the precipitation performance of SCAM, we use a neural network with 1 input layer, 2 hidden layers, and 1 output layer. Figure 3 shows the detailed structure of the neural network.

At the training stage, we reuse the 768 sets of different parameters and their corresponding total precipitation output values as the training sets and validation. In this case, the training and validation sets are split in a ratio of 8:2. We set the learning rate
to be 0.001, and the batch size for each training was 32.

As can be seen from the training loss accuracy in Figure 4, the network obtained from the training has a good fit for the Sobol samples in each case. The accuracy rate exceeds 90% in all the five cases. As the Sobol sequence of samples have



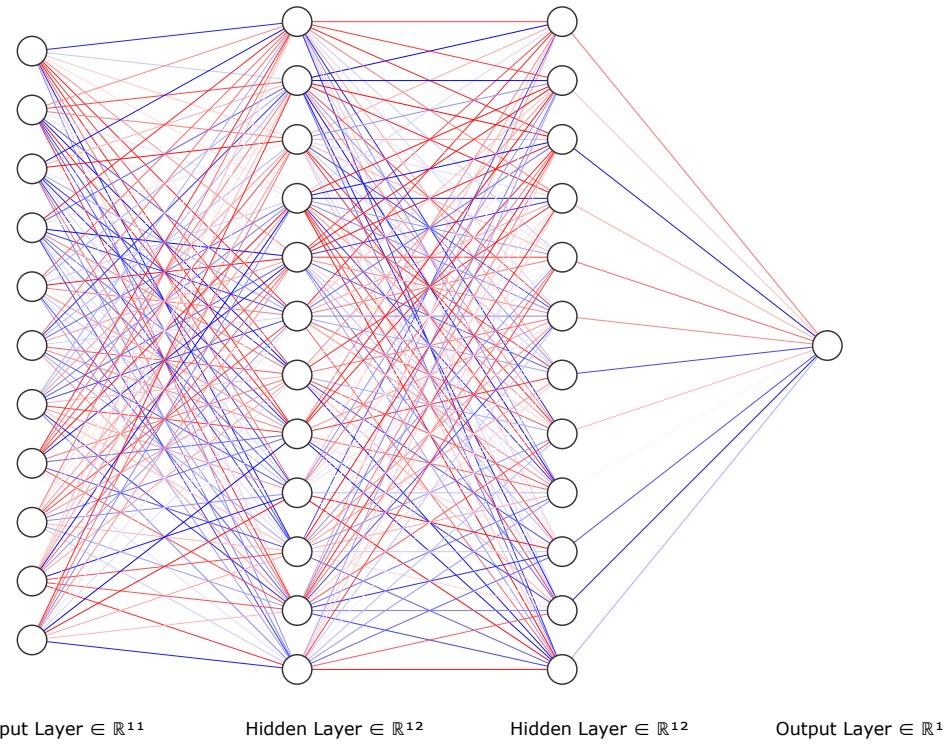

Input Layer $\in \mathbb{R}^{11}$     Hidden Layer $\in \mathbb{R}^{12}$     Hidden Layer $\in \mathbb{R}^{12}$     Output Layer $\in \mathbb{R}^{1}$

**Figure 3.** Neural network structure of an surrogate model of SCAM, targeting for providing the total precipitation output. There are 11 neurons in the input layer, corresponding to the input parameters of the model; 2 hidden layers, each with 12 neurons; and 1 neuron in the output layer representing the PRECT (i.e. total precipitation) of the surrogate model simulation output.

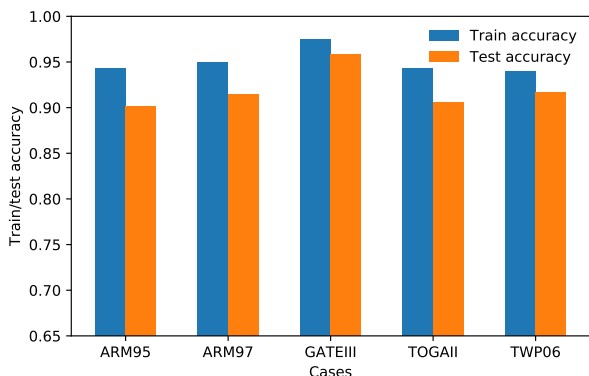

**Figure 4.** The accuracy of the resulting NN-based training model on the training and test sets of each case.

a good representation of the entire parameter space, we expect the model to perform generally well in different parameter combinations.





### 3.3 Sensitivity analysis for a single parameter and combinations of parameters (enabled by the NN-based surrogate model)

The SA methods, similar to the climate model itself, have their corresponding uncertainties. The SA methods provide a best estimation of each parameter's sensitivity, not a direct evaluation. Therefore, each different method might its advantages and disadvantages in different ranges of the parameter values.

As a result, in our workflow shown in Figure 1, we choose to integrate multiple SA methods, including the ones that can be built on the Sobol sequence, such as Delta, HDMR, and RBD-FAST, and the Morrison method, which is still used often for climate models, due to its efficiency advantage. The integration of multiple methods enables us to evaluate the uncertainty of different SA methods. As the sensitivity values calculated by the different methods are of different orders of magnitude, these sensitivities have been normalised for comparison purposes.

The parameters may interact with each other, and the effects of multiple parameters on the simulation results may be superimposed. Therefore, tuning multiple parameters generally have a more significant effect than tuning a single parameter. With the help of commonly used sensitivity analysis methods mentioned above, it is easy to obtain the sensitivity of individual parameters. However, in cases where we need to identify and tune a set of parameters in a combined way, both the SA and the tuning task would involve a significantly improved level of compute resources.

Here, we take an example of analyzing a combination of $M$ different parameters, and adopt a grid-based sampling approach to explore the sensitivity. Assuming that we divide the possible value ranges of each parameter into L levels, to cover a complete grid with possible changes of all $M$ parameters, we need to explore $L^M$ different combinations. Thus, in the sampling stage, we would need to apply the above $L^M$ parameter values to SCAM to obtain the same number of simulation results, and find the result with the largest difference from the outputs from default value and its corresponding parameter value combination. In this scenario, letting N be the number of processes required to carry out a set of tests, we have:

$$N_{MPP} = C \times \binom{M}{p} \times L^p \tag{1}$$

where $C$ represents the number of cases to be tuned by user, $M$ represents the total number of parameters used for study, $p$ represents the number of parameters we perturb in each test, and $L$ represents the number of levels we cover within the value range of each parameter.

When we do a combined study to identify and tune a most sensitive set of three different parameters, the number of tasks to run is already at the level of multiple thousand. If we expand the case to four or five parameters, the number of tasks would grow rapidly to tens of millions of runs. Even for the SCAM model, which takes more than one hour to finish a run, such a combined cost becomes impractical, for combined studies of multiple parameters.

An obvious benefit of using a surrogate model for training is that it is very fast to compute. Since the model has been trained, the time taken to output the surrogate results is negligible. Therefore, we can try as many combinations of parameters as possible without being limited by computational resources. By applying the trained surrogate model, we tested the maximum fluctuation in the output of each case when perturbing one to four parameters, as shown in Figure 5. As can be seen from the figure, as the number of parameters to be tuned simultaneously increases, the fluctuations that can be brought about are also



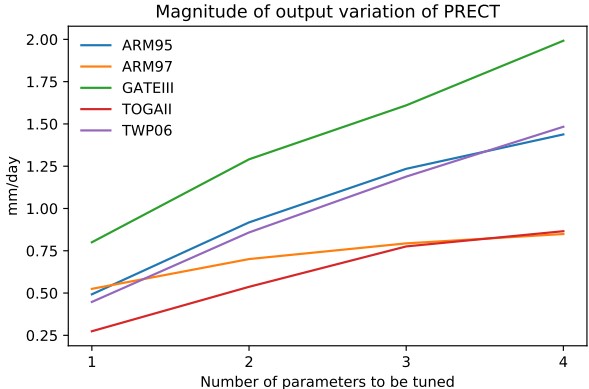

**Figure 5.** The maximum fluctuation in the output of each case when perturbing one to four parameters.

greater. However, the complexity of the calculation increases exponentially during the test, as shown in Table 4. Therefore,
although the effect of tuning four parameters was better than tuning three parameters, the advantages of the surrogate model
in the parameter tuning process could not be exploited at this point. Therefore, after considering the tuning effect and the
computational overhead, we decided to use the experiment of tuning three parameters as a demonstration. At the same time,
the three-dimensional parameter space is easier to represent visually later.

Here we look at which combinations of three parameters lead to the most significance in output while taking into account the
computational overhead (using PRECT as an example). The utilization of the NN-based surrogate model, on the other hand,
provides a more feasible solution to this problem. With the compute time reduced to less than five percent of the original model,
and an error of only less than three percent in most cases, sampling tens of thousands of tasks becomes a practical option.

### 3.4 Parameter tuning enhanced with NN-based surrogate model and grid search

After identifying the most sensitive set of parameters, we then perform parameter tuning to optimize the model for a certain
output. In this paper, we focus on SA and tuning towards a better precipitation result.

A natural idea is to also bring the NN-based surrogate model enhancement to the tuning part. A first thought is to replace the
run of the actual SCAM model with the NN-based surrogate where possible. A second thought, which is even more meaningful
for improving both the efficiency and optimization performance, is to integrate a surrogate-based grid search with customized
range and grid size, so as to avoid the falling into local optima and to accelerate the convergence of the tuning task.
The specific implementation of our tuning scheme (integrating the existing optimization algorithms with the NN-based
surrogate) is as follows.

1. Determine the overall parameter space range for conducting the grid search based on the initial setup of the experiments.





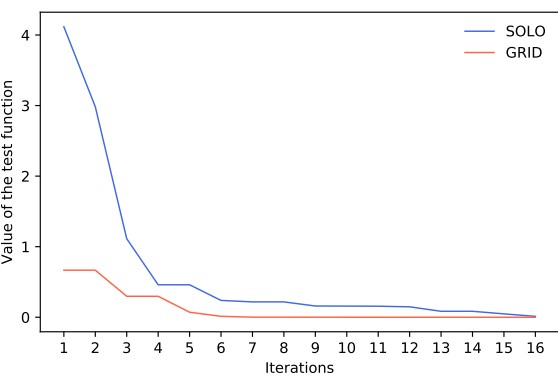

**Figure 6.** Take the sphere test function as an example of the optimization process after combining it with grid search. SOLO means using the optimization algorithm alone, GRID means combined with grid search.

2. The simulation output values corresponding to these grids are calculated in the surrogate model. Then the best performing points are selected and the clustering results of these points are identified as the parameter space for the new finer-grained

search.

3. The optimization algorithm is carried out in the newly defined space and depending on the available computational resources it is decided whether SCAM is invoked to compute in each iteration round. However, the final results must be substituted into SCAM for verification.

4. If the same optimization result is obtained in three consecutive iterations, or less than a certain threshold of improvement

compared to the last, the grid search operation of the first two steps is performed again, thus further reducing the search space, as can be seen from Algorithm 1. In this step, a new parameter space will be constructed with the position of the current best point at the centre and the distance of each parameter from the last best point as the radius.

5. After narrowing down the range, if the result remains the same, the optimization process can be considered to be over and SCAM can be run again using the derived parameters to verify the optimization results.

In Algorithm 1, $Y_i$ is the value of the function in the $i$th iteration, $x_{j,i}$ is the coordinates of $x_j$ in the $i$th iteration, $X_j$ denotes the maximum and minimum values initially set for $x_j$, and $X_j^*$ denotes the maximum and minimum values of $x_j$ in the upcoming grid search. Note that if one of these coordinates is outside the initial parameter space, the original parameter space boundary will be used as the new boundary.

    In order to verify the effectiveness of the method, we first choose Sphere function (Karaboga and Basturk, 2008) as an

example to carry out an optimization test. The trend of the meaning average error during the test is shown in the Figure 6. It can be seen that the introduction of grid search not only helps us to understand the distribution pattern of the parameters, but





---

**Algorithm 1** Procedure for calling grid search.

---

**if** $Y_i - Y_{i-1} < \epsilon$ **and** $Y_{i-1} - Y_{i-2} < \epsilon$ **then**

    **for** $j = 1$ **to** $j = p$ **do**

        **if** $x_{j,i} - |x_{j,i} - x_{j,i-3}| > X_{j,min}$ **then**

            $X_{j,min}^{*} = x_{j,i} - |x_{j,i} - x_{j,i-3}|$

        **else**

            $X_{j,min}^{*} = X_{j,min}$

        **end if**

        **if** $x_{j,i} + |x_{j,i} - x_{j,i-3}| < X_{j,max}$ **then**

            $X_{j,max}^{*} = x_{j,i} + |x_{j,i} - x_{j,i-3}|$

        **else**

            $X_{j,max}^{*} = X_{j,max}$

        **end if**

    **end for**

**end if**

Grid seach.

**return** $x_{j,i+1}$ **and** $Y_{i+1}$

---

also helps to improve the performance of optimization process. The optimization process can converge earlier, while obtaining better tuning results.

### 3.5 Case correlation analysis based on clustering

Following the end of the tuning process, we further perform a comparison study among different cases, to derive valuable insights that would potentially lead to a physics module design that can accommodate the different features in different locations.

In our proposed method, the Pearson correlation coefficient method is used to measure the similarity between two cases (Schober et al., 2018). This coefficient is defined as the quotient of the covariance and standard deviation between two variables. By analyzing the similarity between the 'optimal' parameters of each case, these cases can be clustered. In this process, we

can explore whether for similar types of case, they may have similar responses to various parameters. This will also help us choose better parameter combinations when analyzing other cases, and even regional and global models, so as to achieve better simulations. The implementation is shown below.

1. Search the sensitive parameter sets for each output variable in each case.

2. For each case, the occurrence times of each parameter in the sensitive parameter combination are accumulated to obtain

a variable of $M$ dimension, where $M$ is the number of parameters.

3. The correlation between vectors of each case can be computed by the Pearson correlation coefficient method.



By the above procedure, we can obtain the similarity of the set of sensitive parameters among the cases. The correlation between different cases can also be analyzed for the case of finding the optimal values in the same parameter space. After each case has found its optimal parameter values, this set of parameter values can also be represented by vectors. The same approach

can be used to analyze the correlation between these vectors.

## 4 Experimental results

### 4.1 Hardware platform

The hardware platform for our research is Sunway TaihuLight supercomputer (Fu et al., 2016), hosted at the National Super-computing Center in Wuxi, which is the world's first supercomputer with a peak performance above 100Pflops (Guo et al.,

2019). In addition, compute nodes equipped with NVIDIA GeForce RTX 2080Ti are used to train the surrogate models.

### 4.2 Sampling of the SCAM cases

As mentioned earlier, the sampling scheme, which determines the set of points to represent the entire parameter space, is of essential importance for the following sensitivity analysis and parameter tuning steps. Considering the compatibility between the sampling methods and the SA methods, our platform includes both the Morris (driving the MOAT SA method) and the

Sobol sampling scheme (driving the Sobol, Delta, HDMR, and RBD-FAST SA methods). We use a total of 7,680 samples, with 1,536 samples for each of the five SCAM case. Of these, half (i.e. 768 samples) are used for MOAT, while the remaining half are used for the Sobol sequence. In this part of the experiment, we use the job parallelism mechanism mentioned earlier in the text to execute these sample cases.

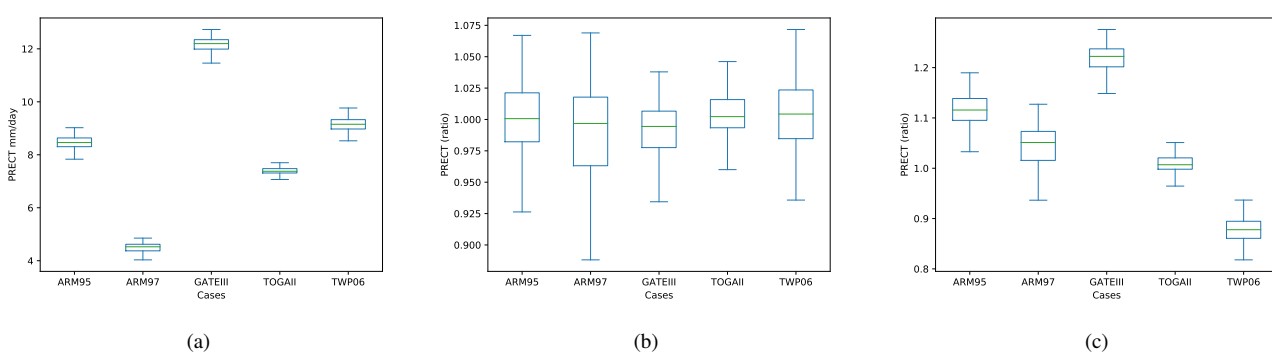

**Figure 7.** Fluctuations in output variation for each case when the parameters are tuned. The output variable is PRECT. In the figure, (a) indicates the range of variation in absolute values in mm per day; (b) indicates the range of proportional variation relative to the control trial; (c) indicates the range of proportional variation relative to the observed values.





In our sampling, SA, and tuning study, we focus on the total precipitation output (PRECT). Figure 7 shows the range of the

PRECT value with the change of the different parameters. We demonstrate the change range of the absolute PRECT value, as well as the relative change range to the default simulation result and the observation value.

The five different cases (representing field measurements in different locations as well as different time periods) demonstrate a similar change range from 1 to 2 mm per day, although the medium point varies from 5 to 12 mm per day, demonstrating clearly different climate patterns. If we look the changes in terms of a relative ratio to the default simulation result, ARM97

shows the largest change range in the sampling space, while TOGAII shows the least.

Another point worth mentioning is the change range in terms of a relative ratio to the observed values. In the cass of ARM97 and TOGAII, the change ranges include the observation values, which means we are likely to converge to the observation point during the parameter tuning process. In contrast, the change ranges of the other three cases (ARM95, GATEII, TWP06) do not include the observation values, which suggests a more challenging tuning task.

**4.3   Single-parameter sensitivity analysis across different cases**

After sampling, five SA methods listed in Table 4 are used to compute the sensitivity of these parameters to the PRECT output. Heat maps are used to characterize the sensitivities of each parameter. As can be seen from Figure 8, there are differences in the results obtained from different SA methods.

When we compare across different SCAM cases, the impacts of different parameters on the PRECT output variable are

different (with a difference of over one order of magnitude). The parameters that are designated for certain regions naturally behave differently in different cases. For example, pz1 (c0_lnd, deep convection precipitation over land) is apparently not a sensitive parameter for the cases located in the tropical ocean, and pz2 (c0_ocn, deep convection precipitation over ocean) would not have a significant influence on the land cases.

The parameters of pu3 (rkm, updraft lateral mixing efficiency) and pc2 (rhminl, threshold relative humidity for stratiform low

clouds) has a greater impact on the tropical convection cases, particularly TOGAII and TWP06, than on the land convection calculations. The sensitivity of pz4(tau, time scale for consumption rate deep CAPE) is significant across almost all cases except ARM97. For the case of ARM97, the only pz1 (c0_lnd, deep convection precipitation over land) is picked as a relatively sensitive parameter, which demonstrates a quite different pattern from ARM95. The reason for this difference probably comes from a different time of year and the forcing field simulated in these two cases.

We also perform a comparison of sensitivity analysis results between different SA methods. To demonstrate the applicability of using our learning-based surrogate model in the SA process, we also show the SA results by using our trained NN-based surrogate models. Instead of calling SA methods directly, we use combinatorial analysis of the magnitude of change to determine the effect that these parameter combinations have on the model output.

For comparison across different methods, the signals for the top sensitive parameters are more or less similar among different

methods. However, we can also observe apparent differences between methods. The MOAT method, which takes the Morris sampling sequence, deviates more from the other methods that rely on the Sobol sequence. Among the Sobol sequence family,

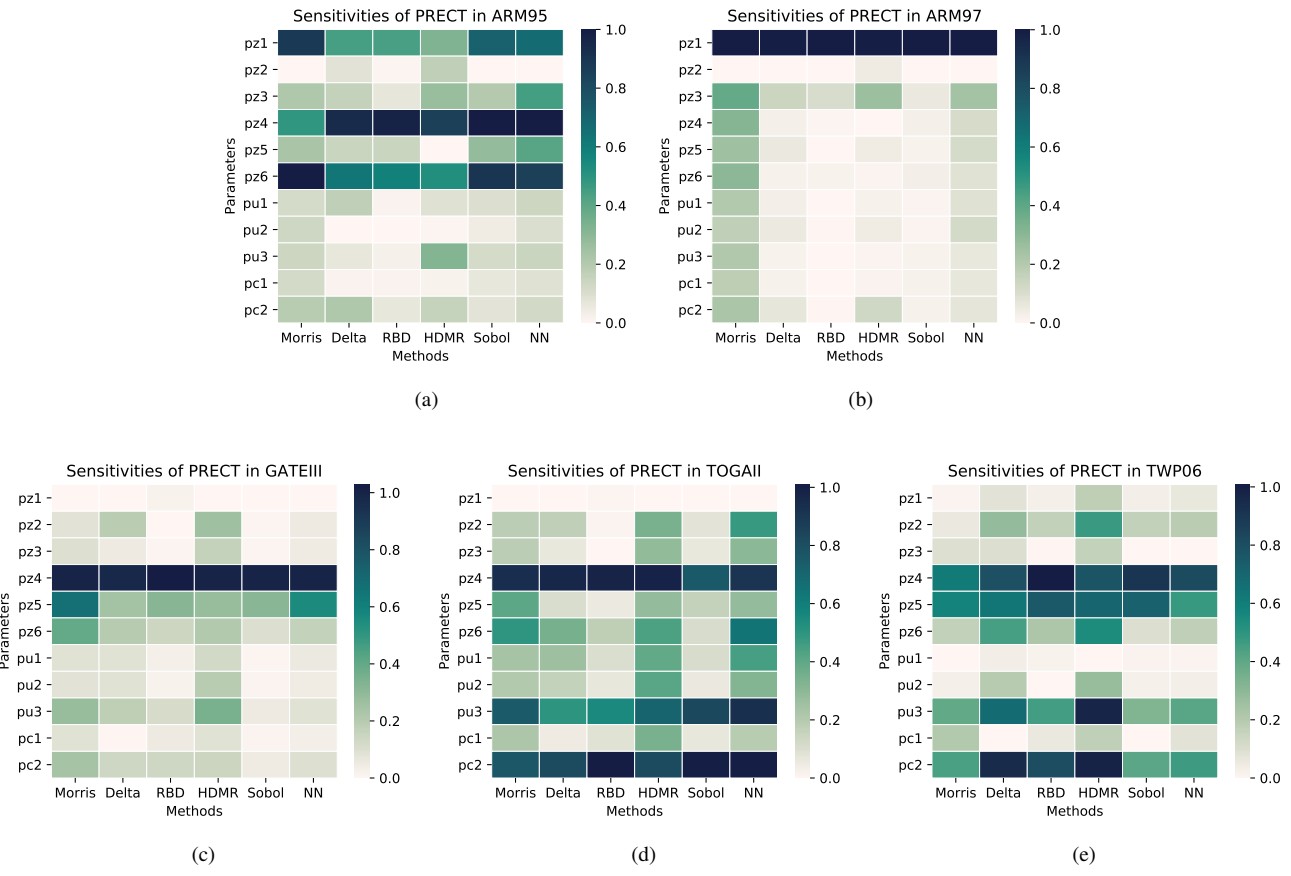

**Figure 8.** The comparison of the sensitivity of each parameter to PRECT comes from different analysis methods (including NN) in five cases. Sensitivity results were normalised. (a)ARM95, (b)ARM97, (c)GATEIII, (d)TOGAII, (e)TWP06

Delta, RBD and HDMR share more similarities, while our learning-based method and Sobol share more similarities. From the performance on these used, it is possible to achieve a better simulation of the effects of parameter changes.

Another interesting pattern to investigate is the direction of the sensitivity. Figure 9 shows the change of PRECT for different pz4 (tau, time scale for consumption rate deep CAPE) values in different cases. Even for the same parameter and the same parameter space, the trend of their effects on PRECT varies, and even in opposite directions. For example, increasing tau tends to increase total precipitation in GATEIII, while in the other cases it brings the opposite result.

### 4.4 Joint multi parameter sensitivity analysis using learning-based models

As mentioned earlier, by using the learning-based surrogate models, we now have the capability to explore the sensitivity of multi-parameter combinations. As our study investigates 11 parameters related to PRECT, there are a total of 165 three-parameter combinations. It would be difficult to test all these combinations using the original SCAM model, due to the high



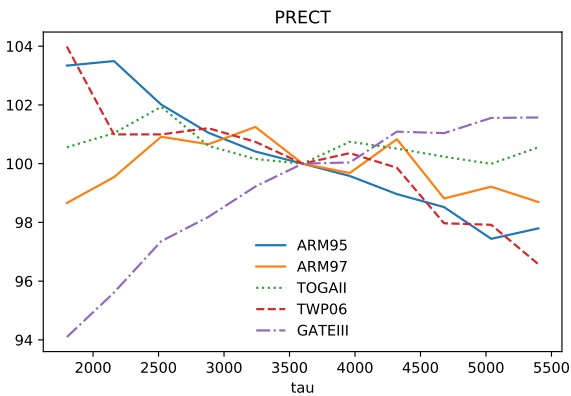

**Figure 9.** Magnitude of output variation of the parameter tau to precipitation related output indicators in five cases. The solid line shows the land convection case and the dashed line shows the tropical convection case.

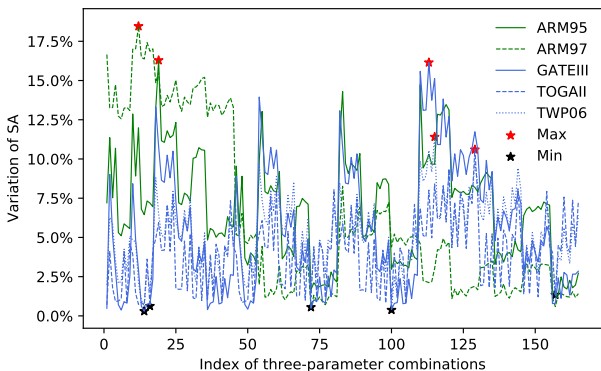

**Figure 10.** The variation of SA among different three-parameter combinations in five different cases. The same index indicates the same combination of parameters.





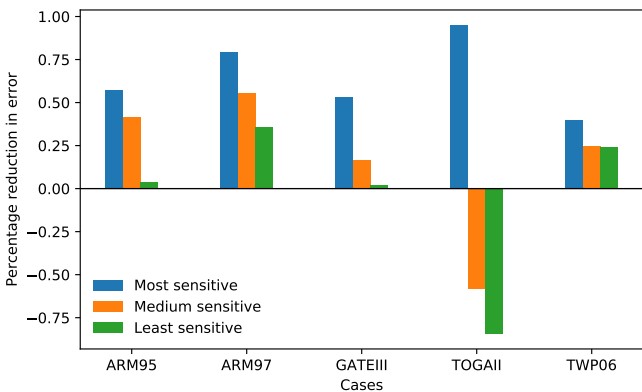

**Figure 11.** Comparison of different combinations of parameters under the same optimization algorithm after SA filtering and sorting. Three representative parameter combinations are selected for each case.

computational overhead. However, with the help of the surrogate model, we can instead accomplish these tests in less than a minute, with a over 90% confidence of the sensitivity analysis results.

For each case, the magnitude of the change in output has been evaluated for all possible combinations of the three parameters
in turn, with the help of the surrogate model trained. This is shown in Figure 10. It can be seen from this that there are significant differences in the magnitude of output variation that can be brought about by different combinations of parameters. The maximum variation relative to the default test is up to 18.6%. The comparison with the previous tests also fully illustrates that the combined tuning effect of the three parameters is more significant.

In order to show the impact of different parameter combinations on the optimization results, the first, middle and last
ranked parameter combinations were selected and combined with the same algorithm to carry out the optimization. In the scenario where the three parameters are tuned, there are 165 combinations of parameters that can be tuned, so the combination ranked 83rd was chosen as the one in the middle of the ranking. These combinations are listed in Table 5. Whale optimization algorithm(WOA, Mirjalili and Lewis (2016)) has been chosen as the method in this stage. Set the number of whales in the optimization algorithm to 32 and the maximum number of loop iterations to 16.

MAE was selected as the metric of optimization effect. As can be seen from Figure 11, the higher ranked parameter combinations do have smaller errors for the same optimization conditions. Especially for the case TOGAII, an inappropriate combination of parameters can even lead to a deterioration of the output results, even when an optimisation algorithm is applied. The impact of such differences is even multiplied, which is therefore a good example of the need to choose the right combination of parameters as the object to be optimized. Next, we will explore further the sensitive parameter combinations.





**Table 5.** The most, middle and last sensitive parameter combinations selected for comparison to evaluate the differences.

| Rank | ARM95 | ARM97 | GATEIII | TOGAII | TWP06 |
|------|-------|-------|---------|--------|-------|
| Most | pz1, pz4, pz6 | pz1, pz3, pz6 | pz4, pz5, pu3 | pz4, pu3, pc2 | pz4, pc2, pz5 |
| Middle | pz1, pz2, pc2 | pz1, pz2, pu2 | pz6, pu2, pc2 | pz2, pu2, pc1 | pu1, pc1, pc2 |
| Last | pu2, pu3, pc2 | pu1, pu2, pc1 | pu1, pu2, pc1 | pz2, pz4, pu3 | pz3, pu1, pu2 |

**4.5 Joint optimization for SCAM cases combined with grid search and learning-based models**

To validate a more efficient search method, we applied typical optimization algorithms and a grid search that combines them to find the optimum. The parameter space used for optimization will be reduced to the range identified in the grid search process above and more fine-grained optimization will be carried out.

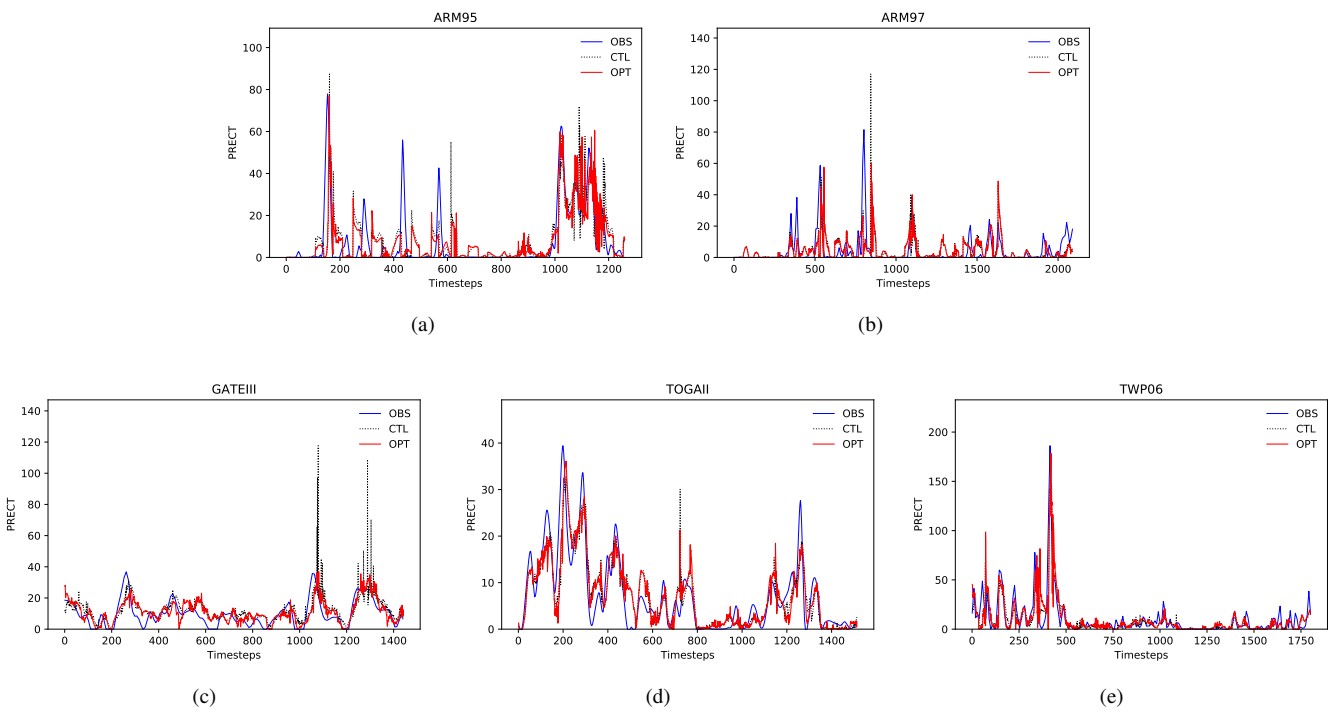

**Figure 12.** Model simulation output before and after tuning versus observed values. Where OBS indicates the observed value, CTL indicates the output before tuning and OPT indicates the output after tuning. (a)ARM95, (b)ARM97, (c)GATEIII, (d)TOGAII, (e)TWP06.

Consequently, experiments combining the grid search and optimization algorithms will be performed. As above, WOA is still
used as the optimization algorithm for this stage. In the above experiments, we performed three different groups of experiments

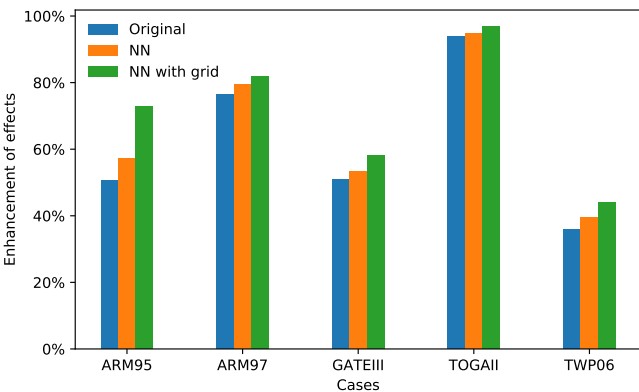

**Figure 13.** The improvement in optimization achieved by using NN-based combinational sensitivity analysis, and further combined with grid search, compared to control experiments with default parameter values.

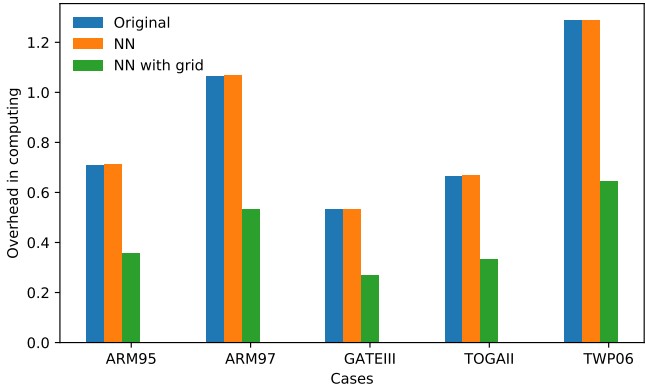

**Figure 14.** A comparison of the total computational overhead for the various strategies used. The meaning of the legend is the same as in the previous figure.

for the joint parameter tuning of multiple operators in SCAM. They also correspond to different principles for the selection of sensitive parameters.

To verify the effectiveness of the parameter tuning, the output after tuning was compared with the output of the control experiment (i.e. before tuning) and the observed data after the individual cases had been tuned, as shown in Figure 12. Here, the experiments with the best optimization results are chosen for comparison. The tuning of the SCAM parameters was quite productive on the time scale. It is easy to see that in the control experiment there were several spikes where the simulated output was significantly higher than the observed values, as was the case in the first four cases. After tuning, these spikes are significantly weakened and the output is much closer to the observed values. This demonstrates the significance of the parameter tuning provided by the workflow for model.





A cross-sectional comparison of the effects of these sets of experiments is shown in Figure 13. Meanwhile, the computational overheads of the various strategies are compared, as shown in Figure 14. In contrast to using the optimization algorithm alone, the grid search combined with the optimization algorithm can achieve better results on these SCAM cases. The use of NN trained surrogate models for parameter tuning can further save computational resource overhead and, in terms of results, can meet or exceed traditional optimization methods in most cases.

Therefore, we can find that it is possible to achieve a win-win situation in terms of computational resources and computational efficiency by training a surrogate model of SCAM based on NN. The model can get a boost in performance from 5%-15% in precipitation output. Thus, using the proposed method, the main computational overhead comes from sampling and training. The computational overhead can be saved by more than 50% compared to the case where the above experiments were all run using the full SCAM. In particular, the proposed method demonstrates its effectiveness and usability for situations

such as large-scale grid testing, which is almost impossible to accomplish using the full SCAM.

    These results also show that the method used in the workflow outperforms previous methods in most cases. Furthermore, the methods in the workflow test a wide range of combinations of values in the parameter space. Thus using the workflow provides a more complete picture of the parametric characteristics of the different cases in SCAM than optimization algorithms that only provide results but no more information about the spatial distribution of the parameters. Doing a finer-grained grid search

in the vicinity of the optimal value point is also an approach that is worth testing in the future. From the experimental results, it can be seen that utilizing the surrogate model through the sampling with the optimization algorithm not only saves resources, but also improves the optimization effect, and meanwhile improves the robustness of the optimization method.

### 4.6    A deeper exploration of the relationships between the cases

    The training of surrogate models makes it possible to conduct larger scale experiments in a shorter period of time. For the

most sensitive combinations of parameters in each of the cases obtained based on the NN method, we are able to explore the distribution pattern of the results using a grid search. These experiments are carried out on the surrogate model for reasons of improving experimental efficiency. If such experiments were run using the full model, the computational overhead would be very high.

    Grid search can also be performed to determine the possible aggregation range of the better-valued solution, and the results

are shown in Figure 15. The value of each parameter is also divided into 11 levels within its upper and lower bounds. The parameter space remains the same as originally set at the beginning of this paper.

    As can be seen, after the parameter space has been replaced, the better-valued solutions for each case show a clear trend towards aggregation, and although the distribution of TOGAII is slightly scattered, it can still be grouped into a cluster. The same parameter space is more conducive to cross-sectional comparisons. It is easy to see that the two land convection regions

are closer and, accordingly, the three tropical convection aggregation regions are also closer. The initial search results for the two cases GATEIII and TWP06 are almost identical. Again, this may reflect commonalities between the cases.

    From the results, it can be seen that the distribution of the better value points is different for different parameters even for the same parameter space. A typical example is the parameter pz4 (tau). In the ARM95/97 cases, a larger value of tau leads to





a better performance of the model. In the other three cases, smaller values of tau lead to better performance. On the one hand,
the response of the output variables to the input parameters is different in the different cases. On the other hand, it also shows
that there are differences in the distribution of parameters that make the results perform better in different types of cases.

It can be got that the optimal value points for the two land convection cases are close, while the points for the three tropical
convection cases are even closer. In the latter three cases, GATEIII and TWP06 are much closer. In addition, another scenario
was considered: where the parameter configuration is the same among cases. In this case, the workflow can also find the
parameter values that minimize the overall bias among cases. Thus, the closeness between the cases could be analyzed. This is
also confirmed by the experiments in the next subsection.

Now that we have discovered the pattern that the aggregation range of the more optimal solution for each case by applying
a full-space grid search in the same parameter space. From the experimental results, it can be seen that the two cases focusing
on land convection are most similar to each other, and the three cases focusing on tropical convection are also more similar to
each other. Combining the previous results also shows that although the two cases ARM95/97 are located in the same region,
the results of the parameter tuning are not fully identical and the similarity does not reach the perfect value. The difference
between the two cases lies mainly in the time, which therefore reflects that there is also a difference in SCAM's simulation
performance for different times. This can provide a basis for designing regional parameter combinations and values later.

Similarly, the optimal parameter values for each case can be represented by a vector since they are in the same parameter
space. Pearson correlation coefficient method is used to compare the best set of values for each case. The obtained similarity
is shown in Figure 16. From this figure, we can see that the parameter values taken between the two cases of land convection
are positively correlated in the same parameter space. The three cases of tropical convection are also positively correlated with
each other. And cases belong to different types are negatively correlated with each other. These results are also well matched
to those obtained in the previous experiments. It is also clear from the results above that SCAM cases in similar locations and
of the same type are more relevant when it comes to parameterization.

## 5 Conclusions

In this paper, a learning-based integrated method for SCAM parameter tuning on the HPC is proposed which enables a fully
automated diagnostic analysis process from sensitive tests to parameter optimization and case comparison. The workflow
makes it possible to run SCAM parameters on a large scale, thus allowing more trials on parametric scenario studies to be
carried out in a shorter period of time. An integration of several sensitivity methods approach is used for sensitivity analysis of
parameters, with just once sampling, the different SA methods can be invoked for analysis and their results combined.

With the enhancement of artificial intelligence and machine learning techniques, the role played by neural networks in
regression analysis has become increasingly evident. In our proposed experimental workflow, NNs incorporating sampling
techniques are likewise used in the parametric analysis process of SCAM. A precursor to sensitivity analysis: the results of
the sampling are used to train an NN-based surrogate model, which validates the accuracy of both the sensitivity analysis and
improves the parameter tuning process by the surrogate model. In these stages, a grid search strategy for parameter space based



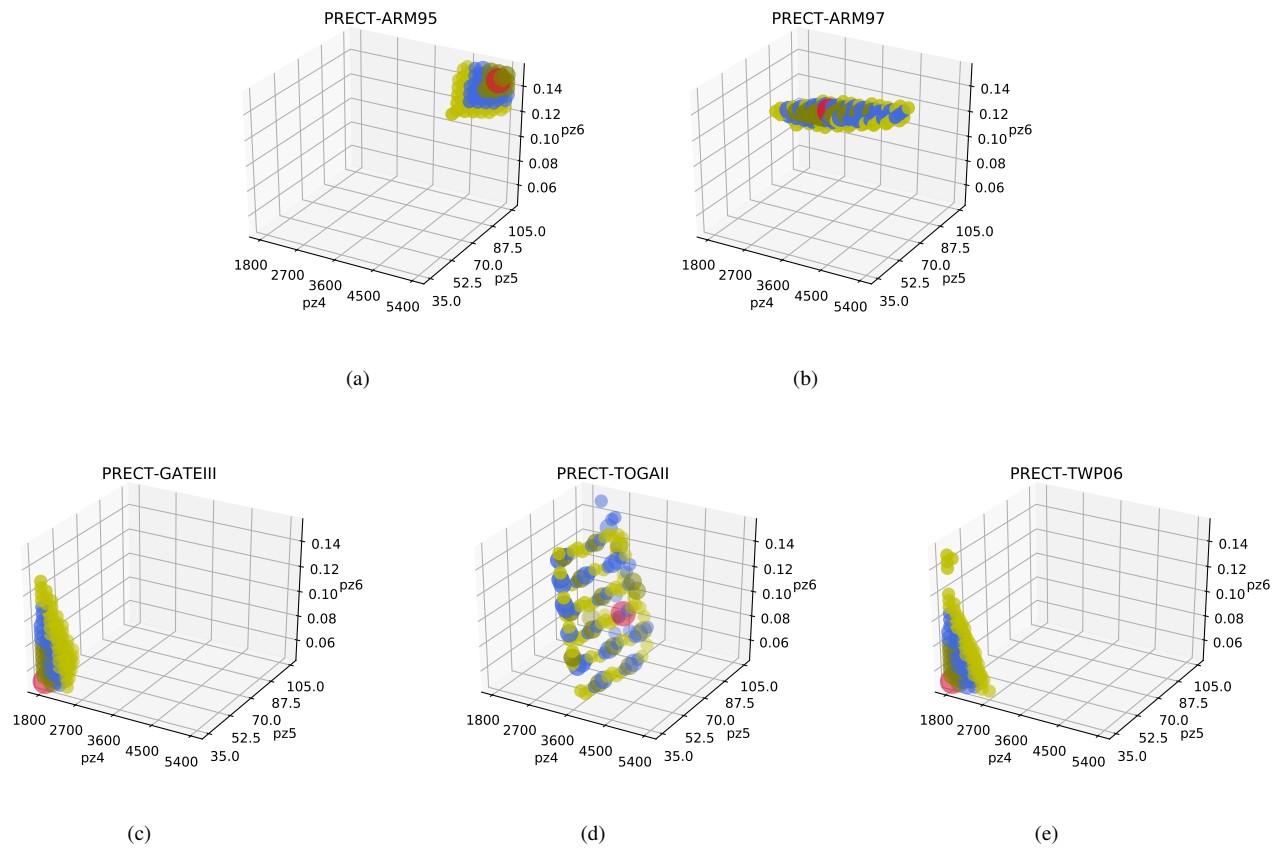

**Figure 15.** The distribution of points that perform better when the same combination of parameters is tuned jointly for all cases. All cases use the respective best performing parameter values. (a)ARM95, (b)ARM97, (c)GATEIII, (d)TOGAII, (e)TWP06. The points closest to the observed data are shown in red, those ranked 2-32 are shown in olive, and those ranked 33-128 are shown in blue.

on multi-parameter perturbations is used. With the computational capabilities of the HPC, the method can search the most suitable parameter values within less iterations. The combination of grid search and optimization algorithms can also improves the performance of the optimization algorithm in model parameter tuning. In addition, the application of neural network-trained surrogate models can also saves computational resources, which is beneficial in achieving the goal of green computing.


To verify the completeness and validity of the proposed workflow, multi-group experiments based on five typical SCAM cases was implemented on the workflow. The sensitivity of the above parameters to typical output variables related to precipitation was analyzed. Experiments based on the proposed workflow have shown that there are differences in the sensitivities of the parameters with respect to the different cases and different output variables. This includes both the differences between the different convection types of cases and the differences in the effects of the deep and shallow convective precipitation param-

eterization schemes on the respective precipitation. To summarise, determining the appropriate values for each of the SCAM



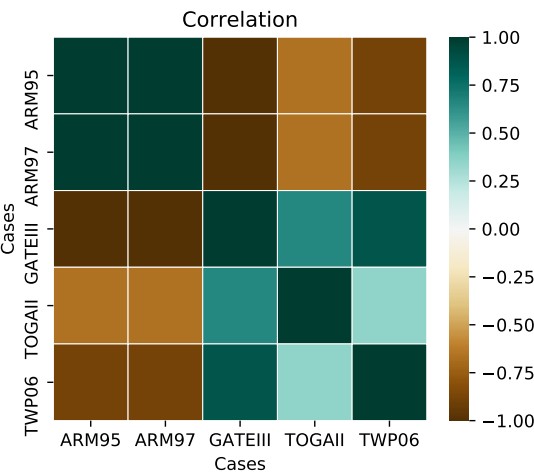

**Figure 16.** Correlation of the optimal solutions of the cases in the same parameter space.

cases located at different locations facilitates is also meaningful for model development. This also provides a heuristic for future research on similar parametric schemes on other models.

*Code and data availability.* Open source code for this article is available at https://doi.org/10.6084/m9.figshare.21407109.v5 (Guo, 2022)
along with the data used. The above is subject to the MIT License Agreement.

*Author contributions.* The manuscript was written by all authors. JG and HF proposed the main idea and method. YX, WX and LW have provided important advice. LH, GX and XC provided assistance and support with the experiments. XW assisted in debugging the program. LG provides help with access to computing resources.

*Competing interests.* The authors declare that they have no conflict of interest.

*Acknowledgements.* This research was supported in part by the National Key Research and Development Plan of China (Grant No. 2020YFB0204800), National Natural Science Foundation of China (Grant No. T2125006, U1839206), Jiangsu Innovation Capacity Building Program (Project No. BM2022028), Project of Jilin Province Development and Reform Commission No. 2019FGWTZC001 and the Science and Technology Development Plan of Jilin Province of China under Grant 20220101115JC.





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
