# Peer review of "A learning-based method for efficient large-scale sensitivity analysis and tuning of single column atmosphere model (SCAM)"

_Geoscientific Model Development, 2022_

## Referee Comment (RC1)

**A learning-based method for efficient large-scale sensitivity analysis and tuning of single column atmosphere model (SCAM)**

In this study, Guo et al. explore the utility of training a neural network-based surrogate model for SCAM (single-column Community Atmosphere Model). The authors conclude that the NN model significantly improves computational efficiency without a significant loss in model performance and is thus particularly useful for parameter tuning, which is explored by studying PRECT biases in five different IOPs. While the NN model is indeed a useful tool for parameter tuning, the study should undergo major revisions prior to publication.

**Major Comments**

The paper relies on the premise that SCAM is time-consuming to run for these IOPs, stating that a single SCAM run can take more than hour to complete. For most user, SCAM should run *much* faster than this and take no more than 5-10 minutes at the most to complete these IOP cases. A NN surrogate model would still be significantly faster, but the benefit of it relative to hour-long SCAM cases is likely significantly overstated here. It would be nice to see more clear motivation/problems outlined in the introduction.

- Lines 217-218: "Even for the SCAM model, which takes more than one hour to finish a run, such a combined cost becomes impractical, for combined studies of multiple parameters.…" - How many nodes are being used for your SCAM baseline case? Properly tuned, SCAM should run in a matter of minutes and should certainly not take more than an hour to run any of these IOP cases.
- Line 231: "With the compute time reduced to less than five percent of the original model" – is this based on a SCAM case taking >1hr to run for an IOP? I would question that baseline performance.
- Line 72-73: "Improved balance between cost and accuracy" – The accuracy of SCAM is not in question, so I don't see that the balance between cost and accuracy has significantly changed. The cost itself is also not a prohibiting factor in running SCAM, as it's already pretty efficient. The issue with assessing a larger number of parameters often lies in the tractability of analysis, not computational cost.

More generally, the paper should undergo some restructuring and refining. There are frequent references in the introduction to points that are not made until the methods (i.e., the number of samples, error improvements in variables that are not specified, etc.). Ideally, there should be a more distinct separation between the introduction (problem statement and relevant background) and the methods (specific details of the approach used and the experiments conducted here).

- Lines 164-168: "For example, the Morris sampling (Morris, 1991)… " – This statement feels like it's nearly identical to the introduction (lines 36-42). Overall, section 3.1 seems to rehash the background given in the introduction rather than describing the specific SCAM cases run.
- Line 183: "At the training stage, we reuse the 768 sets of different parameters and their corresponding total precipitation output …" - What are the 768 different sets of

parameters? How does this arise from the 11 parameters tested? Is it 768 sets per IOP? More detail is needed on the SCAM runs in order to provide context for this section.

- Section 4.1: I'm not sure how relevant this is to the results shown below; ideally, regardless of where SCAM is run, the results should be identical. Would suggest removing this full section on use of Sunway TaihuLight, particularly as it's mentioned already previously.
- Section 4.6: It's unclear to me how this differs from section 4.5; please elaborate further in the main text. Did section 4.5 not use the same range of values for all cases?
- Lines 410-411: "This is also confirmed by the experiments in the next subsection." should there be a section 4.7 here that's being referenced?
- Lines 408-409: "In addition, another scenario was considered: where the parameter configuration is the same among cases": This paragraph seemed to suggest that another experiment was conducted in which the parameters are all set to the same value in order to optimize performance across all five cases; I don't see results from that experiment though.
- Sections 4.5-4.6: I don't see an explicit discussion of cases where the optimal parameter values are carried out in SCAM rather than the NN surrogate model to confirm the results. The authors should clarify which experiments are conducted in SCAM vs. the surrogate, and consider a more explicit discussion of differences that arise when the optimal tuned parameters are used in full, online SCAM runs.

Please also ensure that figures are clearly described both in the text and the captions, and have units consistently on the y-axis, if relevant. Some of the plots also have curious signals that could be better elaborated on in the text.

- Figure 5: Could you define 'maximum fluctuation' more precisely? Is this the difference between lowest and highest PRECT value, and is the value of PRECT output hourly/daily/etc?
- Line 224: "However, the complexity of the calculation increases exponentially during the test, as shown in Table 4." I'm not sure Table 4 is the right reference here; it shows the SA methods used, but does not seem to indicate complexity or cost of these tests.
- Figure 9: Which SA method is being used in this figure? Units on the y-axis would also be helpful.
- Figure 9: TOGAII seems like PRECT is less sensitive to tau than the ARM97 case, which is at odds with the heat map in Figure 8. Is there a reason for the apparent discrepancy?
- Figure 12: Please add units to the y-axis
- Figure 13: A more detailed description of this figure in the text would be helpful. It's unclear what improvement is being plotted, and what the 'original' case in blue is given that the 'enhancement of effects' on the y-axis is due to the addition of NN and/or grid searching. Is the blue not the control case then?
- Figure 14: Similarly, more detail would be useful. What units are the overhead in computing given in? Is it obvious that the overhead should be the same for the Original and NN cases?

- Lines 421-422: "we can see that the parameter values taken between the two cases of land convection are positively correlated in the same parameter space" it's surprising to me that the correlation is equal to 1 between ARM95 and ARM97 despite apparent differences between them in Fig 15. Could the authors elaborate on why this occurs?

**Specific Comments**
- Lines 1-2: "The Single Column Atmospheric Model (SCAM) is an essential tool for analyzing and improving the physics schemes of CAM." Please specify that CAM in this case is the Community Atmosphere Model
- Lines 6-8: "By reusing the 3,840 instances with the variation of 11 parameters…" Suggest avoiding using specific numbers like this in the abstract; without an explanation, it is unclear what "the 3,840 instances" are. Either add clarification/context, or remove the specific number of instances.
- Line 15: Should this read "the *effects* of global climate change"?
- Line 18: This citation of CAM is outdated, please point to the scientific articles describing CAM instead. The title of this particular citation in the references points to CAM3 and the link itself is broken.
- Line 18: "Of these components, the Community Atmosphere Model (CAM) (UCAR., 2020), is the one with the most complexity." It's hard to say that more model *complexity* is contained in one model component than another; could the authors clarify/justify what's intended by this statement?
- Line 20: "Participated in continuous numerical integration," - Consider rephrasing for clarity; do the authors mean that in coupled climate simulations, this is a source of uncertainty?
- Line 22: "However, as a general circulation model (GCM), CAM takes a long time and a large amount of resource to run…" Given that ESM is already defined above, the authors should continue to use that notation rather than also defining GCM (unless a distinction is intended, which could be elaborated on).
- Line 24: "good alternative model" – rephrase for clarity. What's meant by 'good' here – cheaper, more efficient, etc?
- Line 25: What is meant by SCAM only needs "one process"?
- Lines 28-29: "Sensitivity analysis (SA) is a method for investigating how uncertainty in the model output is assigned to the different sources of uncertainty in the model input factors, and the participants (Saltelli et al., 2010)." It's not clear what the participants are; please clarify.
- Line 41: "quasi-random sequence by Sobol (Sobol', 1967) and other researchers)," please specify the other researchers who have established the method so that it can be easily referenced by readers. It may also be useful to briefly explain what the "low-discrepancy quasi-random sequence" is if relevant to the study.
- Line 48: "After we identify the important tuning targets in the SA stage" – how are those targets usually defined? This hasn't been explained in the previous paragraph, only that there are different methods for sampling parameters. Please briefly note how that translates to identification of targets (even a sentence should suffice).

- Line 70: "By reusing the 3,840 instances with variations of 11 parameters" – there is no indication of where these numbers are coming from or what they refer to. This should be introduced in the methods section, so wait until that point to elaborate on specifics like this.
- Lines 75-82: please condense into a paragraph rather than bulleted list.
- Lines 77-79: "By reusing the 3,840 sampling instances…" - Again, where does the number of instances come from? And when the authors say the model achieves good accuracy, which variables are they referring to (i.e., the "error within 10%" is an error across which variables?)
- Lines 89-91: "case-specific tuned parameters would further reduce the precipitation error by 15% when compared to a set of unified tuned parameters, and suggest a potential improvement from location-wise parameter tuning in the future." I did not see the discussion of a case where all cases are combined to find the optimum parameter values. Please elaborate further on that in the results section to support this.
- Lines 98-100: Is it wise to tune for just a single variable (time-mean PRECT)? Realistically, when assessing model performance and tuning accordingly, a *number* of performance metrics need to be accounted for beyond the mean of one variable. Could the authors elaborate on the validity of selecting just one, perhaps?
- Lines 111-114: "In addition, the programs running on Sunway TaihuLight needed to be recompiled due to the adoption of a different archiecture." - Shouldn't the model be recompiled at build time? Is this a unique addition to the model, that enables compilation with a non-supported compiler? Is the source code available and going to be included in CESM?
- Lines 184-185: "We set the learning rate…" – could the authors elaborate on if this is the most suitable choice of learning rate/batch size? Were other values tested?
- Figure 5: There's a relatively wide variability in this across cases, with ARM97 and TOGAII being the least sensitive and GATEIII being very sensitive to the number of parameters to be tuned. Worth elaborating on?
- Lines 225-226: "…although the effect of tuning four parameters was better than tuning three parameters, the advantages of the surrogate model in the parameter tuning process could not be exploited at this point." I'm unclear why "the advantages of the surrogate model in the parameter tuning process could not be exploited" when using 4 rather than 3 parameters.
- Lines 229-230: "combinations of three parameters lead to the most significance in output" – is this meant to imply the three parameters that drive the most significant *change* in PRECT, or the most significant improvement (assuming those are different, they could be the same, but a big change does not necessarily lead to improvement).
- Lines 255-257: For complete reference, please also explain epsilon and p as they are used in Algorithm 1.
- Line 260: "Meaning average error" – is this meant to be the mean absolute error? Or is this something else?
- Lines 290-292: "We use a total of 7,680 samples, with 1,536 samples for each of the five SCAM case." - Please elaborate on the reason for choosing this number of samples –

how many values per parameter are enabled by this choice? Is there a clear reason for running 1,536 samples per IOP? This also seems like a detail that should be included in the methods rather than the results

- Lines 298-299: "…although the medium point varies from 5 to 12 mm per day, demonstrating clearly different climate patterns." - How much of this is due to differences in the length of the IOP (perhaps one captured more dry days than another, for example), or an IOP designed to capture shallow vs. deep convection? This may not necessarily be indicative of obviously varying climate patterns.

- Lines 318-319: "The reason for this difference probably comes from a different time of year and the forcing field simulated in these two cases." It would be good to see a more confident assertion here. How different is the time of year assessed, is it substantial enough to cause such a change in sensitivity? How different is the forcing field (and does this hypothesize that it's the large scale T or Q convergence that's responsible)? Is there a difference in the type of convection that occurs as well?

- Line 322-323: "Instead of calling SA methods directly, we use combinatorial analysis of the magnitude of change to determine the effect that these parameter combinations have on the model output." I'm not sure on what this means, please rephrase for clarity?

- Lines 330-332: "For example, increasing tau tends
- to increase total precipitation in GATEIII, while in the other cases it brings the opposite result." - Is there something special about that case that causes the unique signal?

- Lines 353-353: "The impact of such differences is even multiplied" – unclear what is meant by this statement; what is being multiplied here?

- Lines 366-368: "It is easy to see that in the control experiment there were several spikes where the simulated output was significantly higher than the observed values, as was the case in the first four cases. After tuning, these spikes are significantly weakened and the output is much closer to the observed values." It looks like this is really only an issue in the land-based ARM cases; is that true? It looks like the tuning is still unable to match some of the largest rain rates in GATEIII especially but also TOGAII – is there a reason for that?

- Lines 368-369: "This demonstrates the significance of the parameter tuning provided by the workflow for model." Could the authors be more quantitative here? How much is the bias reduced by, for example?

- Lines 399-400: "It is easy to see that the two land convection regions are closer and, accordingly, the three tropical convection aggregation regions are also closer." It looks like the land cases might be fairly different, particularly in terms of optimal pz4 value. A table would make it easier to compare the 'optimal' values, even if they're given by a range of what's marked in red in Figure 15.

- Lines 402-403: "the distribution of the better value points is different for different parameters even for the same parameter space." Should this read that the better values are different for different *cases* within the same parameter space?

- Line 404: "In the other three cases, smaller values of tau lead to better performance." It looks like the optimal value of pz4 occurs at/near the minimum range for both GATEIII

and TWP06 – have the authors tested expanding the lower limit of this variable further to see if this is the optimum value or if it's being cut off?

- Lines 405-406: "On the other hand, it also shows that there are differences in the distribution of parameters that make the results perform better in different types of cases." Rephrase? This sounds like it's saying the same thing as the sentence before it.
- Lines 407-408: "It can be got that the optimal value points for the two land convection cases are close, while the points for the three tropical convection cases are even closer." While the GATEIII and TWP06 cases are very similar, the difference in the TOGA case challenges the notion that tropical convection cases are closer than what we see in the ARM case. Over land, it also looks like the pz4 optimal values are actually rather different; a table would make this argument more convincing and easier to see, potentially. Or may point to the need for a more nuanced statement.
- Lines 416-417: "The difference between the two cases lies mainly in the time, which therefore reflects that there is also a difference in SCAM's simulation performance for different times." I don't see a time-based sensitivity analysis in this; could the authors clarify/elaborate? Is this a seasonal difference? Were other alternative hypotheses explored to explain the difference in ARM95 and ARM97 (different synoptic conditions, etc)?
- Line 419: "the optimal parameter values for each case can be represented by a vector" – I'm not entirely clear on what this vector would look like; it might help the reader to plot said vector on Fig 15.

---

## Author Comment (AC1)

**Replies to Referee #1, GMD-2022-264**

Jiaxu Guo on behalf of all authors

April 18, 2023

Thank you very much for your patient and detailed comments on our work[1]. These valuable comments are very helpful for us to improve this paper. After carefully reading all the questions, we have answered each of them and will make appropriate corrections in the revised version of our manuscript.

In this attachment, the red paragraphs represent your comments, and the black paragraphs below are our corresponding replies.

**1   Replies to major comments**

Lines 217-218: "Even for the SCAM model, which takes more than one hour to finish a run, such a combined cost becomes impractical, for combined studies of multiple parameters...." - How many nodes are being used for your SCAM baseline case? Properly tuned, SCAM should run in a matter of minutes and should certainly not take more than an hour to run any of these IOP cases.

We apologize for the unclear statements and the confusion that we may have caused. We totally agree that a single SCAM job would take only a few minutes. The platform we use is based on Sunway Processors. For the 5 cases covered in our article, the shortest one took about 10 minutes and the longest one was no more than 20 minutes. Each Sunway processor consists of 4 Core-Groups (CG). Each CG can support a single MPI process. We normally run one SCAM on one CG (note that the Sunway processor is running at a frequency that is roughly one third of an Intel or AMD processor).

The case we refer to here is a workflow of parameter sensitivity analysis and tuning, which consists of 768 SCAM jobs to run. The one hour mentioned here is the time it takes to assign 768 jobs to the job queue, and to collect the results after all jobs are finished. We do experiments when there are enough resources for multiple times, to compute the time on average. We will add these descriptions in the revised manuscript and try to avoid ambiguities.

Line 231: "With the compute time reduced to less than five percent of the original model" - is this based on a SCAM case taking >1hr to run for an IOP? I would question that baseline performance.

As explained above, we apologize for the one hour confusion we have made. When we say greater than 1 hour, we do not mean that a case run with a single IOP would take 1 hour, but that a run with 768 different cases would take 1 hour. We record the time in a normal supercomputing environment, so as to demonstrate the related time overhead for scheduling, running the job, as well as collecting the results.

Line 72-73: "Improved balance between cost and accuracy" - The accuracy of SCAM is not in question, so I don't see that the balance between cost and accuracy has significantly changed. The cost itself is also not a prohibiting factor in running SCAM, as it's already pretty efficient. The issue with assessing a larger number of parameters often lies in the tractability of analysis, not computational cost.

We agree that our work is primarily about making a large-scale parametric analyses possible, which is what you mean by "tractability". Although SCAM itself has a very short run time, the results of the analysis can be obtained faster by further reducing the resource overhead of the experiment in a large-scale experiment. Again, this is an effort to improve the tractability of the analysis. We will adjust the description of this key issue in the revised manuscript.

Lines 164-168: "For example, the Morris sampling (Morris, 1991)... " - This statement feels like it's nearly identical to the introduction (lines 36-42). Overall, section 3.1 seems to rehash the background given in the introduction rather than describing the specific SCAM cases run.

We will try to avoid redundancy with the background described earlier and elaborate more on the methodology we have used in the revised manuscript.

Line 183: "At the training stage, we reuse the 768 sets of different parameters and their corresponding total precipitation output ..." - What are the 768 different sets ofparameters? How does this arise from the 11 parameters tested? Is it 768 sets per IOP? More detail is needed on the SCAM runs in order to provide context for this section.

This number is based on the number of samples from the MOAT and Saltelli sampling methods. In order to reconcile the two sampling methods used in the text, a number was chosen that is large enough and that matches the relationship between the number of samples generated by both methods. 768 is the number of samples per IOP case. We will add this in a revised version of the manuscript.

Section 4.1: I'm not sure how relevant this is to the results shown below; ideally, regardless of where SCAM is run, the results should be identical. Would suggest removing this full section on use of Sunway TaihuLight, particularly as it's mentioned already previously.

Here we are mainly describing the environment in which the experiments in this paper were run. This platform is really not strongly correlated with the implementation of the experiments, and their results. We will remove this full section on use of Sunway TaihuLight in the revised version of the manuscript.

Section 4.6: It's unclear to me how this differs from section 4.5; please elaborate further in the main text. Did section 4.5 not use the same range of values for all cases?

In Section 4.6, we use the optimal solution of each case as a vector, combined with the Pearson correlation coefficient method to calculate the similarity between the individual cases. By using the coefficient as a metric, it is possible to get a more intuitive view of the relationship between these cases. This part of the vectorization analysis is based on Section 4.5. The range of values taken in the Section 4.5 test is the same for all cases. We will add an experimental exploration of all cases, and cases of the same type using the same parameters, as shown in AC-Figure 1.

Lines 410-411: "This is also confirmed by the experiments in the next subsection.", should there be a section 4.7 here that's being referenced?

We are sorry for the misunderstanding. When a draft of this article was written, there was a section 4.7. It was removed in a later edit, but we did not correct the narrative in time.

"In addition, another scenario was considered: where the parameter configuration is the same among cases": This paragraph seemed to suggest that another experiment was conducted in which the parameters are all set to the same value in order to optimize performance across all five cases; I don't see results from that experiment though.

This is a copy editing error in the collaboration and should read as follows, "After the scenario where the parameter configuration is the same among cases has been considered, the closeness between the cases could be analyzed." We have also added experiments using the same parameter values for all cases. This is shown in AC-Figure 1. From this we can see the distribution of the output in the parameter space for all cases or cases belonging to the same type, when they take the same parameter values.

[Figure]

AC-Figure 1: Results of a three-parameter full-space grid search for the multi-objective scenarios using the surrogate model. The points closest to the observed data are shown in red, those ranked 2-64 are shown in olive, and those ranked 65-256 are shown in blue. Where (a) is the scenario of optimizing two land convection cases with the same set of parameters, (b) is the scenario of optimizing three tropical convection cases with the same set of parameters, (c) is the scenario of optimizing two western tropical Pacific cases with the same set of parameters, and (d) is the scenario of optimizing all five cases with the same set of parameters.

Sections 4.5-4.6: I don't see an explicit discussion of cases where the optimal parameter values are carried out in SCAM rather than the NN surrogate model to confirm the results. The authors should clarify which experiments are conducted in SCAM vs. the surrogate, and consider a more explicit discussion of differences that arise when the optimal tuned parameters are used in full, online SCAM runs.

The results in our paper are derived from experiments using SCAM to confirm optimal parameter values. In fact, this confirmation is already included in the parameter tuning workflow proposed in this paper. We will describe the experiments we have carried out more explicitly. We will clarify that the final results used for comparison are the result of the SCAM simulation process and that the surrogate models are only used to provide parameter values. This is in line with the ultimate aim of this paper, which is to apply the optimized results to SCAM.

*We have also carefully read each of the detailed descriptions you mentioned in relation to the text and captions, including the units consistently on the y-axis.*

Figure 5: Could you define 'maximum fluctuation' more precisely? Is this the difference between lowest and highest PRECT value, and is the value of PRECT output hourly/daily/etc?

It is the difference between lowest and highest value of PRECT output. The output value here refers to the average value throughout the simulation, for each case. We will add some necessary information in our revised manuscript.

Line 224: "However, the complexity of the calculation increases exponentially during the test, as shown in Table 4." I'm not sure Table 4 is the right reference here; it shows the SA methods used, but does not seem to indicate complexity or cost of these tests.

We are sorry that the mistaken reference has caused confusion. The reference here should be to Equation 1 and not to Table 4 in the original manuscript. We will add more detail here in the revised version of our manuscript. What we are trying to convey here is that, as can be seen by Equation 1, when calculating the effect of the combined parameters on the results, the $N_{MPP}$ increases exponentially as $p$ increases due to the position of $p$ in the exponential. For ease of reading, we have also included Equation 1 in this attachment.

$$N_{MPP} = C \times \binom{M}{p} \times L^p \tag{1}$$

Figure 9: Which SA method is being used in this figure? Units on the y-axis would also be helpful.

The single parameter perturbation method is used here, i.e. keeping the other parameter values constant at their default values and tuning only the value of one parameter linearly. To illustrate the problem more clearly, we will also add the units of the y-axis.

Figure 12: Please add units to the y-axis.

We will add units to the y-axis in Figure 12 in the next manuscript submission.

Figure 13: A more detailed description of this figure in the text would be helpful. It's unclear what improvement is being plotted, and what the 'original' case in blue is given that the 'enhancement of effects' on the y-axis is due to the addition of NN and/or grid searching. Is the blue not the control case then?

The blue bars indicate the improvement in effectiveness relative to the control experiment when using the traditional SA method combined with a single optimization method. The three bars show the improvement of the three different test approaches compared to the control test. The control test is used as the baseline for the three scenarios here. We will add more appropriate descriptions in the revised manuscript.

Figure 14: Similarly, more detail would be useful. What units are the overhead in computing given in? Is it obvious that the overhead should be the same for the Original and NN cases?

The units used are the number of hours it takes to perform a simulation. As NN's improved approach relative to *Original* is mainly reflected in the sensitivity analysis, and this part of the experiment does not involve running more SCAM instances, the change in computing time is not reflected significantly, and therefore the difference in computing time is less reflected. We will refine these descriptions in the revised manuscript.

Lines 421-422: "we can see that the parameter values taken between the two cases of land convection are positively correlated in the same parameter space" it's surprising to me that the correlation is equal to 1 between ARM95 and ARM97 despite apparent differences between them in Fig 15. Could the authors elaborate on why this occurs?

This might be related to the pre-processing that the vectors undergo before they are involved in the calculation. Given that we have re-trained the model, new results will also be presented in our revised version, as shown in AC-Figure 2.

[Figure]

AC-Figure 2: Correlation of the optimal solutions of the cases in the same parameter space.

**2   Replies to specific comments**

Lines 1-2: "The Single Column Atmospheric Model (SCAM) is an essential tool for analyzing and improving the physics schemes of CAM." Please specify that CAM in this case is the Community Atmosphere Model.

We will add this specification in a revised version of the manuscript.

Lines 6-8: "By reusing the 3,840 instances with the variation of 11 parameters..." Suggest avoiding using specific numbers like this in the abstract; without an explanation, it is unclear what "the 3,840 instances" are. Either add clarification/context, or remove the specific number of instances.

We will make the appropriate changes in the revised abstract and remove the specific number of instances.

Line 15: Should this read "the effects of global climate change"?

We will correct this in the revised version of the manuscript according to your comment.

Line 18: This citation of CAM is outdated, please point to the scientific articles describing CAM instead. The title of this particular citation in the references points to CAM3 and the link itself is broken.

We will refine the citations to the references in the revised manuscript and ensure that the links are all accessible. References [2] and [3] will be added to make the descriptions more precise.

Line 18: "Of these components, the Community Atmosphere Model (CAM) (UCAR., 2020), is the one with the most complexity." It's hard to say that more model complexity is contained in one model component than another; could the authors clarify/justify what's intended by this statement?

The intention here is to illustrate the complexity of CAM and thus set the scene for the introduction of SCAM below. We will revise these descriptions as: "The use of SCAM for large-scale experiments is more practicable due to its advantage of lower requirements for computing resources."

Line 20: "Participated in continuous numerical integration," - Consider rephrasing for clarity; do the authors mean that in coupled climate simulations, this is a source of uncertainty?

The main purpose here is to highlight the complexity of GCM and thus illustrate where the advantages of SCAM lie. We will rephrase these descriptions in the revised version as :"Considering the uncertainty of schemas, it is more convenient to select models with low computational cost, such as SCAM, to conduct large-scale parameter tuning experiments."

Line 22: "However, as a general circulation model (GCM), CAM takes a long time and a large amount of resource to run..." Given that ESM is already defined above, the authors should continue to use that notation rather than also defining GCM (unless a distinction is intended, which could be elaborated on).

We will correct this issue in the revised version. All notations for the same definition will be unified.

Line 24: "good alternative model" – rephrase for clarity. What's meant by 'good' here – cheaper, more efficient, etc?

Cheaper computational overhead and higher efficiency are both advantages. We will rephrase it to make this more clearly expressed.

Line 25: What is meant by SCAM only needs "one process"?

Since SCAM is a small and fast model, it runs only on one processor. In one simulation of SCAM, only one process is required for each run of one case to complete the computation.

Lines 28-29: "Sensitivity analysis (SA) is a method for investigating how uncertainty in the model output is assigned to the different sources of uncertainty in the model input factors, and the participants (Saltelli et al., 2010)." It's not clear what the participants are; please clarify.

The term 'participants' refers to the independent variables in the problem under study. We will describe this in more concise terms in a revised version of the manuscript.

Line 41: "quasi-random sequence by Sobol (Sobol', 1967) and other researchers)," please specify the other researchers who have established the method so that it can be easily referenced by readers. It may also be useful to briefly explain what the "low-discrepancy quasi-random sequence" is if relevant to the study.

We will refine these descriptions so that the reader can more easily understand the role that this quasi-random sequence has played in previous studies as well as in the present study.

Line 48: "After we identify the important tuning targets in the SA stage" – how are those targets usually defined? This hasn't been explained in the previous paragraph, only that there are different methods for sampling parameters. Please briefly note how that translates to identification of targets (even a sentence should suffice).

Targets, in this context, refer to the parameters (or a combination of them) that are more sensitive to the results obtained during the SA phase of the analysis. Having identified these targets, readers can then know which parameters to tune to have a greater impact on the results, thus making it easier to get better tuning results. We will refine these descriptions as "After we have determined the combination of parameters to be tuned".

Line 70: "By reusing the 3,840 instances with variations of 11 parameters" – there is no indication of where these numbers are coming from or what they refer to. This should be introduced in the methods section, so wait until that point to elaborate on specifics like this.

We will complete the description of the origin of this sample of figures in the Introduction.

Lines 75-82: please condense into a paragraph rather than bulleted list.

We will condense and refine this part of the text in a revised manuscript.

Lines 77-79: "By reusing the 3,840 sampling instances..." - Again, where does the number of instances come from? And when the authors say the model achieves good accuracy, which variables are they referring to (i.e., the "error within 10%" is an error across which variables?)

For the sample size, we will go into more detail as to why this sample number was selected. The latter achievement refers to the improvement in the model fit to total precipitation, i.e. the variable is PRECT.

Lines 89-91: "case-specific tuned parameters would further reduce the precipitation error by 15% when compared to a set of unified tuned parameters, and suggest a potential improvement from location-wise parameter tuning in the future." I did not see the discussion of a case where all cases are combined to find the optimum parameter values. Please elaborate further on that in the results section to support this.

In the revised edition we will add clarifications and refine the descriptions so that the reader can understand them more easily. As shown in Figure 1.

Lines 98-100: Is it wise to tune for just a single variable (time-mean PRECT)? Realistically, when assessing model performance and tuning accordingly, a number of performance metrics need to be accounted for beyond the mean of one variable. Could the authors elaborate on the validity of selecting just one, perhaps?

PRECT is an output variable included in the IOP files of all five cases covered in this manuscript, and its use as an object of study facilitates cross-sectional comparisons between the cases and the analysis of their relationships.

Lines 111-114: "In addition, the programs running on Sunway TaihuLight needed to be recompiled due to the adoption of a different archiecture." - Shouldn't the model be recompiled at build time? Is this a unique addition to the model, that enables compilation with a non-supported compiler? Is the source code available and going to be included in CESM?

As we chose Sunway TaihuLight as our experimental platform, we were able to make the code work on this system by attempting to port and compile it. By modifying the source code of the model, the parameters involved in the paper can be tuned via the namelist input file. As a result, there is no longer any need to recompile each time one experiment is carried out, which also makes it much more efficient. The code is currently available.

Lines 184-185: "We set the learning rate..." – could the authors elaborate on if this is the most suitable choice of learning rate/batch size? Were other values tested?

The values chosen are empirical parameter values of learning rate and batch size that we commonly use for neural network model training. This paper focuses on the feasibility of using neural network methods for large-scale parameter analysis and tuning, and therefore the empirical values were chosen for testing. This set of parameter values is a selection of the better performing values after testing several sets of values. We have conducted an ablation experiment for learning rate and batch size and will detail the process and results of this experiment in a revised manuscript.

Figure 5: There's a relatively wide variability in this across cases, with ARM97 and TOGAII being the least sensitive and GATEIII being very sensitive to the number of parameters to be tuned. Worth

elaborating on?

What this figure reflects is indeed of some interest. The original meaning of the figure was how much output fluctuation (in terms of average precipitation during the simulation) could be produced for each case when tuning one to four parameters, respectively. The graph does show that for the ARM97 and TOGAII, their precipitation response for four parameters is even smaller than the response of the GATEIII for tuning one parameter. An important reason for this is that these two cases themselves have smaller precipitation values than GATEIII, whereas the figure uses the absolute values of precipitation. This phenomenon does deserve further elaboration and will be discussed in the revised version.

Lines 225-226: "...although the effect of tuning four parameters was better than tuning three parameters, the advantages of the surrogate model in the parameter tuning process could not be exploited at this point." I'm unclear why "the advantages of the surrogate model in the parameter tuning process could not be exploited" when using 4 rather than 3 parameters.

Testing combinations of parameters on surrogate models is also computationally costly. As the size of the test increases exponentially with the number of parameters to be tuned, the computational time will no longer be negligible when tuning four parameters. The results also show that the improvement of maximum tuning effect that can be achieved by tuning four parameters is limited compared to tuning three parameters. Therefore, considering both the computational overhead and the tuning effect, we chose to tune three parameters for the experiments in this paper.

Lines 229-230: "combinations of three parameters lead to the most significance in output" – is this meant to imply the three parameters that drive the most significant change in PRECT, or the most significant improvement (assuming those are different, they could be the same, but a big change does not necessarily lead to improvement).

Theoretically, the more parameters that can be tuned in one experiment, the greater the variation in the results that can be brought about. The reason for choosing to tune three parameters here is also to achieve a balance between computational overhead and tuning effect. This allows the sensitive parameters to be tuned while avoiding the non-critical parameters consuming computational resources.

Lines 255-257: For complete reference, please also explain *epsilon* and $p$ as they are used in Algorithm 1.

We will explain in detail the meaning of these two variables and the role they play in this algorithm. $\epsilon$ is the threshold at which the results converge and $p$ is the total number of parameters to be adjusted.

Line 260: "Meaning average error" – is this meant to be the mean absolute error? Or is this something else?

It refers to MAPE (Mean Absolute Percentage Error). In addition, after careful discussion,

we will instead use RMSE (Root Mean Square Error) as a measure of error. We will correct it in the revised version.

Lines 290-292: "We use a total of 7,680 samples, with 1,536 samples for each of the five SCAM case." - Please elaborate on the reason for choosing this number of samples – how many values per parameter are enabled by this choice? Is there a clear reason for running 1,536 samples per IOP? This also seems like a detail that should be included in the methods rather than the results.

As with the answer to the question on line 70, since this research involves two methods of generating sample sequences, MOAT and Saltelli, each of which has a different formula for generating the number of samples. Their respective base numbers are different and the choice to generate 1536 samples per case (768 for each of the two methods) was also made in view of the fact that 768 is a appropriate sample size that can be generated by both methods.

Lines 298-299: "...although the medium point varies from 5 to 12 mm per day, demonstrating clearly different climate patterns." - How much of this is due to differences in the length of the IOP (perhaps one captured more dry days than another, for example), or an IOP designed to capture shallow vs. deep convection? This may not necessarily be indicative of obviously varying climate patterns.

Indeed, the five IOPs involved in this study differed in their location, length of capture and time of day. For example, GATEIII has a longer capture time, while TOGAII has a relatively shorter capture time.

Lines 318-319: "The reason for this difference probably comes from a different time of year and the forcing field simulated in these two cases." It would be good to see a more confident assertion here. How different is the time of year assessed, is it substantial enough to cause such a change in sensitivity? How different is the forcing field (and does this hypothesize that it's the large scale T or Q convergence that's responsible)? Is there a difference in the type of convection that occurs as well?

Forcing fields are an important cause of the different properties of these IOPs. The difference between the cases is also the focus of our interest and attention. Indeed, the gap between ARM95 and ARM97 indicated by the experimental results in this paper is worthy of attention. By comparing their forcing fields, we find that there is indeed a certain degree of difference between them in T and Q.

Line 322-323: "Instead of calling SA methods directly, we use combinatorial analysis of the magnitude of change to determine the effect that these parameter combinations have on the model output." I'm not sure on what this means, please rephrase for clarity?

This refers to the invocation of the method proposed in this paper for combined parameter analysis, rather than the existing fixed method.

Lines 330-332: "For example, increasing tau tends to increase total precipitation in GATEIII, while in the other cases it brings the opposite result." - Is there something special about that case that causes the unique signal?

This case does show a unique signal of interest at this point, which may be related to

the nature of the GATEIII case itself. A similar conclusion was reached when we explored the parameters with the help of the surrogate model.

Lines 353-353: "The impact of such differences is even multiplied" - unclear what is meant by this statement; what is being multiplied here?

In this sentence, we are trying to express the level of impact by using the word *multiplied*.

Lines 366-368: "It is easy to see that in the control experiment there were several spikes where the simulated output was significantly higher than the observed values, as was the case in the first four cases. After tuning, these spikes are significantly weakened and the output is much closer to the observed values." It looks like this is really only an issue in the land-based ARM cases; is that true? It looks like the tuning is still unable to match some of the largest rain rates in GATEIII especially but also TOGAII – is there a reason for that?

This phenomenon was indeed more pronounced in the two land-based ARM cases. On average, our tuning is also effective for the three tropical convection cases. The inability of the tuning to match the total precipitation for GATEIII does exist, as can also be seen from the analysis of the sampling results in Figure 7(c). The failure to cover the total precipitation that matches the observations in the sampling results means that the likelihood of finding the optimal solution through tuning is small. In contrast, this possibility is still present in TOGAII. The existence of this result is justified by the fact that there is a certain margin of error in the simulation of the model itself.

Lines 368-369: "This demonstrates the significance of the parameter tuning provided by the workflow for model." Could the authors be more quantitative here? How much is the bias reduced by, for example?

The main purpose here is to highlight the important role that scientific workflow plays in the work of this paper. The methods we present in this paper are organized in the form of a workflow, and the entire reconciliation process is done coherently. We will describe this more quantitatively in the revised version.

Lines 399-400: "It is easy to see that the two land convection regions are closer and, accordingly, the three tropical convection aggregation regions are also closer." It looks like the land cases might be fairly different, particularly in terms of optimal pz4 value. A table would make it easier to compare the 'optimal' values, even if they're given by a range of what's marked in red in Figure 15.

We agree that the specific values should be more intuitive and readable for the reader. We will add this to the revised edition.

Lines 402-403: "the distribution of the better value points is different for different parameters even for the same parameter space." Should this read that the better values are different for different cases within the same parameter space?

We will correct this in the revised version so that it will be easier for the reader to understand.

Line 404: "In the other three cases, smaller values of tau lead to better performance." It looks like the optimal value of pz4 occurs at/near the minimum range for both GATEIII and TWP06 - have the authors tested expanding the lower limit of this variable further to see if this is the optimum value or if it's being cut off?

Indeed, as can be seen from the figure, the optimized values obtained for both cases do lie at the boundary. Thus the possibility exists that there may be better values outside the bound than inside that bound. However, the values of the parameters cannot be infinitely large or small, and we should also take into account their physical meaning.

Lines 405-406: "On the other hand, it also shows that there are differences in the distribution of parameters that make the results perform better in different types of cases." Rephrase? This sounds like it's saying the same thing as the sentence before it.

Indeed, as you say, we will use more concise phrases in the revised version: "This reflects the fact that it may be useful and necessary to adopt different parameter configurations for different cases or regions."

[Figure]

(a)                              (b)

AC-Figure 3: Results of a three-parameter full-space grid search for ARM95 and ARM97 using the surrogate model. The points closest to the observed data are shown in red, those ranked 2-64 are shown in olive, and those ranked 65-256 are shown in blue. Where, (a) is ARM95 and (b) is ARM97.

[Figure]

(a)                              (b)                              (c)

AC-Figure 4: Results of a three-parameter full-space grid search for GATEIII, TOGAII and TWP06 using the surrogate model. The points closest to the observed data are shown in red, those ranked 2-64 are shown in olive, and those ranked 65-256 are shown in blue. Where, (a) is GATEIII, (b) is TOGAII and (c) is TWP06.

Lines 407-408: "It can be got that the optimal value points for the two land convection cases are close, while the points for the three tropical convection cases are even closer." While the GATEIII and TWP06 cases are very similar, the difference in the TOGA case challenges the notion that tropical convection cases are closer than what we see in the ARM case. Over land, it also looks like the pz4

optimal values are actually rather different; a table would make this argument more convincing and easier to see, potentially. Or may point to the need for a more nuanced statement.

Indeed, as you have said, further details could be given in terms of specific optimized values. A more detailed elaboration would make our experimental results and views more convincing. At the same time, we conducted another complementary experiment, which was to try to introduce different methods to train the surrogate models for SCAM cases. Through our research, we learned that both XGBoost[4] and ResNet[5] can be used to perform regression tasks and train surrogate models. Here, we will compare the effectiveness of several methods such as LR (Linear Regression), RF (Random Forest), MLP (Multi-Layer Perceptron), XGBoost and ResNet for training surrogate models. The results are shown in AC-Table 1. The RMSE was used to measure the error generated during training. Based on these results, we used ResNet to retrain the surrogate models, and when we used these later trained models to perform a grid search in the same parameter space, we found that TOGAII and TWP06 have more similar distribution patterns in the parameter space, as can be seen from AC-Figure 4. This is also consistent with the above distribution of the two cases in terms of position. This is due to errors in the previous training models, and we would appreciate your prompt correction.

Lines 416-417: "The difference between the two cases lies mainly in the time, which therefore reflects that there is also a difference in SCAM's simulation performance for different times." I don't see a time-based sensitivity analysis in this; could the authors clarify/elaborate? Is this a seasonal difference? Were other alternative hypotheses explored to explain the difference in ARM95 and ARM97 (different synoptic conditions, etc)?

The term 'time' here refers not to a time-based sensitivity analysis, but to the historical time simulated by these SCAM cases. At the same time, we conducted another complementary experiment, which was to try to introduce different methods to train the surrogate models for SCAM cases. Through our research, we learned that both XGBoost[4] and ResNet[5] can be used to perform regression tasks and train surrogate models. Here, we will compare the effectiveness of several methods such as LR, RF, MLP, XGBoost and ResNet for training surrogate models. The results are shown in AC-Table 1. The RMSE was used to measure the error generated during training. The best performers are shown in bold. Based on these results, we used ResNet to retrain the surrogate models, and when we used these later trained models to perform a grid search in the same parameter space, we found that their distributions were very similar, as can be seen from AC-Figure 3. This is due to errors in the previous training models, and we would appreciate your prompt correction.

AC-Table 1: RMSE of different surrogate models for five cases.

| Case | LR | RF | MLP | XGBoost | ResNet |
|---|---|---|---|---|---|
| ARM95 | 0.235 | 0.197 | 0.751 | 0.184 | **0.038** |
| ARM97 | 0.188 | 0.158 | 0.555 | 0.136 | **0.045** |
| GATEIII | 0.646 | 0.432 | 1.335 | 0.538 | **0.137** |
| TOGAII | 0.179 | 0.112 | 0.223 | 0.118 | **0.041** |
| TWP06 | 0.344 | 0.220 | 0.594 | 0.220 | **0.040** |

Line 419: "the optimal parameter values for each case can be represented by a vector" - I'm not

entirely clear on what this vector would look like; it might help the reader to plot said vector on Fig 15.

The term *vector* here refers to the optimized solution of a set of parameters as a vector. This is because the computation of the Pearson correlation coefficients is done on the basis of vectors. We will refine this description in a revised version.

**References**

[1]  J. Guo, Y. Xu, H. Fu, W. Xue, L. Wang, L. Gan, X. Wu, L. Hu, G. Xu, and X. Che, "A learning-based method for efficient large-scale sensitivity analysis and tuning of single column atmosphere model (scam)," *Geoscientific Model Development Discussions*, vol. 2022, pp. 1–28, 2022. DOI: 10.5194/gmd-2022-264. [Online]. Available: `https://gmd.copernicus.org/preprints/gmd-2022-264/`.

[2]  J. T. Bacmeister, M. F. Wehner, R. B. Neale, A. Gettelman, C. Hannay, P. H. Lauritzen, J. M. Caron, and J. E. Truesdale, "Exploratory high-resolution climate simulations using the community atmosphere model (cam)," *Journal of Climate*, vol. 27, no. 9, pp. 3073–3099, 2014, Cited by: 163; All Open Access, Green Open Access. DOI: 10.1175/JCLI-D-13-00387.1.

[3]  J. M. Dennis, J. Edwards, K. J. Evans, O. Guba, P. H. Lauritzen, A. A. Mirin, A. St-Cyr, M. A. Taylor, and P. H. Worley, "Cam-se: A scalable spectral element dynamical core for the community atmosphere model," *International Journal of High Performance Computing Applications*, vol. 26, no. 1, pp. 74–89, 2012, Cited by: 258. DOI: 10.1177/1094342011428142.

[4]  W. XingFen, Y. Xiangbin, and M. Yangchun, "Research on User Consumption Behavior Prediction Based on Improved XGBoost Algorithm," en, in *2018 IEEE International Conference on Big Data (Big Data)*, Seattle, WA, USA: IEEE, Dec. 2018, pp. 4169–4175, ISBN: 978-1-5386-5035-6. DOI: 10.1109/BigData.2018.8622235. [Online]. Available: `https://ieeexplore.ieee.org/document/8622235/` (visited on 04/03/2023).

[5]  L. Shi, C. Copot, and S. Vanlanduit, "Evaluating Dropout Placements in Bayesian Regression Resnet," en, *Journal of Artificial Intelligence and Soft Computing Research*, vol. 12, no. 1, pp. 61–73, Jan. 2022, ISSN: 2449-6499. DOI: 10.2478/jaiscr-2022-0005. [Online]. Available: `https://www.sciendo.com/article/10.2478/jaiscr-2022-0005` (visited on 04/03/2023).

---

## Author Comment (AC2)

**Replies to Referee #2, GMD-2022-264**

Jiaxu Guo on behalf of all authors

April 18, 2023

Thank you very much for your patient and detailed comments on our work[1]. These valuable comments are very helpful for us to improve this paper. After carefully reading all the questions, we have answered each of them and will make appropriate corrections in the revised version of our manuscript.

In this attachment, the red paragraphs represent your refree comments, and the black paragraphs below are our corresponding replies.

**1 Replies to major issues**

In this paper, although the authors evaluate the accuracy of the NN model in terms of precipitation, it probably exists the inconsistent between ML and real model, which don't be highlighted in this paper.

We agree that it may be difficult for a surrogate model to be fully consistent with a real model. We use the RMSE in training as a loss function to verify the correctness of the method, that is, whether the parameter tuning of SCAM can be accelerated by training a surrogate model, and to compare different regression methods. We will add more descriptions about the inconsistency to the manuscript from two perspectives: From a model training perspective, the value of loss function indicates the error in the training. However, the process by which we train the model is also the process by which the error is gradually reduced. From a practical perspective, when we use the surrogate model for optimization, the solutions obtained are also validated in the original SCAM case to ensure as much consistency as possible.

The authors believe that due to the high computational cost of the GCM, the SCAM can be the alternative model for parameter SA and tuning. In reality, the optimal parameters tuned in SCAM could not be suitable for GCM, due to the global regions and more complex large-scale circulation.

We agree that the solution set obtained by parameter tuning on SCAM is not directly applicable to GCM. However, through our attempts at parameter tuning on SCAM cases located in different regions, we can find commonalities and patterns in the parameter response of these

cases. We believe such an idea can be applied to the parameterization scheme of the GCM. Our exploration of parameterization solutions using SCAM is mainly from a methodological and ideological point of view. That is, SCAM is a simpler and less costly way to perform numerical simulations. The training of a surrogate model for SCAM is a further extension of this idea.

Overall this manuscript, the organization and writing are not clear and should be well structured. There are a very large number of language errors. The English writing should be greatly improved.

We will reorganize the structure of the article in a revised version to rationalize the logic. We will also work on improving the English writing style to make it more fluent.

[Figure]

AC-Figure 1: Results of a three-parameter full-space grid search for ARM95 and ARM97 using the surrogate model. The points closest to the observed data are shown in red, those ranked 2-64 are shown in olive, and those ranked 65-256 are shown in blue. Where, (a) is ARM95 and (b) is ARM97.

[Figure]

AC-Figure 2: Results of a three-parameter full-space grid search for GATEIII, TOGAII and TWP06 using the surrogate model. The points closest to the observed data are shown in red, those ranked 2-64 are shown in olive, and those ranked 65-256 are shown in blue. Where, (a) is GATEIII, (b) is TOGAII and (c) is TWP06.

The authors separately tune the parameters in SCAM for each site and get the different sensitive parameters and different optimal values. It is difficult to transfer this information to GCM. If the authors can do the multi-objective tuning for these sites with the same parameters, it could be helpful for global model tuning because these SCAM sites indeed represent the different regimes.

We agree that a combined optimization that tries to minimize the differences against observations across all five cases would be more meaningful. Such tests and results were included in a previous version of the draft. However, we removed the results at certain stage for a more focused description in the result section.

As suggested by the reviewers, we have carried out a careful analysis and done corresponding experiments. After retraining the surrogate model for each of the five cases by using ResNet,

[Figure]

AC-Figure 3: Results of a three-parameter full-space grid search for the multi-objective scenarios using the surrogate model. The points closest to the observed data are shown in red, those ranked 2-64 are shown in olive, and those ranked 65-256 are shown in blue. Where (a) is the scenario of two land convection cases, (b) is the scenario of three tropical convection cases, (c) is the scenario of two western tropical Pacific cases, and (d) is the scenario of all five cases.

we also carried out multi-objective optimization for each of the four scenarios. Separately, they are: for ARM95 and ARM97 cases, for three tropical convection cases, for TOGAII/TWP06 (they're close in location) and all five cases.

Combined with results shown in AC-Figure 1 and 2 of our grid search for the five cases individually, it can be seen that cases located at the same or similar locations really have similar distributions of precipitation output in response to parameters. Therefore, this kind of joint optimization based on the multi-objective idea is of general interest.

For the workflow, there should be a "metrics" component because it is very important for tuning. No matter SCAM or GCM, the tuning metrics could be the cost function between model simulations and observations. The different designs could affect the optimization. In terms of the metrics, it could consider the 1) different statistic errors between simulation and observation, such as RMSE, performance score like Yang et al. (2013), 2) one objective or multiple objectives, and how to deal with the multiple objectives.

I agree that "metrics" are important factors to consider in the workflow. We will use RMSE as the metric of error in the revised version.

$$RMSE = \sqrt{\frac{1}{N}\sum_{i=1}^{N}(y_i - \hat{y}_i)^2} \tag{1}$$

After using RMSE as the metric for training the model and parameter optimization, the corresponding results were recalculated. In particular, the losses during the training of the model are shown in AC-Table 2. The 3D parameter space responses for the five cases themselves and for the four multi-objective joint conditioning scenarios are shown in AC-Figure 3 to 2.

Regarding multi-objective tuning, we also designed corresponding experiments when we originally wrote this paper. Four scenarios are included, separately, they are: combined tuning for ARM95 and ARM97 cases, combined tuning for three tropical convection cases, combined tuning for TOGAII/TWP06 and all five cases. From results we can see that the parameter responses for ARM95 and ARM97 have a similar distribution and thus the results of the multiobjective optimisation for both of them reflect this. The same trend is reflected similarly in the TOGAII and TWP06.

Line 35: The statement that the Morris SA method cannot get the interactive sensitivity could be wrong. Aurally, the standard deviation of MOAT samplings can stand for the interactive effect of one parameter with others (Morris, 1991).

We will refine the description of the Morris SA method in the revised manuscript so as not to introduce ambiguity: "Morris SA can give the individual sensitivity of each parameter, including their interaction sensitivity. However, this is not intuitive enough if the user wants to know directly from a combined perspective which set of parameters has the most significant effect on the results."

Line 45: as the part of introduction, the authors should explain the challenge of the SA methods, why you choose Morris and Sobol, the computational cost issue, surrogate problems using machine learning. If there are previous works, what's your contribution?

We will complete this part based on a further full investigation. In the revised version, we will give a more detailed explanation. Both Morris and Sobol are typical SA methods that have a wide range of applications in many fields. As there are already proven application examples, it makes sense to conduct experiments based on the above methods. In addition to this, we have introduced several new SA methods that have been proposed in recent years. Although SCAM, as a single-column model, already consumes less computational resources than GCM, the computational resources of a system are not infinite. To further explore their parametric features, a study using surrogate model is necessary. Surrogate models [2] can significantly reduce the computation time of individual tasks, thus making it possible to scale up experiments.

Our contribution lies in the fact that we have trained the surrogate models on SCAM using a regression-based approach, and with the help of the surrogate models we have conducted parameter sensitivity tests for combinations, as well as tuning for the most significant parameter combinations.

Line 55: the authors should do comprehensive literature research, even for GCM, there are a large number of work for tuning, such as Yang et al. (2013) and Zou et al. (2014). In addition, the NN surrogate model is used to tune as well. But the authors don't introduce the previous work and challenge in terms of this issue. The introduce section should be more clear.

We will complete the introduction section by conducting a more detailed literature survey of the work related to the content of this paper. This also includes the literature you mention, such as [3] and [4]. Yang et al. [3] analysed the sensitivity of nine parameters in the ZM deep convection scenario for CAM5 and used the SSAA (Simulated stochastic approximation annealing) method to optimize the precipitation performance in different regions by zoning. Zou et al. [4] conducted a sensitivity analysis for seven parameters in the MIT-Emanuel cumulus parameterization scheme in RegCM3. The precipitation optimization process for the CORDEX East Asia domain was carried out using the MVFSA (Multiple very fast simulated annealing)

method.

The current challenges lie in the following areas. (1)No similar experiments have been conducted on SCAM. The short run time of the SCAM makes it easier to obtain more samples in a short period of time and thus scale up the experiments. (2)The usual SA methods will give the sensitivity of the individual parameters. But when we look at a set of parameters, is the best combination of N parameters the top N sensitivity of a single parameter? This is a question worth exploring. (3) As various neural network methods are applied to the field of regression, more appropriate regression methods should be applied to the parametric study of Earth system models. Unlike the various public data sets commonly used, ESM-based experiments will also further enrich the practical implications of research in the field of regression analysis.

In our manuscript, we innovatively use a neural network-based agent model for parameter tuning of different cases in SCAM. For each case studied in the paper, the surrogate models are trained separately based on different methods, and the best performing model is selected by their training errors (RMSE).

Line 75, Acutely, there are existing SA and tuning workflow used in climate models, such as PSUADE and DAKOTA, the authors don't compare their workflow to these packages. It's not new for the community.

This is indeed something we need to improve further in the literature survey. We have done a survey of the packages such as PSUADE, DAKOTA and STATA etc., including some studies based on their work in different fields. These packages can indeed implement the functionality of SA and tuning. We also compared the workflow proposed in this paper with the above software packages. From a method perspective, we added the comparison of the new SA methods in recent years, such as RBD-FAST, Delta and HDMR. The above approaches haven't been fully supported by all the packages above. For using neural networks to train surrogate models, DAKOTA currently only supports neural networks with a single hidden layer. Meanwhile, the proposed method uses more types of neural networks and supports the adaptive selection of the best performing network to train the surrogate model.

Line 88: The authors don't mention the 30% improvement for tuning error and computational cost. How do they come from?

The tuning error is compared with control experiments. This is the result of a simulation using the default parameter values of SCAM. The improvement in computational overhead comes from comparing it with a traditional optimization workflow. In the original optimization workflow, the original SCAM is invoked for each calculation of the objective function. In the approach proposed in this paper, the objective function is replaced with a surrogate model. This allows the overall workflow execution time to be compressed, thus the computational overhead could be reduced.

Table 2 is wrong. Each IOP file includes many variables, not just these four variables. Therefore, the statement that you choose precipitation is wrong.

AC-Table 1: Observed variables included in the IOP file of each case.

| Variable | Description | ARM95 | ARM97 | GATEIII | TOGAII | TWP06 |
|----------|-------------|-------|-------|---------|--------|-------|
| Prec | Precipitation rate | ✓ | ✓ | ✓ | ✓ | ✓ |
| totcld | Total cloud | ✓ | ✓ | - | - | ✓ |
| shflx | Surface sensible heat flux | ✓ | ✓ | - | ✓ | ✓ |
| lhflx | Surface latent heat flux | ✓ | ✓ | - | ✓ | ✓ |
| U | Eastward wind speed | ✓ | ✓ | ✓ | ✓ | ✓ |
| V | Northward wind speed | ✓ | ✓ | ✓ | ✓ | ✓ |
| Q | W.V. Mixing Ratio | ✓ | ✓ | ✓ | ✓ | ✓ |
| T | Temperature | ✓ | ✓ | ✓ | ✓ | ✓ |
| omega | vertical motion | ✓ | ✓ | ✓ | ✓ | ✓ |
| windsrf | Surface wind speed | ✓ | ✓ | - | ✓ | ✓ |
| REHUM | Relative humidity | - | - | ✓ | - | ✓ |
| CAPE | Convective available potential energy | - | - | ✓ | - | - |

We are sorry for the misunderstanding. We agree that there are many variables in the IOP file for each case, not just these four, as can be seen from AC-Table 1. We include Table 2 in our manuscript mainly to illustrate that the variables contained in different IOP files are different. We will add this in the revised manuscript.

Line 120: This issue could not be a significant challenge. Some simple scripts can collect the data.

Yes, scripts are able to do that. Of course, we will also achieve efficient management of experimental data in our method. Due to the amount of data generated during this stage, better data management is necessary. We will adopt a more reasonable description in the revised manuscript.

We will revise these statements as follows: "In this step, we also integrate the collection and processing script for the post-sampling results. As the output of SCAM is stored in binary files in NetCDF format, the precipitation variables we want to study need to be extracted from a large number of output files in order to proceed to the next step. This will further accelerate the degree of automation of scientific workflows and thus accelerate the conduct of research in this area of the earth system models."

The statement about Morris is wrong, see 3.

We plan to correct this issue as follows: "Morris SA can give the individual sensitivity of each parameter, including their interaction sensitivity. However, this is not intuitive enough if the user wants to know directly from a combined perspective which set of parameters has the most significant effect on the results."

In section 3.1: sampling is not equal to SA. In this section, the authors introduce the SA methods. You should consider change the structure or change the title.

We agree that sampling can't be confused with SA. We will refine the article structure of this section in a revised version to make the expression easier to understand. The sampling method and the SA method will be split into two subsections to be described.

AC-Table 2: RMSE of different surrogate models for five cases.

| Case | LR | RF | MLP | XGBoost | ResNet |
|------|-----|-----|------|---------|--------|
| ARM95 | 0.235 | 0.197 | 0.294 | 0.184 | **0.038** |
| ARM97 | 0.188 | 0.158 | 0.555 | 0.136 | **0.045** |
| GATEIII | 0.646 | 0.432 | 0.108 | 0.538 | **0.137** |
| TOGAII | 0.179 | 0.112 | 0.245 | 0.118 | **0.041** |
| TWP06 | 0.344 | 0.220 | 0.304 | 0.220 | **0.040** |

Subsection 3.1, entitled "Sampling methods used in this framework", will be devoted to the sampling methods covered in this paper and the rationale for their selection.

Subsection 3.2, entitled "SA methods used in this framework", will provide a more detailed description of the SA methods covered in this paper and the rationale for their selection. At the end of this subsection, our proposed combined SA methods for supplementary validation will be introduced.

Line 177: it could be better to compare NN with other surrogate models, such as xgboost, ResNet.

Yes, the properties of the various surrogate models are something we all care about. Through our research, we learned that both XGBoost[5] and ResNet[6] can be used to perform regression tasks and train surrogate models. Here, we will compare the effectiveness of several methods such as LR (Linear Regression), RF (Random Forest), MLP (Multi-Layer Perceptron), XGBoost and ResNet for training surrogate models. The results are shown in AC-Table 2. Since ResNet has better performance in model training, we will choose ResNet as the network for training the surrogate model. Subsequent experiments will also be supplemented in the revised paper.

Line 184: The 768 samples seem not enough for training NN, do the authors evaluate the performance of NN? In Figure 4, how do you define the accuracy?

Yes, we have evaluated the performance of NN and measured its ability on regression. After our analysis of the sampling results, we found that 768 samples can already cover the range of values of the parameters and output variables. Therefore, the number of parameters selected is sufficient to obtain a good training effect. The results of the training process and the experimental results also support our conclusion. For the second point, The $R^2$ score is used as the accuracy rate in Figure 4. It is defined as

$$R^2 = \frac{\sum_i (\hat{y}_i - \bar{y})^2}{\sum_i (y_i - \bar{y})^2} \tag{2}$$

where $\hat{y}$ is the predicted value and $\bar{y}$ is the mean of the test set. In addition to this, RMSE was also used as an important metric to measure the loss during training.

Line 185: Do the authors do the hyper-parameter tuning for NN?

In the initial version of our manuscript, our aim was mainly to verify the feasibility of this approach, so we used a set of empirical parameters to train the NN. Now we will conduct

hyperparameter tuning trials in NN training to make the proposed method more solid. We will refine these in a revised version of the manuscript.

Line 195: Due to the uncertainties of each method, ensemble can't guarantee to reduce the error.

We agree that there are errors in out initial manuscript. We feel sorry for the term "integrate" in the original paper may bring some ambiguity. We've used several SA methods to make a more intuitive side-by-side comparison, which allows us to find the best method for each case. It's not necessarily about reducing the error.

Line 210: Equation (1), the left hand of this equation is not the number of processes. It should be the number of simulations. The number of simulations could depend on different sampling method. For Morris, it could not require such number of samples. It is not clear for this description.

Since SCAM is a single-process task, one simulation is equivalent to one process. Of course, as you said, it is necessary to distinguish between simulation and process more accurately. For Morris, 768 was used as the sample number to keep the sample number of various sampling methods consistent. We will add the above contents to refine these issues in a revised version of the manuscript.

For a SCAM simulation, it usually requires 10-20 minutes, why do you require more than one hour?

We agree that for a single SCAM job it only takes one process to run, and it does not take very long for a single SCAM to run. The one hour mentioned here is mainly the time it takes to complete a whole workflow of parameter analysis and tuning. This includes multiple iterations and delays in queuing batch jobs. We will refine these descriptions above in the revised manuscript and try to avoid ambiguities.

It is confusing that you can re-use the sampling from SA to train the surrogate model for tuning, but in section 3.3, you mention that the surrogate model can be also used in SA?

We agree that the trained surrogate model can be also used in SA. Since we want to find the most sensitive set of parameters in combination, this process requires a larger experiment scale. Therefore, to perform large-scale parametric analysis experiments faster, we use surrogate models to speed up the process.

Line 233: how do you get the conclusion? It is not convinced.

Here, the running time is compared to one full simulation of the original SCAM. Here we are trying to make two points. On the one hand, the surrogate model can simulate the output of SCAM more accurately. This makes it an effective alternative to SCAM in terms of parameter response. On the other hand, the execution time of the surrogate model is very short, which saves a lot of time for parameter analysis and optimization tuning. Combining these two reasons above, the use of a surrogate model for parameter tuning of SCAM is a very reasonable strategy. We will add some necessary information in our revised manuscript to make our conclusion more convincing.

It is very confusing for section 3.4 and is difficult to follow your idea. You could consider to re-organize this section.

We're sorry for not being able to make it easy for you to follow our idea. We will reorganize the language to make it easier to understand. In this subsection, we focus on an enhanced SCAM parameter tuning process. There are two main contributions. On the one hand, we use the trained surrogate model as the objective function for optimization, which allows the optimization time to be compressed considerably. On the other hand, we propose an enhanced parameter optimization process based on grid search. We use grid search to reduce the search range in the optimization process and thus obtain better results in fewer iterations. In the revised manuscript we will elaborate further on these two points above.

Line 270: how many samples do you have for the correlation? Do you do the p-value test?

Here, we carried out correlation analysis on the respective optimal solutions of the five SCAM cases. Therefore, a total of five vectors were used to calculate the correlation For the second point, now we will add tests on p-values in the revised manuscript.

Figure 7: it is difficult to evaluate the tuning performance in Figure 7. It could be better to use metrics like Yang et al. (2013).

What we want to express in Figure 7 is the statistics of the sampling results. The phrase "when the parameters are tuned" refers to the fact that the parameters are given different values during the sampling stage, not the final tuning stage. We're sorry for this ambiguity and will respect your suggestions to revise the description appropriately. When revising, we will fully refer to the relevant statements of [3] in order to make the results more clear and easy to understand.

Figure 4, Figure 5 are the results but appear in section 3 (methods). It could be reorganized.

The results described in these two figures belong to the pilot test, which is to prove the rationality of our experimental ideas. We will respect your suggestions and provide appropriate revisions to make the organization of our paper more rigorous.

Line 313: pz2 (c0_ocn) should be high influence on the ocean case. But in Figure 8, it doesn't have the high effect on PRECT at TWP. Could you explain the reason?

It can be seen from Figure 8 that no matter which sensitivity analysis method is used, pz2 (c0_ocn) has a certain influence on the results. It is a deep convection parameter related to the ocean-land intersection, but this does not mean that it must be the one that has the most influence on TWP. The influence of the ocean has been demonstrated here.

Line 317: the reason for the different between ARM 95 and ARM 97 is not convinced.

The difference between the cases is also the focus of our interest and attention. Indeed, the gap between ARM95 and ARM97 indicated by the experimental results in this paper is worthy of attention. We retrained the surrogate model using ResNet and used RMSE as the

metric of error. We found that the response of the precipitation output to the parameters in the new surrogate model was very similar for both cases which can be shown in AC-Figure 1. This suggests that in the original version, the difference between the two examples is due to the error in the model. Your advice on the choice of the neural network would also be much appreciated here.

Line 345: are the 16 iterations enough for convergence? Why don't use the general optimization, such as PSO, GA that you mentioned in the introduction section?

In our experiment, 16 rounds of iteration can already make the results converge, as shown in Figure 6 (in our preprint paper[1]). So the selection of the number of iterations is sufficient. WOA[7] is chosen here mainly because it is an optimization algorithm proposed later. Theoretically speaking, in the optimization stage of the workflow proposed in this paper, no matter what kind of optimization algorithm is selected, the purpose of optimization can be achieved, including PSO, GA, etc. With full respect for your suggestions, we can also compare the performance of these optimization algorithms in our framework.

**2    Replies to minor issues**

In Fig. 1, is there an arrow pointing from "SA methods" to "testing of combinations"?

There isn't an arrow here because only the trained surrogate models are used for testing combinations of parameters.

Caption in Figure 1: The sentence "SCAM launcher, the data collector and the jobs therein represent the batch execution of the SCAM algorithm" should be rephrased. What are the "jobs" and "batch execution"? It is not clear.

There isn't an arrow here because only the method of the surrogate model trained using neural networks will be used for parameter combination SA. Here, a job refers to a single simulation of SCAM. A batch is a collection of jobs that are submitted to the computing queue at once. We will elaborate more in detail in the revised manuscript.

Line 25: should explain "ne30"

This is a description of the spectral element method grids[8]. It refers to a model grid with a $ne30np4$ spectral element (approximately 1-degree) atmosphere and land grids. $ne[X]np[Y]$ are cubed sphere resolutions where X and Y are integers. The short name generally is $ne[X]$. We will add more details in our revised manuscript.

Line 38: the reference of Sobol method is wrong, pls use the original paper.

We will correct it [9] in the revised manuscript.

Line 43: The QMC and LHC are sampling methods, not SA methods.

We will correct this in the revised manuscript. Sampling methods and SA methods will be more clearly distinguished to avoid confusion.

In Table 3, how do you select these parameters? And how do you define the range of each individual?

These parameters were chosen mainly from reference [10]. The parameters range from 50% to 150% of the default value.

Line 94: there should be a reference for SCAM5.

We will add reference [11] in the revised manuscript to refine the introduction of SCAM5.

Line 102: all sites belong to ARM.

We will correct this issue in the revised manuscript. We will also carry out a more detailed study later.

Line 108: "in the code" change to "in the model"

We will correct it in the revised manuscript.

Line 116: "is an important issue" change to "are important issues"

We will correct it in the revised manuscript.

Line 130: It is only suit for SCAM. For GCM, it is impossible.

Indeed, as you say, it is not practical to run large batches of GCM jobs even on HPC. We will refine this description in the revised manuscript.

Line 165: please explain the "second-order sensitivity"

It refers to the mutual influence between two parameters. We will refine this exposition in the revised manuscript.

Table 4: the reference of Sobol is wrong.

We will correct this reference [9] in the revised manuscript.

Line 174: please consider the correct position of this sentence.

We will correct it in the revised manuscript.

Line 193: It is not clear for "not a direct evaluation."

We will add some necessary information and details in our revised manuscript. What we want to express here is the following: using the above methods, the set of M parameters that most influence the result may be difficult to obtain directly. The method proposed in this paper

just fills this gap.

Line 195: add "have" before "its"

We will correct it in the revised manuscript.

Line 215: how do you get the number of multiple thousand?

Combining Equation (1) with the practical problem studied in this paper, we can see that when $C = 5, p = 3$ and $L = 10$, 5,000 simulations are needed even if only one parameter combination is considered. This will be astronomical when more combinations are considered. We will also refine the above in the revised manuscript.

**References**

[1]  J. Guo, Y. Xu, H. Fu, W. Xue, L. Wang, L. Gan, X. Wu, L. Hu, G. Xu, and X. Che, "A learning-based method for efficient large-scale sensitivity analysis and tuning of single column atmosphere model (scam)," *Geoscientific Model Development Discussions*, vol. 2022, pp. 1–28, 2022. DOI: 10.5194/gmd-2022-264. [Online]. Available: https://gmd.copernicus.org/preprints/gmd-2022-264/.

[2]  K. Cheng, Z. Lu, C. Ling, and S. Zhou, "Surrogate-assisted global sensitivity analysis: An overview," en, *Structural and Multidisciplinary Optimization*, vol. 61, no. 3, pp. 1187–1213, Mar. 2020, ISSN: 1615-147X, 1615-1488. DOI: 10.1007/s00158-019-02413-5. [Online]. Available: http://link.springer.com/10.1007/s00158-019-02413-5 (visited on 04/03/2023).

[3]  B. Yang, Y. Qian, G. Lin, L. R. Leung, P. J. Rasch, G. J. Zhang, S. A. Mcfarlane, C. Zhao, Y. Zhang, and H. Wang, "Uncertainty quantification and parameter tuning in the cam5 zhang-mcfarlane convection scheme and impact of improved convection on the global circulation and climate," *Journal of Geophysical Research Atmospheres*, vol. 118, no. 2, pp. 395–415, 2013.

[4]  L. Zou, Y. Qian, T. Zhou, and B. Yang, "Parameter tuning and calibration of regcm3 with mit–emanuel cumulus parameterization scheme over cordex east asia domain," *Journal of Climate*, vol. 27, no. 20, pp. 7687–7701, 2014. DOI: https://doi.org/10.1175/JCLI-D-14-00229.1. [Online]. Available: https://journals.ametsoc.org/view/journals/clim/27/20/jcli-d-14-00229.1.xml.

[5]  W. XingFen, Y. Xiangbin, and M. Yangchun, "Research on User Consumption Behavior Prediction Based on Improved XGBoost Algorithm," en, in *2018 IEEE International Conference on Big Data (Big Data)*, Seattle, WA, USA: IEEE, Dec. 2018, pp. 4169–4175, ISBN: 978-1-5386-5035-6. DOI: 10.1109/BigData.2018.8622235. [Online]. Available: https://ieeexplore.ieee.org/document/8622235/ (visited on 04/03/2023).

[6]  L. Shi, C. Copot, and S. Vanlanduit, "Evaluating Dropout Placements in Bayesian Regression Resnet," en, *Journal of Artificial Intelligence and Soft Computing Research*, vol. 12, no. 1, pp. 61–73, Jan. 2022, ISSN: 2449-6499. DOI: 10.2478/jaiscr-2022-0005. [Online]. Available: `https://www.sciendo.com/article/10.2478/jaiscr-2022-0005` (visited on 04/03/2023).

[7]  S. Mirjalili and A. Lewis, "The Whale Optimization Algorithm," *Advances in Engineering Software*, vol. 95, pp. 51–67, 2016, ISSN: 18735339. DOI: 10.1016/j.advengsoft.2016.01.008.

[8]  (). "Model grids – cime master documentation," [Online]. Available: `http://esmci.github.io/cime/versions/master/html/users%5C_guide/grids.html`.

[9]  I. Sobol, "Global sensitivity indices for nonlinear mathematical models and their monte carlo estimates," *Mathematics and Computers in Simulation*, vol. 55, no. 1, pp. 271–280, 2001, The Second IMACS Seminar on Monte Carlo Methods, ISSN: 0378-4754. DOI: `https://doi.org/10.1016/S0378-4754(00)00270-6`.

[10]  Y. Qian, H. Yan, Z. Hou, G. Johannesson, S. Klein, D. Lucas, R. Neale, P. Rasch, L. Swiler, and J. Tannahill, "Parametric sensitivity analysis of precipitation at global and local scales in the community atmosphere model cam5," *Journal of Advances in Modeling Earth Systems*, vol. 7, no. 2, 2015.

[11]  P. A. Bogenschutz, A. Gettelman, H. Morrison, V. E. Larson, D. P. Schanen, N. R. Meyer, and C. Craig, "Unified parameterization of the planetary boundary layer and shallow convection with a higher-order turbulence closure in the community atmosphere model: Single-column experiments," *Geoscientific Model Development*, vol. 5, no. 6, pp. 1407–1423, 2012. DOI: 10.5194/gmd-5-1407-2012. [Online]. Available: `https://gmd.copernicus.org/articles/5/1407/2012/`.

---

## Author Response (AR1)

**Summary of changes, GMD-2022-264**

**Jiaxu Guo on behalf of all authors**

**May 16, 2023**

Dear Editor and Referees:

Thank you very much for your patient and detailed comments on our work. These valuable comments are very helpful for us to improve this paper. We have answered each of comments and made appropriate corrections in the revised version of our manuscript.

In this attachment, we will summarize the main modification changes in this paper.

(1) In training the surrogate model, we introduced five different regression methods to train separately to compare their fitting effects. Based on this, we rerun all the experiments. The corresponding figures of experimental results have been also replotted. We also used more quantitative data to show the effectiveness and reliability of our method. The newly trained surrogate models have a better fit and less inconsistency with the actual model.

(2) When training the surrogate model, we also included the samples generated by the Morris sampling method in the dataset, which doubles the size of the original dataset and further improves the fit. RMSE was used as the loss function instead in all experiments.

(3) For multi-site problems, we also carried out optimization for each of the four scenarios. Separately, they are: for ARM95 and ARM97 cases, for three tropical convection cases, for TOGAII/TWP06 (they're close in location) and all five cases, as shown in Figure 14.

(4) In Section 1, we rearranged the word order of the narrative, and elaborate according to the status of pattern research, existing parameter analysis and regulation methods, the challenges faced, and the contributions of this paper. At the same time, we also expanded the scope of literature survey and introduced more literature to further introduce the current research status.

(5) We further clarified the running time of SCAM and try to avoid any ambiguity. Meanwhile, some statements in the manuscript have been revised to make them more transparent.

*In the following, we will reply to comments one by one.*

**Replies to Referee #1, GMD-2022-264**

Jiaxu Guo on behalf of all authors

May 16, 2023

Thank you very much for your patient and detailed comments on our work[1]. These valuable comments are very helpful for us to improve this paper. After carefully reading all the questions, we have answered each of them and will make appropriate corrections in the revised version of our manuscript.

In this attachment, the red paragraphs represent your comments, and the black paragraphs below are our corresponding replies.

**1 Replies to major comments**

Lines 217-218: "Even for the SCAM model, which takes more than one hour to finish a run, such a combined cost becomes impractical, for combined studies of multiple parameters...." - How many nodes are being used for your SCAM baseline case? Properly tuned, SCAM should run in a matter of minutes and should certainly not take more than an hour to run any of these IOP cases.

We apologize for the unclear statements and the confusion that we may have caused. We totally agree that a single SCAM job would take only a few minutes. The platform we use is based on Sunway Processors. For the 5 cases covered in our article, the shortest one took about 10 minutes and the longest one was no more than 20 minutes. Each Sunway processor consists of 4 Core-Groups (CG). Each CG can support a single MPI process. We normally run one SCAM on one CG (note that the Sunway processor is running at a frequency that is roughly one third of an Intel or AMD processor).

The case we refer to here is a workflow of parameter sensitivity analysis and tuning, which consists of 768 SCAM jobs to run. The one hour mentioned here is the time it takes to assign 768 jobs to the job queue, and to collect the results after all jobs are finished. We do experiments when there are enough resources for multiple times, to compute the time on average.

We remove the ambiguous statements and focus on clarifying the problem in terms of the number of computations. The revised text is on Lines 229-241.

Line 231: "With the compute time reduced to less than five percent of the original model" - is this based on a SCAM case taking >1hr to run for an IOP? I would question that baseline performance.

As explained above, we apologize for the one hour confusion we have made. When we say greater than 1 hour, we do not mean that a case run with a single IOP would take 1 hour, but that a run with 768 different cases would take 1 hour. We record the time in a normal supercomputing environment, so as to demonstrate the related time overhead for scheduling, running the job, as well as collecting the results. The revised text is shown on Lines 240-241.

Line 72-73: "Improved balance between cost and accuracy" - The accuracy of SCAM is not in question, so I don't see that the balance between cost and accuracy has significantly changed. The cost itself is also not a prohibiting factor in running SCAM, as it's already pretty efficient. The issue with assessing a larger number of parameters often lies in the tractability of analysis, not computational cost.

We agree that our work is primarily about making a large-scale parametric analyses possible, which is what you mean by "tractability". Although SCAM itself has a very short run time, the results of the analysis can be obtained faster by further reducing the resource overhead of the experiment in a large-scale experiment. Again, this is an effort to improve the tractability of the analysis. We have adjusted the description of this key issue in the revised manuscript, as detailed in Line 79-86.

Lines 164-168: "For example, the Morris sampling (Morris, 1991)..." - This statement feels like it's nearly identical to the introduction (lines 36-42). Overall, section 3.1 seems to rehash the background given in the introduction rather than describing the specific SCAM cases run.

We have removed this part of the redundant description, as detailed in the revised Section 3.1.

Line 183: "At the training stage, we reuse the 768 sets of different parameters and their corresponding total precipitation output ..." - What are the 768 different sets ofparameters? How does this arise from the 11 parameters tested? Is it 768 sets per IOP? More detail is needed on the SCAM runs in order to provide context for this section.

This number is based on the number of samples from the MOAT and Saltelli sampling methods. In order to reconcile the two sampling methods used in the text, a number was chosen that is large enough and that matches the relationship between the number of samples generated by both methods. 768 is the number of samples per IOP case. We have added this in a revised version of the manuscript, as detailed in Section 3.1.

Section 4.1: I'm not sure how relevant this is to the results shown below; ideally, regardless of where SCAM is run, the results should be identical. Would suggest removing this full section on use of Sunway TaihuLight, particularly as it's mentioned already previously.

Here we are mainly describing the environment in which the experiments in this paper were run. This platform is really not strongly correlated with the implementation of the experiments, and their results. We have removed this whole subsection in the revised version of the

manuscript, as detailed in Section 4.

Section 4.6: It's unclear to me how this differs from section 4.5; please elaborate further in the main text. Did section 4.5 not use the same range of values for all cases?

Section 4.5 mainly looks at a more efficient optimization process and the final result of the individual case optimizations. Section 4.6 focused more on the distribution of results across cases in the same parameter space. We show this rule through 3D figures, which is more beneficial for us to analyze the similarities and differences between these cases. Meanwhile, in Section 4.6, we use the optimal solution of each case as a vector, combined with the Pearson correlation coefficient method to calculate the similarity between the individual cases. By using the coefficient as a metric, it is possible to get a more intuitive view of the relationship between these cases. The range of values taken in the Section 4.6 test is the same for all cases. We redo the experiments again, and we also added the experimental exploration of all cases, and cases of the same type using the same parameters. The content of this subsection has been comprehensively revised, including Figure 13-15.

Lines 410-411: "This is also confirmed by the experiments in the next subsection.", should there be a section 4.7 here that's being referenced?

We are sorry for the misunderstanding. When a draft of this article was written, there was a section 4.7. It was removed in a later edit, but we did not correct the narrative in time. We have corrected the error here.

"In addition, another scenario was considered: where the parameter configuration is the same among cases": This paragraph seemed to suggest that another experiment was conducted in which the parameters are all set to the same value in order to optimize performance across all five cases; I don't see results from that experiment though.

This is a copy editing error in the collaboration and should read as follows, "After the scenario where the parameter configuration is the same among cases has been considered, the closeness between the cases could be analyzed." We have also added experiments using the same parameter values for all cases. This is shown in Figure 14. From this we can see the distribution of the output in the parameter space for all cases or cases belonging to the same type, when they take the same parameter values.

Sections 4.5-4.6: I don't see an explicit discussion of cases where the optimal parameter values are carried out in SCAM rather than the NN surrogate model to confirm the results. The authors should clarify which experiments are conducted in SCAM vs. the surrogate, and consider a more explicit discussion of differences that arise when the optimal tuned parameters are used in full, online SCAM runs.

The results in our paper are derived from experiments using SCAM to confirm optimal parameter values. In fact, this confirmation is already included in the parameter tuning workflow proposed in this paper. The experiments in Section 4.5 were performed by finding the optimal solutions using the surrogate model and testing them in the complete SCAM. The resulting error pairs between the surrogate model and SCAM are shown in Table 8 in the revised version. Experiments in Section 4.6 were performed in surrogate mode due to their heavy computational burden. We have revised the above description.

*We have also carefully read each of the detailed descriptions you mentioned in relation to the text and captions, including the units consistently on the y-axis.*

Figure 5: Could you define 'maximum fluctuation' more precisely? Is this the difference between lowest and highest PRECT value, and is the value of PRECT output hourly/daily/etc?

It is the difference between lowest and highest value of PRECT output. The output value here refers to the average value throughout the simulation, for each case. Due to the adjustment of the graph order, we have revised on the caption of Figure 6.

Line 224: "However, the complexity of the calculation increases exponentially during the test, as shown in Table 4." I'm not sure Table 4 is the right reference here; it shows the SA methods used, but does not seem to indicate complexity or cost of these tests.

We are sorry that the mistaken reference has caused confusion. The reference here should be to Equation (4) but not to Table 4 in the original manuscript. We will add more detail here in the revised version of our manuscript. What we are trying to convey here is that, as can be seen by Equation (4), when calculating the effect of the combined parameters on the results, the $N_{MPP}$ increases exponentially as $p$ increases due to the position of $p$ in the exponential.

Figure 9: Which SA method is being used in this figure? Units on the y-axis would also be helpful.

The single parameter perturbation method is used here, i.e. keeping the other parameter values constant at their default values and tuning only the value of one parameter linearly. To illustrate the problem more clearly, we have also added the units of the y-axis.

Figure 12: Please add units to the y-axis.

We have added units $mm/day$ to the y-axis in Figure 12 in the next manuscript submission.

Figure 13: A more detailed description of this figure in the text would be helpful. It's unclear what improvement is being plotted, and what the 'original' case in blue is given that the 'enhancement of effects' on the y-axis is due to the addition of NN and/or grid searching. Is the blue not the control case then?

We redo the experiment and draw the figure. In the revised version, it is Figure 11. CTL refers to the control experiment using the default value. Baseline refers to an experiment using only one optimization method. Optimized refers to the optimization experiment that combines the surrogate model and grid search.

Figure 14: Similarly, more detail would be useful. What units are the overhead in computing given

in? Is it obvious that the overhead should be the same for the Original and NN cases?

The units used are the total computational hours it takes to perform a simulation. As NN's improved approach relative to *Original* is mainly reflected in the sensitivity analysis, and this part of the experiment does not involve running more SCAM instances, the change in computing time is not reflected significantly, and therefore the difference in computing time is less reflected.

Lines 421-422: "we can see that the parameter values taken between the two cases of land convection are positively correlated in the same parameter space" it's surprising to me that the correlation is equal to 1 between ARM95 and ARM97 despite apparent differences between them in Fig 15. Could the authors elaborate on why this occurs?

This might be related to the pre-processing that the vectors undergo before they are involved in the calculation. Given that we have re-trained the model, new results will also be presented in our revised version, as shown in Figure 15.

**2   Replies to specific comments**

Lines 1-2: "The Single Column Atmospheric Model (SCAM) is an essential tool for analyzing and improving the physics schemes of CAM." Please specify that CAM in this case is the Community Atmosphere Model.

We have added this specification in a revised version of the manuscript.

Lines 6-8: "By reusing the 3,840 instances with the variation of 11 parameters..." Suggest avoiding using specific numbers like this in the abstract; without an explanation, it is unclear what "the 3,840 instances" are. Either add clarification/context, or remove the specific number of instances.

We have removed the specific number of instances.

Line 15: Should this read "the effects of global climate change"?

We have corrected this in the revised version of the manuscript according to your comment.

Line 18: This citation of CAM is outdated, please point to the scientific articles describing CAM instead. The title of this particular citation in the references points to CAM3 and the link itself is broken.

We will refine the citations to the references in the revised manuscript and ensure that the links are all accessible. References [2] and [3] have been added to make the descriptions more precise.

Line 18: "Of these components, the Community Atmosphere Model (CAM) (UCAR., 2020), is the one with the most complexity." It's hard to say that more model complexity is contained in one model

component than another; could the authors clarify/justify what's intended by this statement?

The intention here is to illustrate the complexity of CAM and thus set the scene for the introduction of SCAM below. We have revised these descriptions as: "The use of SCAM for large-scale experiments is more practicable due to its advantage of lower requirements for computing resources."

Line 20: "Participated in continuous numerical integration," - Consider rephrasing for clarity; do the authors mean that in coupled climate simulations, this is a source of uncertainty?

The main purpose here is to highlight the complexity of GCM and thus illustrate where the advantages of SCAM lie. We will rephrase these descriptions in the revised version as :"The use of SCAM for large-scale experiments is more practicable due to its advantage of lower requirements for computing resources."

Line 22: "However, as a general circulation model (GCM), CAM takes a long time and a large amount of resource to run..." Given that ESM is already defined above, the authors should continue to use that notation rather than also defining GCM (unless a distinction is intended, which could be elaborated on).

We have corrected this issue in the revised version. All notations for the same definition will be unified.

Line 24: "good alternative model" – rephrase for clarity. What's meant by 'good' here – cheaper, more efficient, etc?

Cheaper computational overhead and higher efficiency are both advantages. We have rephrased it to make this more clearly expressed.

Line 25: What is meant by SCAM only needs "one process"?

Since SCAM is a small and fast model, it runs only on one processor. In one simulation of SCAM, only one process is required for each run of one case to complete the computation.

Lines 28-29: "Sensitivity analysis (SA) is a method for investigating how uncertainty in the model output is assigned to the different sources of uncertainty in the model input factors, and the participants (Saltelli et al., 2010)." It's not clear what the participants are; please clarify.

The term 'participants' refers to the independent variables in the problem under study. We have rephrased it in the revised version of the manuscript, as detailed in Line 28-31.

Line 41: "quasi-random sequence by Sobol (Sobol', 1967) and other researchers)," please specify the other researchers who have established the method so that it can be easily referenced by readers. It may also be useful to briefly explain what the "low-discrepancy quasi-random sequence" is if relevant to the study.

Thanks for your suggestion, we have revised the description according to the relevance.

Line 48: "After we identify the important tuning targets in the SA stage" – how are those targets usually defined? This hasn't been explained in the previous paragraph, only that there are different methods for sampling parameters. Please briefly note how that translates to identification of targets (even a sentence should suffice).

Targets, in this context, refer to the parameters (or a combination of them) that are more sensitive to the results obtained during the SA phase of the analysis. Having identified these targets, readers can then know which parameters to tune to have a greater impact on the results, thus making it easier to get better tuning results. We have refined these descriptions as "After we have determined the combination of parameters to be tuned".

Line 70: "By reusing the 3,840 instances with variations of 11 parameters" – there is no indication of where these numbers are coming from or what they refer to. This should be introduced in the methods section, so wait until that point to elaborate on specifics like this.

We have corrected this in the revised manuscript.

Lines 75-82: please condense into a paragraph rather than bulleted list.

We have condensed this part of the text in a revised manuscript.

Lines 77-79: "By reusing the 3,840 sampling instances..." - Again, where does the number of instances come from? And when the authors say the model achieves good accuracy, which variables are they referring to (i.e., the "error within 10%" is an error across which variables?)

For the sample size, we will go into more detail as to why this sample number was selected. The latter achievement refers to the improvement in the model fit to total precipitation, i.e. the variable is PRECT.

Lines 89-91: "case-specific tuned parameters would further reduce the precipitation error by 15% when compared to a set of unified tuned parameters, and suggest a potential improvement from location-wise parameter tuning in the future." I did not see the discussion of a case where all cases are combined to find the optimum parameter values. Please elaborate further on that in the results section to support this.

We have included a corresponding discussion in the revised Section 4.6, as shown in Figure 14.

Lines 98-100: Is it wise to tune for just a single variable (time-mean PRECT)? Realistically, when assessing model performance and tuning accordingly, a number of performance metrics need to be accounted for beyond the mean of one variable. Could the authors elaborate on the validity of selecting just one, perhaps?

PRECT is an output variable included in the IOP files of all five cases covered in this manuscript, and its use as an object of study facilitates cross-sectional comparisons between the cases and the analysis of their relationships.

Lines 111-114: "In addition, the programs running on Sunway TaihuLight needed to be recompiled due to the adoption of a different archiecture." - Shouldn't the model be recompiled at build time? Is this a unique addition to the model, that enables compilation with a non-supported compiler? Is the source code available and going to be included in CESM?

As we chose Sunway TaihuLight as our experimental platform, we were able to make the code work on this system by attempting to port and compile it. By modifying the source code of the model, the parameters involved in the paper can be tuned via the namelist input file. As a result, there is no longer any need to recompile each time one experiment is carried out, which also makes it much more efficient. The code is currently available.

Lines 184-185: "We set the learning rate..." – could the authors elaborate on if this is the most suitable choice of learning rate/batch size? Were other values tested?

The values chosen are empirical parameter values of learning rate and batch size that we commonly use for neural network model training. This paper focuses on the feasibility of using neural network methods for large-scale parameter analysis and tuning, and therefore the empirical values were chosen for testing. This set of parameter values is a selection of the better performing values after testing several sets of values. We have conducted an ablation experiment for learning rate and batch size and added details in a revised manuscript.

Figure 5: There's a relatively wide variability in this across cases, with ARM97 and TOGAII being the least sensitive and GATEIII being very sensitive to the number of parameters to be tuned. Worth elaborating on?

What this figure reflects is indeed of some interest. The original meaning of the figure was how much output fluctuation (in terms of average precipitation during the simulation) could be produced for each case when tuning one to four parameters, respectively. The figure does show that for the ARM97 and TOGAII, their precipitation response for four parameters is even smaller than the response of the GATEIII for tuning one parameter. An important reason for this is that these two cases themselves have smaller precipitation values than GATEIII, whereas the figure uses the absolute values of precipitation.

Lines 225-226: "...although the effect of tuning four parameters was better than tuning three parameters, the advantages of the surrogate model in the parameter tuning process could not be exploited at this point." I'm unclear why "the advantages of the surrogate model in the parameter tuning process could not be exploited" when using 4 rather than 3 parameters.

Testing combinations of parameters on surrogate models also needs computational resources. As the size of the test increases exponentially with the number of parameters to be tuned, the computational time will no longer be negligible when tuning four parameters. The results also show that the improvement of maximum tuning effect that can be achieved by tuning four parameters is limited compared to tuning three parameters. Therefore, considering both the computational overhead and the tuning effect, we chose to tune three parameters for the experiments in this paper.

 "combinations of three parameters lead to the most significance in output" – is this meant to imply the three parameters that drive the most significant change in PRECT, or the most significant improvement (assuming those are different, they could be the same, but a big change does not necessarily lead to improvement).

Theoretically, the more parameters that can be tuned in one experiment, the greater the variation in the results that can be brought about. The reason for choosing to tune three parameters here is also to achieve a balance between computational overhead and tuning effect. This allows the sensitive parameters to be tuned while avoiding the non-critical parameters consuming computational resources.

Lines 255-257: For complete reference, please also explain *epsilon* and $p$ as they are used in Algorithm 1.

We have explained in detail the meaning of these two variables and the role they play in this algorithm. $\epsilon$ is the threshold at which the results converge and $p$ is the total number of parameters to be adjusted.

Line 260: "Meaning average error" – is this meant to be the mean absolute error? Or is this something else?

It refers to MAPE (Mean Absolute Percentage Error). In addition, after careful discussion, we instead use RMSE (Root Mean Square Error) as a measure of error.

Lines 290-292: "We use a total of 7,680 samples, with 1,536 samples for each of the five SCAM case." - Please elaborate on the reason for choosing this number of samples – how many values per parameter are enabled by this choice? Is there a clear reason for running 1,536 samples per IOP? This also seems like a detail that should be included in the methods rather than the results.

As with the answer to the question on line 70, since this research involves two methods of generating sample sequences, MOAT and Saltelli, each of which has a different formula for generating the number of samples. Their respective base numbers are different and the choice to generate 1536 samples per case (768 for each of the two methods) was also made in view of the fact that 768 is a appropriate sample size that can be generated by both methods. We have made this point in addition in Section 3.1.

Lines 298-299: "...although the medium point varies from 5 to 12 mm per day, demonstrating clearly different climate patterns." - How much of this is due to differences in the length of the IOP (perhaps one captured more dry days than another, for example), or an IOP designed to capture shallow vs. deep convection? This may not necessarily be indicative of obviously varying climate patterns.

Indeed, the five IOPs involved in this study differed in their location, length of capture and time of day. For example, GATEIII has a longer capture time, while TOGAII has a relatively shorter capture time. We have reorganized the language of Section 4.1 (originally Section 4.2).

Lines 318-319: "The reason for this difference probably comes from a different time of year and the forcing field simulated in these two cases." It would be good to see a more confident assertion here.

How different is the time of year assessed, is it substantial enough to cause such a change in sensitivity? How different is the forcing field (and does this hypothesize that it's the large scale T or Q convergence that's responsible)? Is there a difference in the type of convection that occurs as well?

We are sorry for the mistake. This problem comes from the error generated in our previously trained model. This problem has been resolved after retraining the surrogate model. ARM95 and ARM97 show consistency in all aspects, which is also in line with our expectations.

Line 322-323: "Instead of calling SA methods directly, we use combinatorial analysis of the magnitude of change to determine the effect that these parameter combinations have on the model output." I'm not sure on what this means, please rephrase for clarity?

This refers to the invocation of the method proposed in this paper for combined parameter analysis, rather than the existing fixed method. We have reformulated the expression.

Lines 330-332: "For example, increasing tau tends to increase total precipitation in GATEIII, while in the other cases it brings the opposite result." - Is there something special about that case that causes the unique signal?

This case does show a unique signal of interest at this point, which may be related to the nature of the GATEIII case itself. A similar conclusion was reached when we explored the parameters with the help of the surrogate model. Different from other cases, this case is a ocean case and is located in the Atlantic Ocean.

Lines 353-353: "The impact of such differences is even multiplied" - unclear what is meant by this statement; what is being multiplied here?

In this sentence, we are trying to express the level of impact by using the word *multiplied*. We have rephrased it in the revised version.

Lines 366-368: "It is easy to see that in the control experiment there were several spikes where the simulated output was significantly higher than the observed values, as was the case in the first four cases. After tuning, these spikes are significantly weakened and the output is much closer to the observed values." It looks like this is really only an issue in the land-based ARM cases; is that true? It looks like the tuning is still unable to match some of the largest rain rates in GATEIII especially but also TOGAII – is there a reason for that?

This phenomenon was indeed more pronounced in the two land-based ARM cases. On average, our tuning is also effective for the three tropical convection cases. The inability of the tuning to match the total precipitation for GATEIII does exist, as can also be seen from the analysis of the sampling results in Figure 7(c). The failure to cover the total precipitation that matches the observations in the sampling results means that the likelihood of finding the optimal solution through tuning is small. In contrast, this possibility is still present in TOGAII. The existence of this result is justified by the fact that there is a certain margin of error in the simulation of the model itself.

Lines 368-369: "This demonstrates the significance of the parameter tuning provided by the workflow

for model." Could the authors be more quantitative here? How much is the bias reduced by, for example?

The main purpose here is to highlight the important role that scientific workflow plays in the work of this paper. The methods we present in this paper are organized in the form of a workflow, and the entire reconciliation process is done coherently. We have added quantitative analysis in the revised version.

Lines 399-400: "It is easy to see that the two land convection regions are closer and, accordingly, the three tropical convection aggregation regions are also closer." It looks like the land cases might be fairly different, particularly in terms of optimal pz4 value. A table would make it easier to compare the 'optimal' values, even if they're given by a range of what's marked in red in Figure 15.

We agree that the specific values should be more intuitive and readable for the reader. We have added Table 9 in the revised edition.

Lines 402-403: "the distribution of the better value points is different for different parameters even for the same parameter space." Should this read that the better values are different for different cases within the same parameter space?

We will correct this in the revised version so that it will be easier for the reader to understand.

Line 404: "In the other three cases, smaller values of tau lead to better performance." It looks like the optimal value of pz4 occurs at/near the minimum range for both GATEIII and TWP06 - have the authors tested expanding the lower limit of this variable further to see if this is the optimum value or if it's being cut off?

Indeed, as can be seen from the figure, the optimized values obtained for both cases do lie at the boundary. Thus the possibility exists that there may be better values outside the bound than inside that bound. However, the values of the parameters cannot be infinitely large or small, and we should also take into account their physical meaning.

Lines 405-406: "On the other hand, it also shows that there are differences in the distribution of parameters that make the results perform better in different types of cases." Rephrase? This sounds like it's saying the same thing as the sentence before it.

Indeed, as you say, we will use more concise phrases in the revised version: "This reflects the fact that it may be useful and necessary to adopt different parameter configurations for different cases or regions."

Lines 407-408: "It can be got that the optimal value points for the two land convection cases are close, while the points for the three tropical convection cases are even closer." While the GATEIII and TWP06 cases are very similar, the difference in the TOGA case challenges the notion that tropical convection cases are closer than what we see in the ARM case. Over land, it also looks like the pz4 optimal values are actually rather different; a table would make this argument more convincing and easier to see, potentially. Or may point to the need for a more nuanced statement.

Indeed, as you have said, further details could be given in terms of specific optimized values. A more detailed elaboration would make our experimental results and views more convincing. At the same time, we conducted another complementary experiment, which was to try to introduce different methods to train the surrogate models for SCAM cases. Through our research, we learned that both XGBoost[4] and ResNet[5] can be used to perform regression tasks and train surrogate models. Here, we will compare the effectiveness of several methods such as LR (Linear Regression), RF (Random Forest), MLP (Multi-Layer Perceptron), XGBoost and ResNet for training surrogate models. The results are shown in Table 6. The RMSE was used to measure the error generated during training. Based on these results, we used ResNet to retrain the surrogate models, and when we used these later trained models to perform a grid search in the same parameter space, as can be seen from Figure 14. This is also consistent with the above distribution of the two cases in terms of position. This is due to errors in the previous training models, and we would appreciate your prompt correction.

Lines 416-417: "The difference between the two cases lies mainly in the time, which therefore reflects that there is also a difference in SCAM's simulation performance for different times." I don't see a time-based sensitivity analysis in this; could the authors clarify/elaborate? Is this a seasonal difference? Were other alternative hypotheses explored to explain the difference in ARM95 and ARM97 (different synoptic conditions, etc)?

The term 'time' here refers not to a time-based sensitivity analysis, but to the historical time simulated by these SCAM cases. At the same time, we conducted another complementary experiment, which was to try to introduce different methods to train the surrogate models for SCAM cases. Through our research, we learned that both XGBoost[4] and ResNet[5] can be used to perform regression tasks and train surrogate models. Here, we will compare the effectiveness of several methods such as LR, RF, MLP, XGBoost and ResNet for training surrogate models. The results are shown in Table 6. The RMSE was used to measure the error generated during training. The best performers are shown in bold. Based on these results, we used ResNet to retrain the surrogate models, and when we used these later trained models to perform a grid search in the same parameter space, we found that their distributions were very similar, as can be seen from Figure 14. This is due to errors in the previous training models, and we would appreciate your prompt correction.

Line 419: "the optimal parameter values for each case can be represented by a vector" - I'm not entirely clear on what this vector would look like; it might help the reader to plot said vector on Fig 15.

The term *vector* here refers to the optimized solution of a set of parameters as a vector. This is because the computation of the Pearson correlation coefficients is done on the basis of vectors. We have refined this description in Section 3.5.

**References**

[1]   J. Guo, Y. Xu, H. Fu, W. Xue, L. Wang, L. Gan, X. Wu, L. Hu, G. Xu, and X. Che, "A learning-based method for efficient large-scale sensitivity analysis and tuning of single column atmosphere model (scam)," *Geoscientific Model Development Discussions*, vol. 2022, pp. 1–28, 2022. DOI: 10.5194/gmd-2022-264. [Online]. Available: https://gmd.copernicus.org/preprints/gmd-2022-264/.

[2]   J. T. Bacmeister, M. F. Wehner, R. B. Neale, A. Gettelman, C. Hannay, P. H. Lauritzen, J. M. Caron, and J. E. Truesdale, "Exploratory high-resolution climate simulations using the community atmosphere model (cam)," *Journal of Climate*, vol. 27, no. 9, pp. 3073–3099, 2014, Cited by: 163; All Open Access, Green Open Access. DOI: 10.1175/JCLI-D-13-00387.1.

[3]   J. M. Dennis, J. Edwards, K. J. Evans, O. Guba, P. H. Lauritzen, A. A. Mirin, A. St-Cyr, M. A. Taylor, and P. H. Worley, "Cam-se: A scalable spectral element dynamical core for the community atmosphere model," *International Journal of High Performance Computing Applications*, vol. 26, no. 1, pp. 74–89, 2012, Cited by: 258. DOI: 10.1177/1094342011428142.

[4]   W. XingFen, Y. Xiangbin, and M. Yangchun, "Research on User Consumption Behavior Prediction Based on Improved XGBoost Algorithm," en, in *2018 IEEE International Conference on Big Data (Big Data)*, Seattle, WA, USA: IEEE, Dec. 2018, pp. 4169–4175, ISBN: 978-1-5386-5035-6. DOI: 10.1109/BigData.2018.8622235. [Online]. Available: https://ieeexplore.ieee.org/document/8622235/ (visited on 04/03/2023).

[5]   L. Shi, C. Copot, and S. Vanlanduit, "Evaluating Dropout Placements in Bayesian Regression Resnet," en, *Journal of Artificial Intelligence and Soft Computing Research*, vol. 12, no. 1, pp. 61–73, Jan. 2022, ISSN: 2449-6499. DOI: 10.2478/jaiscr-2022-0005. [Online]. Available: https://www.sciendo.com/article/10.2478/jaiscr-2022-0005 (visited on 04/03/2023).

**Replies to Referee #2, GMD-2022-264**

Jiaxu Guo on behalf of all authors

May 16, 2023

Thank you very much for your patient and detailed comments on our work[1]. These valuable comments are very helpful for us to improve this paper. After carefully reading all the questions, we have answered each of them and will make appropriate corrections in the revised version of our manuscript.

In this attachment, the red paragraphs represent your refree comments, and the black paragraphs below are our corresponding replies.

**1 Replies to major issues**

In this paper, although the authors evaluate the accuracy of the NN model in terms of precipitation, it probably exists the inconsistent between ML and real model, which don't be highlighted in this paper.

We agree that it may be difficult for a surrogate model to be fully consistent with a real model. We use the RMSE in training as a loss function to verify the correctness of the method, that is, whether the parameter tuning of SCAM can be accelerated by training a surrogate model, and to compare different regression methods. We will add more descriptions about the inconsistency to the manuscript from two perspectives: From a model training perspective, the value of loss function indicates the error in the training. However, the process by which we train the model is also the process by which the error is gradually reduced. From a practical perspective, when we use the surrogate model for optimization, the solutions obtained are also validated in the original SCAM case to ensure as much consistency as possible, as shown in Table 6 and 8.

The authors believe that due to the high computational cost of the GCM, the SCAM can be the alternative model for parameter SA and tuning. In reality, the optimal parameters tuned in SCAM could not be suitable for GCM, due to the global regions and more complex large-scale circulation.

We agree that the solution set obtained by parameter tuning on SCAM is not directly applicable to GCM. However, through our attempts at parameter tuning on SCAM cases located in different regions, we can find commonalities and patterns in the parameter response of these

cases. We believe such an idea can be applied to the parameterization scheme of the GCM. Our exploration of parameterization solutions using SCAM is mainly from a methodological and ideological point of view. That is, SCAM is a simpler and less costly way to perform numerical simulations. The training of a surrogate model for SCAM is a further extension of this idea.

Overall this manuscript, the organization and writing are not clear and should be well structured. There are a very large number of language errors. The English writing should be greatly improved.

We will reorganize the structure of the article in a revised version to rationalize the logic. We will also work on improving the English writing style to make it more fluent.

The authors separately tune the parameters in SCAM for each site and get the different sensitive parameters and different optimal values. It is difficult to transfer this information to GCM. If the authors can do the multi-objective tuning for these sites with the same parameters, it could be helpful for global model tuning because these SCAM sites indeed represent the different regimes.

We agree that a combined optimization that tries to minimize the differences against observations across all five cases would be more meaningful. Such tests and results were included in a previous version of the draft. However, we removed the results at certain stage for a more focused description in the result section.

As suggested by the reviewers, we have carried out a careful analysis and done corresponding experiments. After retraining the surrogate model for each of the five cases by using ResNet, we also carried out multi-objective optimization for each of the four scenarios. Separately, they are: for ARM95 and ARM97 cases, for three tropical convection cases, for TOGAII/TWP06 (they're close in location) and all five cases, as shown in Figure 14.

Combined with results shown in Figure 13 of our grid search for the five cases individually, it can be seen that cases located at the same or similar locations really have similar distributions of precipitation output in response to parameters. Therefore, this kind of joint optimization based on the multi-objective idea is of general interest.

For the workflow, there should be a "metrics" component because it is very important for tuning. No matter SCAM or GCM, the tuning metrics could be the cost function between model simulations and observations. The different designs could affect the optimization. In terms of the metrics, it could consider the 1) different statistic errors between simulation and observation, such as RMSE, performance score like Yang et al. (2013), 2) one objective or multiple objectives, and how to deal with the multiple objectives.

I agree that "metrics" are important factors to consider in the workflow. We will use RMSE as the metric of error in the revised version.

$$RMSE = \sqrt{\frac{1}{N} \sum_{i=1}^{N} (y_i - \hat{y}_i)^2} \tag{1}$$

After using RMSE as the metric for training the model and parameter optimization, the corresponding results were recalculated. In particular, the losses during the training of the model

are shown in Table 6. The 3D parameter space responses for the five cases themselves and for the four multi-objective joint conditioning scenarios are shown in Figure 13 to 14.

Regarding multi-objective tuning, we also designed corresponding experiments when we originally wrote this paper. Four scenarios are included, separately, they are: combined tuning for ARM95 and ARM97 cases, combined tuning for three tropical convection cases, combined tuning for TOGAII/TWP06 and all five cases. From results we can see that the parameter responses for ARM95 and ARM97 have a similar distribution and thus the results of the multi-objective optimisation for both of them reflect this. The same trend is reflected similarly in the TOGAII and TWP06.

Line 35: The statement that the Morris SA method cannot get the interactive sensitivity could be wrong. Aurally, the standard deviation of MOAT samplings can stand for the interactive effect of one parameter with others (Morris, 1991).

We have refined the description of the Morris SA method in the revised manuscript so as not to introduce ambiguity in Line 37-39: "Morris SA can give the individual sensitivity of each parameter, including their interaction sensitivity. However, this is not intuitive enough if the user wants to know directly from a combined perspective which set of parameters has the most significant effect on the results."

Line 45: as the part of introduction, the authors should explain the challenge of the SA methods, why you choose Morris and Sobol, the computational cost issue, surrogate problems using machine learning. If there are previous works, what's your contribution?

We will complete this part based on a further full investigation. In the revised version, we will give a more detailed explanation. Both Morris and Sobol are typical SA methods that have a wide range of applications in many fields. As there are already proven application examples, it makes sense to conduct experiments based on the above methods. In addition to this, we have introduced several new SA methods that have been proposed in recent years. Although SCAM, as a single-column model, already consumes less computational resources than GCM, the computational resources of a system are not infinite. To further explore their parametric features, a study using surrogate model is necessary. Surrogate models [2] can significantly reduce the computation time of individual tasks, thus making it possible to scale up experiments.

Our contribution lies in the fact that we have trained the surrogate models on SCAM using a regression-based approach, and with the help of the surrogate models we have conducted parameter sensitivity tests for combinations, as well as tuning for the most significant parameter combinations.

Line 55: the authors should do comprehensive literature research, even for GCM, there are a large number of work for tuning, such as Yang et al. (2013) and Zou et al. (2014). In addition, the NN surrogate model is used to tune as well. But the authors don't introduce the previous work and challenge in terms of this issue. The introduce section should be more clear.

We will complete the introduction section by conducting a more detailed literature survey of the work related to the content of this paper. This also includes the literature you mention, such as [3] and [4]. Yang et al. [3] analysed the sensitivity of nine parameters in the ZM deep convection scenario for CAM5 and used the SSAA (Simulated stochastic approximation annealing) method to optimize the precipitation performance in different regions by zoning. Zou et al. [4] conducted a sensitivity analysis for seven parameters in the MIT-Emanuel cumulus parameterization scheme in RegCM3. The precipitation optimization process for the CORDEX East Asia domain was carried out using the MVFSA (Multiple very fast simulated annealing) method.

The current challenges lie in the following areas. (1)No similar experiments have been conducted on SCAM. The short run time of the SCAM makes it easier to obtain more samples in a short period of time and thus scale up the experiments. (2)The usual SA methods will give the sensitivity of the individual parameters. But when we look at a set of parameters, is the best combination of N parameters the top N sensitivity of a single parameter? This is a question worth exploring. (3) As various neural network methods are applied to the field of regression, more appropriate regression methods should be applied to the parametric study of Earth system models. Unlike the various public data sets commonly used, ESM-based experiments will also further enrich the practical implications of research in the field of regression analysis.

In our manuscript, we innovatively use a neural network-based agent model for parameter tuning of different cases in SCAM. For each case studied in the paper, the surrogate models are trained separately based on different methods, and the best performing model is selected by their training errors (RMSE).

Line 75, Acutely, there are existing SA and tuning workflow used in climate models, such as PSUADE and DAKOTA, the authors don't compare their workflow to these packages. It's not new for the community.

This is indeed something we need to improve further in the literature survey. We have done a survey of the packages such as PSUADE, DAKOTA and STATA etc., including some studies based on their work in different fields. These packages can indeed implement the functionality of SA and tuning. We also compared the workflow proposed in this paper with the above software packages. From a method perspective, we added the comparison of the new SA methods in recent years, such as RBD-FAST, Delta and HDMR. The above approaches haven't been fully supported by all the packages above. For using neural networks to train surrogate models, DAKOTA currently only supports neural networks with a single hidden layer. Meanwhile, the proposed method uses more types of neural networks and supports the adaptive selection of the best performing network to train the surrogate model.

Line 88: The authors don't mention the 30% improvement for tuning error and computational cost. How do they come from?

The tuning error is compared with control experiments. This is the result of a simulation using the default parameter values of SCAM. The improvement in computational overhead comes from comparing it with a traditional optimization workflow. In the original optimization

workflow, the original SCAM is invoked for each calculation of the objective function. In the approach proposed in this paper, the objective function is replaced with a surrogate model. This allows the overall workflow execution time to be compressed, thus the computational overhead could be reduced.

Table 2 is wrong. Each IOP file includes many variables, not just these four variables. Therefore, the statement that you choose precipitation is wrong.

We are sorry for the misunderstanding. We agree that there are many variables in the IOP file for each case, not just these four, as can be seen from Table 2. We include Table 2 in our manuscript mainly to illustrate that the variables contained in different IOP files are different. We will add this in the revised manuscript.

Line 120: This issue could not be a significant challenge. Some simple scripts can collect the data.

Yes, scripts are able to do that. Of course, we will also achieve efficient management of experimental data in our method. Due to the amount of data generated during this stage, better data management is necessary. We have adopt a more reasonable description in the revised manuscript.

We have revised these statements in Line 127 as follows: "In this step, we also integrate the collection and processing script for the post-sampling results. As the output of SCAM is stored in binary files in NetCDF format, the precipitation variables we want to study need to be extracted from a large number of output files in order to proceed to the next step. This will further accelerate the degree of automation of scientific workflows and thus accelerate the conduct of research in this area of the earth system models."

The statement about Morris is wrong, see 3.

We plan to correct this issue in Line 38 as follows, "Morris SA can give the individual sensitivity of each parameter, including their interaction sensitivity. However, this is not intuitive enough if the user wants to know directly from a combined perspective which set of parameters has the most significant effect on the results."

In section 3.1: sampling is not equal to SA. In this section, the authors introduce the SA methods. You should consider change the structure or change the title.

We agree that sampling can't be confused with SA. We will refine the article structure of this section in a revised version to make the expression easier to understand. The sampling method and the SA method will be split into two subsections to be described.

Subsection 3.1, entitled "Sampling of SCAM", will be devoted to the sampling methods covered in this paper and the rationale for their selection.

Subsection 3.3, entitled "Sensitivity analysis for a single parameter and combinations of parameters (enabled by the NN-based surrogate model)", will provide a more detailed description of the SA methods covered in this paper and the rationale for their selection. At the end

of this subsection, our proposed combined SA methods for supplementary validation will be introduced.

Line 177: it could be better to compare NN with other surrogate models, such as xgboost, ResNet.

Yes, the properties of the various surrogate models are something we all care about. Through our research, we learned that both XGBoost[5] and ResNet[6] can be used to perform regression tasks and train surrogate models. Here, we will compare the effectiveness of several methods such as LR (Linear Regression), RF (Random Forest), MLP (Multi-Layer Perceptron), XGBoost and ResNet for training surrogate models. The results are shown in Table 6. Since ResNet has better performance in model training, we will choose ResNet as the network for training the surrogate model. Subsequent experiments will also be supplemented in the revised paper.

Line 184: The 768 samples seem not enough for training NN, do the authors evaluate the performance of NN? In Figure 4, how do you define the accuracy?

Yes, we have evaluated the performance of NN and measured its ability on regression. After our analysis of the sampling results, we found that 768 samples can already cover the range of values of the parameters and output variables. Therefore, the number of parameters selected is sufficient to obtain a good training effect. The results of the training process and the experimental results also support our conclusion. For the second point, The $R^2$ score is used as the accuracy rate in Figure 4. It is defined as

$$R^2 = \frac{\sum_i (\hat{y}_i - \bar{y})^2}{\sum_i (y_i - \bar{y})^2} \tag{2}$$

where $\hat{y}$ is the predicted value and $\bar{y}$ is the mean of the test set. In addition to this, RMSE was also used as an important metric to measure the loss during training.

Line 185: Do the authors do the hyper-parameter tuning for NN?

In the initial version of our manuscript, our aim was mainly to verify the feasibility of this approach, so we used a set of empirical parameters to train the NN. Now we will conduct hyperparameter tuning trials in NN training to make the proposed method more solid. We have refined these in Table 5.

Line 195: Due to the uncertainties of each method, ensemble can't guarantee to reduce the error.

We agree that there are errors in out initial manuscript. We feel sorry for the term "integrate" in the original paper may bring some ambiguity. We've used several SA methods to make a more intuitive side-by-side comparison, which allows us to find the best method for each case. It's not necessarily about reducing the error.

Line 210: Equation (1), the left hand of this equation is not the number of processes. It should be the number of simulations. The number of simulations could depend on different sampling method. For Morris, it could not require such number of samples. It is not clear for this description.

Since SCAM is a single-process task, one simulation is equivalent to one process. Of course, as you said, it is necessary to distinguish between simulation and process more accurately. For Morris, 768 was used as the sample number to keep the sample number of various sampling methods consistent. We will add the above contents to refine these issues in a revised version of the manuscript.

For a SCAM simulation, it usually requires 10-20 minutes, why do you require more than one hour?

We agree that for a single SCAM job it only takes one process to run, and it does not take very long for a single SCAM to run. The one hour mentioned here is mainly the time it takes to complete a whole workflow of parameter analysis and tuning. This includes multiple iterations and delays in queuing batch jobs. We will refine these descriptions above in the revised manuscript and try to avoid ambiguities.

It is confusing that you can re-use the sampling from SA to train the surrogate model for tuning, but in section 3.3, you mention that the surrogate model can be also used in SA?

We agree that the trained surrogate model can be also used in SA. Since we want to find the most sensitive set of parameters in combination, this process requires a larger experiment scale. Therefore, to perform large-scale parametric analysis experiments faster, we use surrogate models to speed up the process.

Line 233: how do you get the conclusion? It is not convinced.

Here, the running time is compared to one full simulation of the original SCAM. Here we are trying to make two points. On the one hand, the surrogate model can simulate the output of SCAM more accurately. This makes it an effective alternative to SCAM in terms of parameter response. On the other hand, the execution time of the surrogate model is very short, which saves a lot of time for parameter analysis and optimization tuning. Combining these two reasons above, the use of a surrogate model for parameter tuning of SCAM is a very reasonable strategy. We have also revised the description in this subsection.

It is very confusing for section 3.4 and is difficult to follow your idea. You could consider to re-organize this section.

We're sorry for not being able to make it easy for you to follow our idea. We have reorganized this subsection to make it easier to understand. In this subsection, we focus on an enhanced SCAM parameter tuning process. There are two main contributions. On the one hand, we use the trained surrogate model as the objective function for optimization, which allows the optimization time to be compressed considerably. On the other hand, we propose an enhanced parameter optimization process based on grid search. We use grid search to reduce the search range in the optimization process and thus obtain better results in fewer iterations. In the revised manuscript we will elaborate further on these two points above.

Line 270: how many samples do you have for the correlation? Do you do the p-value test?

Here, we carried out correlation analysis on the respective optimal solutions of the five

SCAM cases. Therefore, a total of five vectors were used to calculate the correlation For the second point, now we have added tests on p-values in the revised manuscript.

Figure 7: it is difficult to evaluate the tuning performance in Figure 7. It could be better to use metrics like Yang et al. (2013).

What we want to express in Figure 7 is the statistics of the sampling results. The phrase "when the parameters are tuned" refers to the fact that the parameters are given different values during the sampling stage, not the final tuning stage. We're sorry for this ambiguity and will respect your suggestions to revise the description appropriately. When revising, we fully refer to the relevant statements of [3] in order to make the results more clear and easy to understand. The results after reanalysis are shown in Figure 3.

Figure 4, Figure 5 are the results but appear in section 3 (methods). It could be reorganized.

The results described in these two figures belong to the pilot test, which is to prove the rationality of our experimental ideas. We respect these suggestions and provide appropriate revisions to make the organization of our paper more rigorous. The relevant statements have already been described in Section 4.

Line 313: pz2 (c0_ocn) should be high influence on the ocean case. But in Figure 8, it doesn't have the high effect on PRECT at TWP. Could you explain the reason?

It can be seen from Figure 8 that no matter which sensitivity analysis method is used, pz2 (c0_ocn) has a certain influence on the results. It is a deep convection parameter related to the ocean-land intersection, but this does not mean that it must be the one that has the most influence on TWP. The influence of the ocean has been demonstrated here.

Line 317: the reason for the different between ARM 95 and ARM 97 is not convinced.

The difference between the cases is also the focus of our interest and attention. Indeed, the gap between ARM95 and ARM97 indicated by the experimental results in this paper is worthy of attention. We retrained the surrogate model using ResNet and used RMSE as the metric of error. We found that the response of the precipitation output to the parameters in the new surrogate model was very similar for both cases which can be shown in Figure 13-14. This suggests that in the original version, the difference between the two examples is due to the error in the model. Your advice on the choice of the neural network would also be much appreciated here.

Line 345: are the 16 iterations enough for convergence? Why don't use the general optimization, such as PSO, GA that you mentioned in the introduction section?

In our experiment, 16 rounds of iteration can already make the results converge, as shown in Figure 9. So the selection of the number of iterations is sufficient. WOA[7] is chosen here mainly because it is an optimization algorithm proposed later. Theoretically speaking, in the optimization stage of the workflow proposed in this paper, no matter what kind of optimization algorithm is selected, the purpose of optimization can be achieved, including PSO, GA, etc.

**2  Replies to minor issues**

*In Fig. 1, is there an arrow pointing from "SA methods" to "testing of combinations"?*

There isn't an arrow here because only the trained surrogate models are used for testing combinations of parameters.

*Caption in Figure 1:  The sentence "SCAM launcher, the data collector and the jobs therein represent the batch execution of the SCAM algorithm" should be rephrased.  What are the "jobs" and "batch execution"?  It is not clear.*

There isn't an arrow here because only the method of the surrogate model trained using neural networks will be used for parameter combination SA. Here, a job refers to a single simulation of SCAM. A batch is a collection of jobs that are submitted to the computing queue at once. We will elaborate more in detail in the revised manuscript.

*Line 25:  should explain "ne30"*

This is a description of the spectral element method grids[8]. It refers to a model grid with a $ne30np4$ spectral element (approximately 1-degree) atmosphere and land grids. $ne[X]np[Y]$ are cubed sphere resolutions where X and Y are integers. The short name generally is $ne[X]$. The corresponding descriptions have been reorganized.

*Line 38:  the reference of Sobol method is wrong, pls use the original paper.*

We will correct it [9] in the revised manuscript.

*Line 43:  The QMC and LHC are sampling methods, not SA methods.*

We will correct this in the revised manuscript. Sampling methods and SA methods will be more clearly distinguished to avoid confusion.

*In Table 3, how do you select these parameters?  And how do you define the range of each individual?*

These parameters were chosen mainly from reference [10]. The parameters range from 50% to 150% of the default value.

*Line 94:  there should be a reference for SCAM5.*

We have added reference [11] in the revised manuscript to refine the introduction of SCAM5.

*Line 102:  all sites belong to ARM.*

We have corrected this issue in the revised manuscript. We will also carry out a more detailed study later.

We have corrected it in the revised manuscript.

Line 116: "is an important issue" change to "are important issues"

We have corrected it in the revised manuscript.

Line 130: It is only suit for SCAM. For GCM, it is impossible.

Indeed, as you say, it is not practical to run large batches of GCM jobs even on HPC. We have refined this description in the revised manuscript.

Line 165: please explain the "second-order sensitivity"

It refers to the mutual influence between two parameters. We have refined this exposition in the revised manuscript.

Table 4: the reference of Sobol is wrong.

We have corrected this reference [12] in the revised manuscript.

Line 174: please consider the correct position of this sentence.

We have corrected it in the revised manuscript.

We have added some necessary information and details in our revised manuscript. What we want to express here is the following: using the above methods, the set of M parameters that most influence the result may be difficult to obtain directly. The method proposed in this paper just fills this gap.

We have corrected it in the revised manuscript.

Line 215: how do you get the number of multiple thousand?

Combining Equation (1) with the practical problem studied in this paper, we can see that when $C = 5, p = 3$ and $L = 10$, 5,000 simulations are needed even if only one parameter combination is considered. This will be astronomical when more combinations are considered. We have also refined the above in the revised manuscript.

**References**

[1] J. Guo, Y. Xu, H. Fu, W. Xue, L. Wang, L. Gan, X. Wu, L. Hu, G. Xu, and X. Che, "A learning-based method for efficient large-scale sensitivity analysis and tuning of single column atmosphere model (scam)," *Geoscientific Model Development Discussions*, vol. 2022, pp. 1–28, 2022. DOI: 10.5194/gmd-2022-264. [Online]. Available: https://gmd.copernicus.org/preprints/gmd-2022-264/.

[2] K. Cheng, Z. Lu, C. Ling, and S. Zhou, "Surrogate-assisted global sensitivity analysis: An overview," en, *Structural and Multidisciplinary Optimization*, vol. 61, no. 3, pp. 1187–1213, Mar. 2020, ISSN: 1615-147X, 1615-1488. DOI: 10.1007/s00158-019-02413-5. [Online]. Available: http://link.springer.com/10.1007/s00158-019-02413-5 (visited on 04/03/2023).

[3] B. Yang, Y. Qian, G. Lin, L. R. Leung, P. J. Rasch, G. J. Zhang, S. A. Mcfarlane, C. Zhao, Y. Zhang, and H. Wang, "Uncertainty quantification and parameter tuning in the cam5 zhang-mcfarlane convection scheme and impact of improved convection on the global circulation and climate," *Journal of Geophysical Research Atmospheres*, vol. 118, no. 2, pp. 395–415, 2013.

[4] L. Zou, Y. Qian, T. Zhou, and B. Yang, "Parameter tuning and calibration of regcm3 with mit–emanuel cumulus parameterization scheme over cordex east asia domain," *Journal of Climate*, vol. 27, no. 20, pp. 7687–7701, 2014. DOI: https://doi.org/10.1175/JCLI-D-14-00229.1. [Online]. Available: https://journals.ametsoc.org/view/journals/clim/27/20/jcli-d-14-00229.1.xml.

[5] W. XingFen, Y. Xiangbin, and M. Yangchun, "Research on User Consumption Behavior Prediction Based on Improved XGBoost Algorithm," en, in *2018 IEEE International Conference on Big Data (Big Data)*, Seattle, WA, USA: IEEE, Dec. 2018, pp. 4169–4175, ISBN: 978-1-5386-5035-6. DOI: 10.1109/BigData.2018.8622235. [Online]. Available: https://ieeexplore.ieee.org/document/8622235/ (visited on 04/03/2023).

[6] L. Shi, C. Copot, and S. Vanlanduit, "Evaluating Dropout Placements in Bayesian Regression Resnet," en, *Journal of Artificial Intelligence and Soft Computing Research*, vol. 12, no. 1, pp. 61–73, Jan. 2022, ISSN: 2449-6499. DOI: 10.2478/jaiscr-2022-0005. [Online]. Available: https://www.sciendo.com/article/10.2478/jaiscr-2022-0005 (visited on 04/03/2023).

[7] S. Mirjalili and A. Lewis, "The Whale Optimization Algorithm," *Advances in Engineering Software*, vol. 95, pp. 51–67, 2016, ISSN: 18735339. DOI: 10.1016/j.advengsoft.2016.01.008.

[8] (). "Model grids – cime master documentation," [Online]. Available: http://esmci.github.io/cime/versions/master/html/users%5C_guide/grids.html.

[9] I. M. Sobol, "Sensitivity analysis for non-linear mathematical models," *Mathematical modelling and computational experiment*, vol. 1, pp. 407–414, 1993.

[10]  Y. Qian, H. Yan, Z. Hou, G. Johannesson, S. Klein, D. Lucas, R. Neale, P. Rasch, L. Swiler, and J. Tannahill, "Parametric sensitivity analysis of precipitation at global and local scales in the community atmosphere model cam5," *Journal of Advances in Modeling Earth Systems,* vol. 7, no. 2, 2015.

[11]  P. A. Bogenschutz, A. Gettelman, H. Morrison, V. E. Larson, D. P. Schanen, N. R. Meyer, and C. Craig, "Unified parameterization of the planetary boundary layer and shallow convection with a higher-order turbulence closure in the community atmosphere model: Single-column experiments," *Geoscientific Model Development*, vol. 5, no. 6, pp. 1407–1423, 2012. DOI: 10.5194/gmd-5-1407-2012. [Online]. Available: https://gmd.copernicus.org/articles/5/1407/2012/.

[12]  I. Sobol, "Global sensitivity indices for nonlinear mathematical models and their monte carlo estimates," *Mathematics and Computers in Simulation*, vol. 55, no. 1, pp. 271–280, 2001, The Second IMACS Seminar on Monte Carlo Methods, ISSN: 0378-4754. DOI: https://doi.org/10.1016/S0378-4754(00)00270-6.

---

## Referee Report (RR1)

**A learning-based method for efficient large-scale sensitivity analysis and tuning of single column atmosphere model (SCAM)**
Guo et al.

In this study, the authors explore the use of a neural network-based surrogate model to find optimal parameter values for SCAM simulations of five IOPs. The conclusion is that the surrogate model out-performs SCAM when run with these optimal parameter values. Again, the paper should undergo major revision before being accepted for publication.

**Major comments**
In general, one of the big arguments that this paper makes is that the surrogate model is an important tool in model development because it enables a more efficient search for optimal parameter values. The authors have expanded that notion slightly by adding a discussion of optimized values computed over all 5 cases vs. individually, but have not explored potential changes in accuracy in that scenario, and have not confirmed that the improvements present in the surrogate model are matched in SCAM simulations. Beyond those topics, there is no indication that these optimal values will in fact improve global CAM simulations; if that could be shown, the impact of the paper would *greatly* increase. In general, some assertions made in the text are not supported by what's shown in the study or cited, and should be more strongly backed up. Additionally, references should again be updated and checked, and figure/table descriptions should fully describe the elements present.

**Specific comments**
- Line 17: Bacmeister et al. (2014) is not the correct citation for CESM, please refer to recent documentation of the full model rather than CAM alone. Similarly, the reference for CAM in the next sentence should be updated; the current citation (Dennis et al. 2012) refers to the development of a spectral element dynamical core; more recent and broader citations are available.
- Line 22: "SCAM…[is] a cheaper and more efficient alternative model for the purpose of tuning physics parameters." This discussion of SCAM and its use deserves more nuance. The authors here seem to suggest that parameter tuning can be done purely in SCAM and that SCAM is a suitable surrogate for running the full global model. While SCAM is a useful tool for initial testing of parameterizations (and perhaps tuning), it is not in fact a full substitute for running CESM globally.
- Line 29: "…has the most significant effect on results. and the Sobol method that uses the…" Sentence fragment re: Sobol method, perhaps left over from previous edits?
- Lines 67-69: These toolkits are not well introduced; if they can implement SA and tuning, what exactly is the downfall of using them? Please elaborate on what niche this is filling. The authors state that the use of more parameters and cases increase the exploration space exponentially and make the task "almost an impossible job," but don't show clear evidence of that.
  - Line 76 states that the benefit of the approach is the addition of "new SA methods in recent years." – which methods? Is that addition the only benefit (in

which case it doesn't seem that the expansion to more cases/parameters is nearly impossible in those packages, as stated)?

- The reference to Pathak et al. (2022) is missing a DOI and is incomplete.
- Line 106: In general, the selection of PRECT is perhaps reasonable given its importance societally and the focus on convective parameterizations. Yet it seems like Pathak et al. assessed the response of a number of climate fields to their perturbations, which more closely mimics the process of model tuning. Is it not worthwhile to expand to more indicators of model performance? Near surface T/Q or surface fluxes, for instance, seem to be available for most IOPs.
- Line 110: The bounds for parameters are selected to be 50% and 150% of the values; are there not more firm limits based in the literature for what a realistic range may be?
- Lines 112-114: please indicate which fields you've now made tunable parameters (vs. which ones already were).
- Lines 116-118: As before, I'm still not entirely clear on what the authors mean by saying the programs needed to be recompiled and that the recompilation has enabled a larger number of concurrent instances on the Sunway supercomputer. I *believe* that this is referring to the fact that all parameters in table 3 are now namelist options, thus the model does not need to be re-built each time and can instead be cloned from a single, pre-built base case whose namelist can be modified. That does cut down on run time, but doesn't seem related to the programs running on Sunway needing recompilation due to the supercomputer platform. Suggest clarifying that this is simply the shift of hard-coded to namelist defined variables.
- Line 174: "we decide another S=768, where NMorris is 64 and NSaltelli is 32." What's the sensitivity to those choices? Does 64 and 32 show some convergence to a solution?
- Line 187: "Compared to other networks, ResNet has the advantage of using less pooling." What does this mean? Should elaborate on what pooling is first if this is an important point to make. If the following explanation is intended to elaborate on why ResNet is preferred, perhaps rephrase for clarity.
- Line 190: "ResNet18…" similar question to above – what's the sensitivity to number of layers? Is 18 enough (and how have the authors determined that)?
- Lines 208-209: "As the sensitivity values calculated by different methods are of different orders of magnitude…" – was this expected, and arises as a by-product of using different SA methods? I.e., are the units not *all* in the form of some response driven by a change in the parameter?
- Lines 232-233: "This way, we can determine the number of parameters in the combination, taking into account the tuning effect and the amount of computation." – Not clear what's meant here. What is meant by 'number of parameters in the combination,' isn't that set? How is the 'tuning effect' taken into account, as this seems like a tool that aids in tuning rather than the tuning have an effect on the process.
- Line 234: "lead to the most significance in output" – should this read the most significant *improvement* in output, or the largest increase in accuracy? Could be interpreted as the biggest change in output as written currently.

- Line 292: "…the distribution of pz4(tau)." –if zmconv_tau in figure, suggest consistent naming conventions.
  - In general, it would be more useful to use the zmconv_tau naming convention as in Fig 3 for Fig. 5 as well, to make it easier for the reader to quickly scan through.
- Figs 3 and 4 – using consistent colors in Figure 4 to represent the cases shown in Fig. 3 would increase the readability.
- Fig 4 – since the main point is the increase in *accuracy* of PRECT rather than just the change, perhaps there's some way to indicate the change in precipitation bias rather than the overall percent change? It's possible this belongs more in Figure 3 (i.e., getting a sense of how much the bias has improved given each parameter, rather than just which results are "better" than the control)
- Line 300: "Different from other cases, this case is a ocean case and is located in the Atlantic Ocean" – This seems to suggest that GATEIII is the only ocean case; does TOGAII not also target ocean? If this isn't the only ocean case, what about it being in the Atlantic makes it particularly unique?
- Table 5 –doesn't seem to add much to the paper; could this not just be said in the text?
- Lines 309-310: "It can be seen that ResNet has the best performance…" – The RMSE is indeed the lowest, and not just by a little but by quite a margin! Was this fully expected a priori? Is this consistent with any other literature?
- Figure 5: How are the sensitivity results normalized?
- Lines 322-323: "differences between the individual parameters are not as wide as in the other cases" – not sure what's meant here; that the range in parameter values is smaller? The ranges should be consistent across cases, yes?
- Line 324: "This is also consistent with its position" – by position, is the intent to say that the geographic location is more comparable? Please consider rephrasing for clarity.
- Section 4.3: The discussion of sensitivity to SA method should be expanded and clarified. It is stated that the SA results are shown, but which column is this? The Morris test seems significantly more sensitive to parameter variations than the other tests; are the others better suited for one precipitation regime vs. another? Or how is the different sequence of samples coming into play here?
- Lines 357-358: "However, the computation amount increases exponentially during the test." But isn't the benefit of using a surrogate model that you can tune more parameters at a time than using the traditional approach? If all tests for 3 parameters can be conducted in less than a minute, why not tune 4?
- Figure 7: It's unclear what the max/min stars represent or add to the plot; the min stars also aren't visible from what. I'm unsure why the y-axis is positive only; do none of the parameter combinations lead to a *decrease* in precipitation?
- Table 7: Presumably, in each case these parameters are being assigned unique values (i.e., pz4 = ?? for ARM95 vs. TOGAII); it would be helpful to include those values here as well for comparison (or reference later table).
- Line 366: "meaning average error" – should this now be RMSE?
- Table 8 is somewhat hard to interpret. Is "error" here referring to the difference in SCAM and the surrogate model? Or is it related to either's agreement with observations? Is

SCAM doing a better job simulating PRECT at all locations vs. the surrogate model (given the lower RMSE values) – that doesn't seem to be reflected in the text.

- Figure 10: Please clarify if the "model simulation output" is from SCAM or the surrogate model. It also makes more sense to plot the x-axis in time coordinates rather than timestep number.
- Lines 377-378: "The tuning of the SCAM parameters is quite productive on the time scale." – not clear what this means with the current wording. What time scale is being assessed? Does 'productive' mean increased accuracy?
- Line 381: "Although still below the observed level at about 1300 steps of TOGAII, improvement is also reflected." – I'm not sure why this timestep/case is singled out here, it seems like there are plenty of other examples of this. Similarly, the point is made that in TWP06 there are instances where control is less than observed; again, that's present in all cases, yes?
- Section 4.5: I'm not clear why the baseline and optimized models should have any different computational cost, unless baseline refers to SCAM and optimized to the surrogate model. If that's the case, that should be stated much more clearly and often. More generally, when considering overall model performance, the same model structure should be used. So of course use the surrogate model to find the optimum parameters, but to assess differences in simulation accuracy, plug those values back into SCAM. If not comparing direct SCAM cases, the accuracy gains aren't very understandable.
- Lines 388-389: "The use of NN trained surrogate models for parameter tuning can further save computational resource overhead and, in terms of results, can meet or exceed traditional optimization methods in most cases." – this may be overstated. In terms of saving computational resources, this is not illustrated by Figure 12. Figure 12 includes time in the job queue for the SCAM case; adding a bar to each case for the computational time *excluding* queue time would be needed to support this statement.
- Lines 391-392: "The model can get an enhancement in performance from 6.4%-24.4% in precipitation output," – again, I think the important part is to plug this back in to SCAM using the parameters determined by the surrogate model. Unless the suggestion is to use the surrogate model for all simulations (rather than for finding optimal parameters and model tuning), this is a critical step.
- Table 9: A better description is needed; "multi" cases undefined here.
- Section 4.6: An important point is lacking from the discussion, which is how well does the model perform (RMSE per site) when using the values found to optimize performance across all 5 sites (multi(d))? The potential benefit of regionally-varying parameters is noted, but it's not illustrated how important is it to use potentially regionally refined values vs. ones that are optimized for global performance.
- Related to above, and though this might be outside what the authors' computational resources allow – the most convincing argument for the study would be to apply the tuned parameters from the multi(d) case to a global run. That would illustrate true benefit from using a surrogate SCAM model to find optimal parameter values; without that, the potential impact of the paper is severely more limited.

---

## Referee Report (RR2)

Comment 1: In this paper, although the authors evaluate the accuracy of the NN model in terms of precipitation, it probably exists the inconsistent between ML and real model, which don't be highlighted in this paper.

The propose of this manuscript provides a method for efficient tuning of SCAM. This comment question the challenge that the offline surrogate method could not guarantee the optimal solutions in surrogate model can transfer to the real model. The authors try to explain it by the high accuracy of the surrogate model. However, the surrogate models are difficult to learn the optimal solutions because they can't be sampled in the training data. Therefore, the efficiency of this method could be affected. The authors do not statement this point and do not give a solution about this issue.

Comment 2: The authors believe that due to the high computational cost of the GCM, the SCAM can be the alternative model for parameter SA and tuning. In reality, the optimal parameters tuned in SCAM could not be suitable for GCM, due to the global regions and more complex large-scale circulation.
I do not figure out a clear path from tuning in SCAM to GCM. The authors claim that "tuning on SCAM cases located in different regions, we can find commonalities and patterns in the parameter response of these cases". However, there are several challenges. The current SCAM can only support several sites and cannot represent the global catachrestic. Second, there are different optimal parameter values at different sites. Even for each site, there are different local optimal solutions. It is difficult to find the so-called commonalities for these optimal parameters.  The forcings are different between SCAM and GCM, optimal parameters could not achieve good performance in GCM.

Comment 4: The authors separately tune the parameters in SCAM for each site and get the different sensitive parameters and different optimal values. It is difficult to transfer this information to GCM. If the authors can do the multi-objective tuning for these sites with the same parameters, it could be helpful for global model tuning because these SCAM sites indeed represent the different regimes.
For figure 13-14, are the 'ranked 1' points local optimal or global optimal? Different regions (a-d) have the different optimal points. It's difficult to determine the optimal point in one region (like a) achieves the optimal solution in the other regions. It's also difficult to find a unified optimal solution for GCM, although it could be helpful to design a new parameterization with different parameter values at different regions.

Comment 5: For the workflow, there should be a "metrics" component because it is very important for tuning. No matter SCAM or GCM, the tuning metrics could be the cost function between model simulations and observations. The different designs could affect the optimization. In terms of the metrics, it could consider the 1) different statistic errors between simulation and observation, such as RMSE, performance score like Yang et al. (2013), 2) one objective or multiple objectives, and how to deal with the multiple objectives.

Does the metric function only consider the precipitation? However, it could consider more variables in GCM. It could make sense using time series RMSE in SCAM. In GCM, mean state, spatial correlation, spatial standard deviation, spatial RMSE should be involved in the metrics. The simple metrics in SCAM also pose the challenge for GCM.

Comment 6: Line 35: The statement that the Morris SA method cannot get the interactive sensitivity could be wrong. Aurally, the standard deviation of MOAT samplings can stand for the interactive effect of one parameter with others (Morris, 1991).
"However, this is not intuitive enough if the user wants to know directly from a combined perspective which set of parameters has the most significant effect on the results." The sentence is confusing. The mean represents the main effect, and the standard deviation represents the interactive effect.

Comment 7: Line 45: as the part of introduction, the authors should explain the challenge of the SA methods, why you choose Morris and Sobol, the computational cost issue, surrogate problems using machine learning. If there are previous works, what's your contribution?
In the revised introduction section, the authors explain why they use Morris and Sobol with the sentence "Both Morris and Sobol are typical SA methods that have a wide range of applications in many fields. As there are already proven application examples". It seems this work use them because the previous work used them. This expression is not serious. There are a lot of sensitive analysis methods. The authors should carefully state the advantage of these two methods. The authors also claim the reason of using surrogate that the SCAM requires not very cheap computational cost. However, it only takes several optimal iterations.

Comment 8: Line 55: the authors should do comprehensive literature research, even for GCM, there are a large number of work for tuning, such as Yang et al. (2013) and Zou et al. (2014). In addition, the NN surrogate model is used to tune as well. But the authors don't introduce the previous work and challenge in terms of this issue. The introduce section should be more clear.
I think the explanation of innovation of this manuscript does not make sense. The motivation of this manuscript tried to tune SCAM and transfer to GCM. Since these methods have been applied in GCM, it is confusing that the authors claim they are not used in SCAM. For the 2nd point, the previous works get the sensitive parameters by perturbing multiple parameters. Then the sensitive parameters can be tuned using optimization algorithms. It is not clear for your point.

Comment 9: Line 75, Acutely, there are existing SA and tuning workflow used in climate models, such as PSUADE and DAKOTA, the authors don't compare their workflow to these packages. It's not new for the community.
The explanation is not convinced. The authors do not analyze the advantage and disadvantage of these existing framework. Why do not you improve the existing framework? The references of PSUADE and DAKOTA are wrong.

Comment 11: Table 2 is wrong. Each IOP file includes many variables, not just these four variables. Therefore, the statement that you choose precipitation is wrong.
Although table 2 lists many variables, this manuscript only consider precipitation in the metrics. It should be considered more variables in GCM.

Comment 16: Line 184: The 768 samples seem not enough for training NN, do the authors evaluate the performance of NN? In Figure 4, how do you define the accuracy?
The authors do not give any result to prove that the 768 samples are sufficient. The offline surrogate model methods are difficult to achieve high accuracy at optimal local regions. The tuning efficiency is easily affected by the local samples.

Comment 27: Line 313: pz2 (c0_ocn) should be high influence on the ocean case. But in Figure 8, it doesn't have the high effect on PRECT at TWP. Could you explain the reason?
Unfortunately, the author is not familiar with the parameters. C0_ocn is parameter of autoconversion coefficient over ocean in ZM deep convection scheme. It is not the ocean-land interscection.

Comment 29: Line 345: are the 16 iterations enough for convergence? Why don't use the general optimization, such as PSO, GA that you mentioned in the introduction section?
The statement is very misleading. Any optimization algorithm cannot converge in 16 iterations! This comparison is not serious and it requires at least several hundred iterations.

---

## Author Response (AR2)

**A Summary of the Major Changes**

Dear Editors and Referees,

We are grateful for your thoughtful and detailed review of our manuscript. Your comments and suggestions have been invaluable in helping us improve the quality and clarity of our work. We appreciate the time and effort you have invested in providing such constructive feedback. We have carefully considered each of your points and have addressed them as follows.

- We have redefined the abstract, fundamental motivations, innovative aspects, and primary contributions of this paper.

- We have conducted more comprehensive research into this field and improved the references accordingly.

- We have restructured the framework of this paper and streamlined the language, aiming for a more concise and straightforward presentation.

- We have expanded the exploration to include additional variables such as humidity, temperature, and cloud.

- We have corrected some minor details in the original content of the manuscript.

Best regards,

All authors

**Replies to Referee #1, GMD-2022-264**

**Jiaxu Guo on behalf of all authors**

Thank you very much for your patient and detailed comments on our work[1]. These valuable comments are very helpful for us to improve this paper. After carefully reading all the questions, we have answered each of them and will make appropriate corrections in the revised version of our manuscript.

In this attachment, the red paragraphs represent your comments, and the black paragraphs below are our corresponding replies.

Comment 1: Line 17: Bacmeister et al. (2014) is not the correct citation for CESM, please refer to recent documentation of the full model rather than CAM alone. Similarly, the reference for CAM in the next sentence should be updated; the current citation (Dennis et al.2012) refers to the development of a spectral element dynamical core; more recent and broader citations are available.

Thanks for your comments. We have corrected the above-mentioned issues in the revised manuscript[2, 3].

Comment 2:Line 22: "SCAM...[is] a cheaper and more efficient alternative model for the purpose of tuning physics parameters." This discussion of SCAM and its use deserves more nuance. The authors here seem to suggest that parameter tuning can be done purely in SCAM and that SCAM is a suitable surrogate for running the full global model. While SCAM is a useful tool for initial testing of parameterizations (and perhaps tuning), it is not in fact a full substitute for running CESM globally.

Thanks for your comments. We apologize for any confusion caused. The main objective of this paper is to use machine learning methods to construct a surrogate model for SCAM and thereby assist in its parameter tuning. It is not suggested here that SCAM itself is a surrogate model for a global model, although some conclusions may provide insights for our study of global models. We have also removed ambiguous sentences and corrected corresponding descriptions in the text.

Comment 3: Line 29: "...has the most significant effect on results. and the Sobol method that uses the..." Sentence fragment re: Sobol method, perhaps leb over from previous edits?

Thanks for your comments. We apologize for the typo here, and we have removed the duplicated sentence fragment.

Comment 4: Lines 67-69: These toolkits are not well introduced; if they can implement SA and tuning, what exactly is the downfall of using them? Please elaborate on what niche this is filling. The authors state that the use of more parameters and cases increase the exploration space exponentially and make the task "almost an impossible job," but don't show clear evidence of that. Line 76 states that the benefit of the approach is the addition of "new SA methods in recent years." – which methods? Is that addition the only benefit (in which case it doesn't seem that the expansion to more cases/parameters is nearly impossible in those packages, as stated)?

Thanks for your comments. We apologize for any confusion in this section. Currently, there are various sensitivity analysis (SA) methods, and this paper introduces 5 different SA methods on an equal footing. Additionally, it compares these methods with our own multi-parameter joint perturbation method based on machine learning.

Regarding the second question, despite SCAM having relatively low computational cost, in cases where a large number of executions are required, the computational cost is still non-negligible. Therefore, the introduction of surrogate models can further reduce the computational cost in the experimental process, expediting the experimental progress.

Comment 5: The reference to Pathak et al. (2022) is missing a DOI and is incomplete.

Thanks for your comments. We have improved the expression of this reference[4].

Comment 6: Line 106: In general, the selection of PRECT is perhaps reasonable given its importance societally and the focus on convective parameterizations. Yet it seems like Pathak et al. assessed the response of a number of climate fields to their perturbations, which more closely mimics the process of model tuning. Is it not worthwhile to expand to more indicators of model performance? Near surface T/Q or surface fluxes, for instance, seem to be available for most IOPs.

Thanks for your comments. We highly appreciate your point of view, and in this revision, we have included an exploration of temperature, humidity, and cloud as you suggested.

Comment 7: Line 110: The bounds for parameters are selected to be 50% and 150% of the values; are there not more firm limits based in the literature for what a realistic range may be?

Thanks for your comments. We determined the range for parameter tuning based on general physical principles. The primary consideration for making this choice is the concern that exceeding certain limits may affect computational stability.

Comment 8: Lines 112-114: please indicate which fields you've now made tunable parameters (vs. which ones already were).

Thanks for your comments. In the original code, there are adjustable parameters such as zmconv_c0_lnd, zmconv_c0_ocn, zmconv_ke, cldfrc_rhminh, cldfrc_rhminl, and others, while the rest are not adjustable. After modifying the code, we made all parameters studied in this paper adjustable. We have also provided corresponding descriptions in the text.

Comment 9: Lines 116-118: As before, I'm still not entirely clear on what the authors mean by

saying the programs needed to be recompiled and that the recompilation has enabled a larger number of concurrent instances on the Sunway supercomputer. I believe that this is referring to the fact that all parameters in table 3 are now namelist options, thus the model does not need to be re-built each time and can instead be cloned from a single, pre-built base case whose namelist can be modified. That does cut down on run time, but doesn't seem related to the programs running on Sunway needing recompilation due to the supercomputer planorm. Suggest clarifying that this is simply the shift of hard-coded to namelist defined variables.

Thanks for your constructive comments. Frankly speaking, the platform on which it runs does not ultimately affect the conclusions of this paper. Therefore, in this revision, we no longer emphasize issues related to the platform and compilation, as they are not the focal points we need to address.

Comment 10: Line 174: "we decide another S=768, where NMorris is 64 and NSaltelli is 32." What's the sensitivity to those choices? Does 64 and 32 show some convergence to a solution?

Thanks for your comments. We apologize for the ambiguity introduced here. The intended meaning is that, with this configuration, the same total number of samples can be obtained under both sampling methods. Since these two N values are not critical factors in the experiment and to avoid any misunderstanding, we have removed the corresponding description in the revised version.

Comment 11: Line 187: "Compared to other networks, ResNet has the advantage of using less pooling." What does this mean? Should elaborate on what pooling is first if this is an important point to make. If the following explanation is intended to elaborate on why ResNet is preferred, perhaps rephrase for clarity.

Thanks for your comments. We apologize for the confusion caused here. The main advantages of ResNet (Residual Network) lie in addressing the issues of vanishing gradients and exploding gradients during the training of deep neural networks. In this paragraph, we have also reorganized the corresponding language.

Comment 12: Line 190: "ResNet18..." similar question to above – what's the sensitivity to number of layers? Is 18 enough (and how have the authors determined that)?

Thanks for your comments. In terms of network depth, taking ResNet50 as an example, due to its deeper architecture, ResNet50 has significantly more parameters than ResNet18. While having more parameters can generally make the network more flexible, it may also lead to overfitting. Considering the specific application context and the size of the dataset in this study, ResNet18 is deemed sufficient to meet the requirements of the research.

Comment 13: Lines 208-209: "As the sensitivity values calculated by different methods are of different orders of magnitude..." – was this expected, and arises as a by-product of using different SA methods? I.e., are the units not all in the form of some response driven by a change in the parameter?

Thanks for your comments. We apologize for the confusion caused here. What we intended

to convey is how to identify the most impactful combination of parameters in the scenario of combined parameter perturbation. To make the expression more concise, we have removed the potentially ambiguous sentences in the revised version.

Comment 14: Lines 232-233: "This way, we can determine the number of parameters in the combination, taking into account the tuning effect and the amount of computation." – Not clear what's meant here. What is meant by 'number of parameters in the combination,' isn't that set? How is the 'tuning effect' taken into account, as this seems like a tool that aids in tuning rather than the tuning have an effect on the process.

Thanks for your comments. This refers to whether we should adjust 3 parameters or just 1 or 2 parameters. How to choose which parameters to adjust? Indeed, our method can assist in tuning, but selecting the most appropriate parameters also plays a crucial role in effectively enhancing the tuning effect.

Comment 15: Line 234: "lead to the most significance in output" – should this read the most significant improvement in output, or the largest increase in accuracy? Could be interpreted as the biggest change in output as written currently.

Thanks for your comments. Here, indeed, it refers to "the biggest change in output." We have improved the corresponding descriptions accordingly.

Comment 16: Line 292: "...the distribution of pz4(tau)." –if zmconv_tau in figure, suggest consistent naming conventions. In general, it would be more useful to use the zmconv_tau naming convention as in Fig 3 for Fig. 5 as well, to make it easier for the reader to quickly scan through.

Thanks for your comments. We have revised the corresponding descriptions in the above figures.

Comment 17: Figs 3 and 4 – using consistent colors in Figure 4 to represent the cases shown in Fig. 3 would increase the readability.

Thanks for your comments. We have readjusted the color scheme of the figures to maintain consistency and enhance readability.

Comment 18: Fig 4 – since the main point is the increase in accuracy of PRECT rather than just the change, perhaps there's some way to indicate the change in precipitation bias rather than the overall percent change? It's possible this belongs more in Figure 3 (i.e., gevng a sense of how much the bias has improved given each parameter, rather than just which results are "better" than the control).

Thanks for your comments. Your suggestion is excellent. Following this line of thought, we reanalyzed the samples and improved the corresponding descriptions accordingly.

Comment 19: Line 300: "Different from other cases, this case is a ocean case and is located in the Atlantic Ocean" – This seems to suggest that GATEIII is the only ocean case; does TOGAII not also target ocean? If this isn't the only ocean case, what about it being in the Atlantic makes it particularly unique?

Thanks for your comments. We apologize for the confusion caused by our wording. Indeed, GATEIII does exhibit uniqueness in parameter response, partly due to its classification as an ocean case—although it is not the only case located in the ocean. Another possible contributing factor may be its unique geographical location, which warrants further investigation.

Comment 20: Table 5 –doesn't seem to add much to the paper; could this not just be said in the text?

Thanks for your comments. We have now described the corresponding content in the main text.

Comment 21: Lines 309-310: "It can be seen that ResNet has the best performance…" – The RMSE is indeed the lowest, and not just by a lisle but by quite a margin! Was this fully expected a priori? Is this consistent with any other literature?

Thanks for your comments. This is indeed a noteworthy issue. Therefore, we conducted additional experiments and presented the most recent results in the revised paper.

Comment 22: Figure 5: How are the sensitivity results normalized?

Thanks for your comments. Due to the different computational processes of various Sensitivity Analysis (SA) methods, the sensitivity values obtained are not necessarily on the same scale. To facilitate a straightforward horizontal comparison, we normalized the values using the min-max scaling method.

Comment 23: Lines 322-323: "differences between the individual parameters are not as wide as in the other cases" – not sure what's meant here; that the range in parameter values is smaller? The ranges should be consistent across cases, yes?

Thanks for your comments. We apologize for any confusion caused by our wording. What we intended to convey here is the magnitude of changes in the results. Indeed, as you mentioned, the range of variation in input parameters is entirely consistent across all cases. This ensures a consistent basis for horizontal comparison.

Comment 24: Line 324: "This is also consistent with its position" – by position, is the intent to say that the geographic location is more comparable? Please consider rephrasing for clarity

Thanks for your comments. We apologize for any misunderstanding caused by our wording. We have rephrased this to better convey the intended meaning. Here, we aim to express that cases located at different sites do indeed have different sensitive parameters and responses.

Comment 25: Section 4.3: The discussion of sensitivity to SA method should be expanded and clarified. It is stated that the SA results are shown, but which column is this? The Morris test seems significantly more sensitive to parameter variations than the other tests; are the others better suited for one precipitation regime vs. another? Or how is the different sequence of samples coming into play here?

Thanks for your comments. We apologize for any confusion caused. The six columns here

represent the results of different Sensitivity Analysis (SA) methods, each providing distinct analytical conclusions. Since each SA method operates based on different principles, variations among their results are normal. With normalization applied, the Morris SA method's results, which you mentioned, reflect that the sensitivity differences between parameters are relatively smaller compared to other methods, and this is understandable. Our objective is to identify parameters that are more sensitive to the results (in a relative sense).

As the Morris SA method has specific requirements for the sample sequence, we constructed a sequence tailored to it. Except for the method proposed in this paper (the rightmost column), the other four methods use sequences constructed by the Saltelli method. The sample size for each method is the same.

Comment 26: Lines 357-358: "However, the computation amount increases exponentially during the test." But isn't the benefit of using a surrogate model that you can tune more parameters at a time than using the traditional approach? If all tests for 3 parameters can be conducted in less than a minute, why not tune 4?

Thanks for your comments. When the number of parameters simultaneously adjusted increases, the required computational complexity also grows exponentially. In our practical observations, we found that the computational time of the surrogate model is not negligible when simultaneously adjusting four parameters. Considering the results of the sensitivity analysis presented earlier, the parameter ranked fourth in sensitivity has a relatively minor impact on the outcomes. Therefore, we believe that adjusting three parameters is the most suitable choice.

Comment 27: Figure 7: It's unclear what the max/min stars represent or add to the plot; the min stars also aren't visible from what. I'm unsure why the y-axis is positive only; do none of the parameter combinations lead to a decrease in precipitation?

Thanks for your comments. Here, we are describing the range of changes in the output variable when different parameter combinations are adjusted. Since the y-axis values represent the range of output changes, they are non-negative. We apologize for any confusion caused in this context. After a thorough review of the entire document, we have decided to remove this figure to prevent any misunderstanding.

Comment 28: Line 366: "meaning average error" – should this now be RMSE?

Thanks for your comments. We apologize for the typo that occurred here, and we have corrected the description in this section.

Comment 29: Table 8 is somewhat hard to interpret. Is "error" here referring to the difference in SCAM and the surrogate model? Or is it related to either's agreement with observations? Is SCAM doing a better job simulating PRECT at all locations vs. the surrogate model (given the lower RMSE values) – that doesn't seem to be reflected in the text.

Thanks for your comments. The purpose of this table is to demonstrate that the surrogate

model fits well with the original SCAM, making it suitable for carrying out parameter tuning experiments. We apologize for any confusion caused in this regard and have provided a more concise description of this meaning in this section.

Comment 30: Figure 10: Please clarify if the "model simulation output" is from SCAM or the surrogate model. It also makes more sense to plot the x-axis in time coordinates rather than timestep number.

Thanks for your comments. This refers to the numerical outputs from the SCAM simulation. We sincerely appreciate your suggestion and proceeded to redraw this figure.

Comment 31: Lines 377-378: "The tuning of the SCAM parameters is quite productive on the time scale." – not clear what this means with the current wording. What time scale is being assessed? Does 'productive' mean increased accuracy?

Thanks for your comments. What we intend to convey here is that, after applying the method proposed in this paper, the time spent on parameter tuning for SCAM has been reduced, indicating an improvement in efficiency. We apologize for any confusion caused by our previous explanation, and we have revised this description in the newest version.

Comment 32: Line 381: "Although still below the observed level at about 1300 steps of TOGAII, improvement is also reflected." – I'm not sure why this timestep/case is singled out here, it seems like there are plenty of other examples of this. Similarly, the point is made that in TWP06 there are instances where control is less than observed; again, that's present in all cases, yes?

Thanks for your comments. We apologize for the confusion caused by our wording. Here, our main emphasis is to highlight that, after optimization, the output for each case is closer to the observed values compared to the default parameters, thereby demonstrating the effectiveness of our approach. We have corrected the description in the revised version for clarity.

Comment 33: Section 4.5: I'm not clear why the baseline and optimized models should have any different computational cost, unless baseline refers to SCAM and optimized to the surrogate model. If that's the case, that should be stated much more clearly and oben. More generally, when considering overall model performance, the same model structure should be used. So of course use the surrogate model to find the optimum parameters, but to assess differences in simulation accuracy, plug those values back into SCAM. If not comparing direct SCAM cases, the accuracy gains aren't very understandable.

Thanks for your comments. This section aims to emphasize that, with the use of a surrogate model, the time required for parameter tuning and testing can be significantly reduced. We apologize for any confusion caused, and it is clarified that the ultimate optimization results obtained are indeed plugged into SCAM for execution. We have rephrased this portion in the revised version for better clarity.

Comment 34: Lines 388-389: "The use of NN trained surrogate models for parameter tuning can further save computational resource overhead and, in terms of results, can meet or exceed traditional optimization methods in most cases." – this may be overstated. In terms of saving computational

resources, this is not illustrated by Figure 12. Figure 12 includes time in the job queue for the SCAM case; adding a bar to each case for the computational time excluding queue time would be needed to support this statement.

Thanks for your comments. Indeed, as you mentioned, the queuing time during job execution on the cluster should be excluded. We will use a more reasonable approach to articulate the advantages of the proposed method in terms of computational overhead in this paper.

Comment 35: Lines 391-392: "The model can get an enhancement in performance from 6.4%-24.4% in precipitation output," – again, I think the important part is to plug this back in to SCAM using the parameters determined by the surrogate model. Unless the suggestion is to use the surrogate model for all simulations (rather than for finding optimal parameters and model tuning), this is a critical step.

Thanks for your comments. We apologize for the confusion. Our experimental procedure involved taking the obtained parameter values and plugging them back in SCAM. We have corrected the corresponding description in the manuscript.

Comment 36: Table 9: A better description is needed; "multi" cases undefined here.

Thanks for your comments. The intention here is to aggregate similar cases into the same category in an attempt to have them represent a larger region of parameter responses.

Comment 37: Section 4.6: An important point is lacking from the discussion, which is how well does the model perform (RMSE per site) when using the values found to optimize performance across all 5 sites (multi(d))? The potential benefit of regionally-varying parameters is noted, but it's not illustrated how important is it to use potentially regionally refined values vs. ones that are optimized for global performance.

Thanks for your comments. After careful consideration, we acknowledge that aggregating these combinations simplistically to represent the global scenario may be unfair, despite their individual representativeness within certain ranges. Therefore, in this revision, we have abandoned this attempt and focused more on each case, as well as the commonalities and individualities among them.

Comment 38: Related to above, and though this might be outside what the authors' computational resources allow – the most convincing argument for the study would be to apply the tuned parameters from the multi(d) case to a global run. That would illustrate true benefit from using a surrogate SCAM model to find optimal parameter values; without that, the potential impact of the paper is severely more limited.

Thanks for your comments. We highly agree with your perspective. If the methods proposed in this paper can bring about better performance improvements for the global model, we believe the significance of this paper will be further highlighted. As you mentioned, considering the resources currently available to us, conducting identical experiments on a global scale still poses significant challenges. The focus of this paper lies in leveraging machine learning methods and surrogate models for parameter optimization, as well as exploring the differences

in parameter responses across different regions. We will continue to approach this field with great enthusiasm and look forward to conducting more exploration in the future, expanding and validating its applicability further, given the conditions allow.

**References**

[1]  J. Guo, Y. Xu, H. Fu, W. Xue, L. Wang, L. Gan, X. Wu, L. Hu, G. Xu, and X. Che, "A learning-based method for efficient large-scale sensitivity analysis and tuning of single column atmosphere model (scam)," *Geoscientific Model Development Discussions*, vol. 2022, pp. 1–28, 2022. DOI: 10.5194/gmd-2022-264. [Online]. Available: https://gmd.copernicus.org/preprints/gmd-2022-264/.

[2]  J. W. Hurrell, M. M. Holland, P. R. Gent, S. Ghan, J. E. Kay, P. J. Kushner, J.-F. Lamarque, W. G. Large, D. Lawrence, K. Lindsay, W. H. Lipscomb, M. C. Long, N. Mahowald, D. R. Marsh, R. B. Neale, P. Rasch, S. Vavrus, M. Vertenstein, D. Bader, W. D. Collins, J. J. Hack, J. Kiehl, and S. Marshall, "The community earth system model: A framework for collaborative research," *Bulletin of the American Meteorological Society*, vol. 94, no. 9, pp. 1339–1360, 2013. DOI: 10.1175/BAMS-D-12-00121.1. [Online]. Available: https://journals.ametsoc.org/view/journals/bams/94/9/bams-d-12-00121.1.xml.

[3]  R. B. Neale, C.-C. Chen, A. Gettelman, P. H. Lauritzen, S. Park, D. L. Williamson, A. J. Conley, R. Garcia, D. Kinnison, J.-F. Lamarque, *et al.*, "Description of the ncar community atmosphere model (cam 5.0)," *NCAR Tech. Note NCAR/TN-486+ STR*, vol. 1, no. 1, pp. 1–12, 2010.

[4]  R. Pathak, H. P. Dasari, S. El Mohtar, A. C. Subramanian, S. Sahany, S. K. Mishra, O. Knio, and I. Hoteit, "Uncertainty quantification and bayesian inference of cloud parameterization in the ncar single column community atmosphere model (scam6)," *Frontiers in Climate*, vol. 3, 2021, ISSN: 2624-9553. DOI: 10.3389/fclim.2021.670740. [Online]. Available: https://www.frontiersin.org/articles/10.3389/fclim.2021.670740.

**Replies to Referee #2, GMD-2022-264**

**Jiaxu Guo on behalf of all authors**

Thank you very much for your patient and detailed comments on our work[1]. These valuable comments are very helpful for us to improve this paper. After carefully reading all the questions, we have answered each of them and will make appropriate corrections in the revised version of our manuscript.

In this attachment, the red paragraphs represent refree comments, the blue paragraphs represent comments added by the refree in this round, and the black paragraphs below are our corresponding replies.

Comment 1: In this paper, although the authors evaluate the accuracy of the NN model in terms of precipitation, it probably exists the inconsistent between ML and real model, which don't be highlighted in this paper.

The propose of this manuscript provides a method for efficient tuning of SCAM. This comment question the challenge that the offline surrogate method could not guarantee the optimal solutions in surrogate model can transfer to the real model. The authors try to explain it by the high accuracy of the surrogate model. However, the surrogate models are difficult to learn the optimal solutions because they can't be sampled in the training data. Therefore, the efficiency of this method could be affected. The authors do not statement this point and do not give a solution about this issue.

Thanks for your comments. We fully agree with your point that the optimal solution within the range may not necessarily be present in the samples. This is precisely why we employ machine learning methods to construct a surrogate model, as it adds a more realistic dimension. By learning from the samples, the trained model can capture the relationship between parameter variations and output responses, facilitating quicker identification of the potential optimal solution range. Validating whether a solution obtained from the surrogate model is optimal involves inserting it into SCAM and comparing it with known outputs.

Comment 2: The authors believe that due to the high computational cost of the GCM, the SCAM can be the alternative model for parameter SA and tuning. In reality, the optimal parameters tuned in SCAM could not be suitable for GCM, due to the global regions and more complex large-scale circulation.

I do not figure out a clear path from tuning in SCAM to GCM. The authors claim that "tuning on SCAM cases located in different regions, we can find commonalities and patterns in the parameter response of these cases". However, there are several challenges. The current SCAM can only support

several sites and cannot represent the global catachrestic. Second, there are different optimal parameter values at different sites. Even for each site, there are different local optimal solutions. It is difficult to find the so-called commonalities for these optimal parameters. The forcings are different between SCAM and GCM, optimal parameters could not achieve good performance in GCM.

Thanks for your comments. We apologize for any misunderstanding. Indeed, as you pointed out, the optimal solution from SCAM may not directly apply to GCM to achieve good results due to differences in forcings. However, through the exploration of parameter tuning in SCAM presented in this paper, we can distill valuable methodologies. On one hand, by exploring the simultaneous tuning of three parameters, we achieve improvements in model output performance compared to tuning 1 to 2 parameters. On the other hand, building different surrogate models for different cases reflects that the response to parameters varies across regions. This has positive implications for future parameterization schemes. For example, we can establish distinct models for different regions in GCM, exploring the use of parameters that are most suitable for each region.

Comment 4: The authors separately tune the parameters in SCAM for each site and get the different sensitive parameters and different optimal values. It is difficult to transfer this information to GCM. If the authors can do the multi-objective tuning for these sites with the same parameters, it could be helpful for global model tuning because these SCAM sites indeed represent the different regimes.

For figure 13-14, are the 'ranked 1' points local optimal or global optimal? Different regions (ad) have the different optimal points. It's difficult to determine the optimal point in one region (like a) achieves the optimal solution in the other regions. It's also difficult to find a unified optimal solution for GCM, although it could be helpful to design a new parameterization with different parameter values at different regions.

Thanks for your comments. The points labeled as Rank 1 represent global optima. Indeed, as you mentioned, each region has its own optimal points, and a solution optimal in one region may not necessarily be optimal in other regions. The main significance of Figures 13-14 lies in the exploration of the three-dimensional space formed by the three parameters. It verifies the feasibility of jointly tuning these three parameters. Within the given range of values for the three parameters, we can find a set of more suitable values to improve model performance. This methodology also provides insights for exploring parameter tuning in GCM.

Comment 5: For the workflow, there should be a "metrics" component because it is very important for tuning. No matter SCAM or GCM, the tuning metrics could be the cost function between model simulations and observations. The different designs could affect the optimization. In terms of the metrics, it could consider the 1) different statistic errors between simulation and observation, such as RMSE, performance score like Yang et al. (2013), 2) one objective or multiple objectives, and how to deal with the multiple objectives.

Does the metric function only consider the precipitation? However, it could consider more variables in GCM. It could make sense using time series RMSE in SCAM. In GCM, mean state, spatial correlation, spatial standard deviation, spatial RMSE should be involved in the metrics. The simple metrics in SCAM also pose the challenge for GCM.

Thanks for your comments. We highly appreciate your point of view, and in this revision, we have included an exploration of temperature, humidity, and cloud as you suggested. Indeed, as you mentioned, there are many metrics worth considering. If we delve further into exploration within GCM in the future, it would be reasonable to judiciously leverage these metrics to enhance optimization outcomes.

Comment 6: Line 35: The statement that the Morris SA method cannot get the interactive sensitivity could be wrong. Aurally, the standard deviation of MOAT samplings can stand for the interactive effect of one parameter with others (Morris, 1991).

"However, this is not intuitive enough if the user wants to know directly from a combined perspective which set of parameters has the most significant effect on the results." The sentence is confusing. The mean represents the main effect, and the standard deviation represents the interactive effect.

Thanks for your comments. We apologize for any confusion caused by our previous statement. We have corrected the description in the revised version.

Comment 7: Line 45: as the part of introduction, the authors should explain the challenge of the SA methods, why you choose Morris and Sobol, the computational cost issue, surrogate problems using machine learning. If there are previous works, what's your contribution?

In the revised introduction section, the authors explain why they use Morris and Sobol with the sentence "Both Morris and Sobol are typical SA methods that have a wide range of applications in many fields. As there are already proven application examples". It seems this work use them because the previous work used them. This expression is not serious. There are a lot of sensitive analysis methods. The authors should carefully state the advantage of these two methods. The authors also claim the reason of using surrogate that the SCAM requires not very cheap computational cost. However, it only takes several optimal iterations.

Thanks for your comments. We apologize for any confusion in this section. Currently, there are various sensitivity analysis (SA) methods, and this paper introduces 5 different SA methods on an equal footing. Additionally, it compares these methods with our own multi-parameter joint perturbation method based on machine learning.

Regarding the second question, despite SCAM having relatively low computational cost, in cases where a large number of executions are required, the computational cost is still non-negligible. Therefore, the introduction of surrogate models can further reduce the computational cost in the experimental process, expediting the experimental progress.

Comment 8: Line 55: the authors should do comprehensive literature research, even for GCM, there are a large number of work for tuning, such as Yang et al. (2013) and Zou et al. (2014). In addition, the NN surrogate model is used to tune as well. But the authors don't introduce the previous work and challenge in terms of this issue. The introduce section should be more clear.

I think the explanation of innovation of this manuscript does not make sense. The motivation of this manuscript tried to tune SCAM and transfer to GCM. Since these methods have been applied in

GCM, it is confusing that the authors claim they are not used in SCAM. For the 2nd point, the previous works get the sensitive parameters by perturbing multiple parameters. Then the sensitive parameters can be tuned using optimization algorithms. It is not clear for your point.

Thanks for your comments. We apologize for any confusion. As mentioned in the previous response, the exploration of SCAM does not necessarily involve directly transferring parameter values to GCM. Instead, the process aims to identify patterns and improve the methodology for our research. There may be complex relationships among various parameters, similar to the three-body problem, where adjusting three or more parameters has a collective effect greater than adjusting 1 or 2 parameters. This paper combines machine learning methods with parameter tuning, accelerating the parameter tuning process.

Comment 9: Line 75, Acutely, there are existing SA and tuning workflow used in climate models, such as PSUADE and DAKOTA, the authors don't compare their workflow to these packages. It's not new for the community.

The explanation is not convinced. The authors do not analyze the advantage and disadvantage of these existing framework. Why do not you improve the existing framework? The references of PSUADE and DAKOTA are wrong.

Thanks for your comments. PSUADE and DAKOTA do have their own advantages, but they primarily focus on traditional numerical and statistical methods, and their deployment is not conducive to running on updated clusters. The proposed approach in this paper introduces a machine learning-based surrogate model construction method. It allows for further exploration of perturbations in multi-parameter combinations, making it flexible and easy to deploy.

We apologize for the shortcomings in the previous literature review and have corrected the references for PSUADE[2] and DAKOTA[3].

Comment 11: Table 2 is wrong. Each IOP file includes many variables, not just these four variables. Therefore, the statement that you choose precipitation is wrong.

Although table 2 lists many variables, this manuscript only consider precipitation in the metrics. It should be considered more variables in GCM.

Thanks for your comments. We highly appreciate your point of view, as answered in the previous question, in this revision, we have included an exploration of temperature(T850), humidity(Q850), and cloud(CLDTOT) as you suggested.

Comment 16: Line 184: The 768 samples seem not enough for training NN, do the authors evaluate the performance of NN? In Figure 4, how do you define the accuracy?

The authors do not give any result to prove that the 768 samples are sufficient. The offline surrogate model methods are difficult to achieve high accuracy at optimal local regions. The tuning efficiency is easily affected by the local samples.

Thanks for your comments. The appropriate training sample size depends on the specific

machine learning task, model complexity, and the available data. Generally, a larger sample size allows the model to better learn the features of the data and improve generalization. However, an excessive number of samples may lead to overfitting, especially when the model is relatively simple or the data contains noise.

In the specific context of this paper, a sample size of 768 has proven effective in fitting the parameter responses, considering the complexity of the given scenario. Additionally, 768 is a common multiple of the sample sizes generated under the two sampling methods. Regarding the issue of local samples, we have optimized the workflow of the entire method. The solutions obtained from the surrogate model are validated in SCAM, and the validation results are used to enhance the training process. This refinement contributes to further improving the model's fitting capabilities.

Comment 27: Line 313: pz2 (c0_ocn) should be high influence on the ocean case. But in Figure 8, it doesn't have the high effect on PRECT at TWP. Could you explain the reason?

Unfortunately, the author is not familiar with the parameters. C0_ocn is parameter of autoconversion coefficient over ocean in ZM deep convection scheme. It is not the ocean-land interscection.

Thanks for your comments. We apologize for any misunderstanding caused here, and we have corrected the wording in this section in the current revision.

Comment 29: Line 345: are the 16 iterations enough for convergence? Why don't use the general optimization, such as PSO, GA that you mentioned in the introduction section?

The statement is very misleading. Any optimization algorithm cannot converge in 16 iterations! This comparison is not serious and it requires at least several hundred iterations.

Thanks for your comments. We apologize for any confusion caused here. In the revised version, we have removed the wording that could lead to ambiguity in this context.

**References**

[1]  J. Guo, Y. Xu, H. Fu, W. Xue, L. Wang, L. Gan, X. Wu, L. Hu, G. Xu, and X. Che, "A learning-based method for efficient large-scale sensitivity analysis and tuning of single column atmosphere model (scam)," *Geoscientific Model Development Discussions*, vol. 2022, pp. 1–28, 2022. DOI: 10.5194/gmd-2022-264. [Online]. Available: https://gmd.copernicus.org/preprints/gmd-2022-264/.

[2]  C. Tong, "Problem solving environment for uncertainty analysis and design exploration," in *Handbook of Uncertainty Quantification*. Cham: Springer International Publishing, 2016, pp. 1–37, ISBN: 978-3-319-11259-6. DOI: 10.1007/978-3-319-11259-6_53-1. [Online]. Available: https://doi.org/10.1007/978-3-319-11259-6_53-1.

[3]  K. R. Dalbey, M. S. Eldred, G. Geraci, J. D. Jakeman, K. A. Maupin, J. A. Monschke, D. T. Seidl, A. Tran, F. Menhorn, and X. Zeng, "Dakota, a multilevel parallel object-oriented

framework for design optimization, parameter estimation, uncertainty quantification, and sensitivity analysis: Version 6.16 theory manual," Jan. 2021. DOI: 10.2172/1868423. [Online]. Available: https://www.osti.gov/biblio/1868423.